# Nucleosome interaction of the CPC secures centromeric chromatin integrity and chromosome segregation fidelity

Anjitha Gireesh [1,9], Maria Alba Abad [1,9], Ryu-Suke Nozawa [2,9], Paula Sotelo-Parrilla[3], Léa C Dury [4], Mariia Likhodeeva[3], Martin Wear[5], Christos Spanos [6], Cristina Cardenal Peralta[6], Juri Rappsilber[6,7], Karl-Peter Hopfner [3], Marcus D Wilson [8], Willem Vanderlinden[4], Toru Hirota [2] & A Arockia Jeyaprakash [1,3]✉

## Abstract

The chromosomal passenger complex (CPC; Borealin-Survivin-INCENP-Aurora B kinase) ensures accurate chromosome segregation by orchestrating sister chromatid cohesion, error correction of kinetochore-microtubule attachments, and spindle assembly checkpoint signaling. Correct spatiotemporal regulation of CPC is critical for its function. Phosphorylations of histone H3 Thr3 and histone H2A Thr120 and modification-independent nucleosome interactions involving Survivin and Borealin contribute to CPC centromere enrichment. However, how various nucleosome binding elements collectively contribute to CPC centromere enrichment at the mechanistic level, and whether CPC has any non-catalytic role at centromere remain open questions. Combining the high-resolution cryo-EM structure of a CPC-bound H3Thr3ph nucleosome with atomic force microscopy and biochemical and cellular assays, we demonstrate that CPC employs multipartite interactions, which facilitate its engagement with nucleosome acidic patch and the DNA entry-exit site. Perturbing the CPC-nucleosome interaction compromises chromatin protection against MNase digestion in vitro, and centromeric chromatin stability and error-free chromosome segregation in cells. Our work suggests a non-catalytic chromatin-stabilizing role of CPC in maintaining centromeric chromatin features critical for kinetochore function.

**Keywords** Centromere Integrity; Chromosomal Passenger Complex; Chromosome Segregation; Kinetochore; Mitosis
**Subject Categories** Cell Cycle; Chromatin, Transcription & Genomics; Structural Biology

## Introduction

Faithful distribution of the genetic material between daughter cells during cell division requires correct assembly and regulation of the kinetochore, a multi-subunit protein complex that physically couples chromosomes to the chromosome segregation apparatus, the mitotic spindle (Luykx, 1965; Goldstein, 1981; Uchida et al, 2009; Verdaasdonk and Bloom, 2011; Foley and Kapoor, 2013; Fukagawa and Earnshaw, 2014; Musacchio, 2015). The kinetochore is assembled on a specialized chromosomal region, the centromere, which, in most eukaryotes, is specified by the enrichment of nucleosomes containing CENP-A, a histone H3 variant (Earnshaw and Rothfield, 1985; Earnshaw et al, 2013; Bodor et al, 2014; McKinley and Cheeseman, 2016). While the centromere acts as a chromatin platform to assemble the kinetochore, the inner centromere, the chromatin region where the inter-sister chromatid and the inter-kinetochore axis intersect, acts as a key regulatory site where crucial mitotic regulators concentrate (Earnshaw et al, 1989; Hindriksen et al, 2017). The inner centromere, by recruiting several enzymatic activities (kinases, phosphatases, and motor proteins) serves as a signaling platform contributing to the regulation of sister chromatid cohesion, kinetochore-microtubule attachments and spindle assembly checkpoint (SAC), processes that are fine-tuned by the interplay between the inner centromere associated kinases and phosphatases (Biggins and Murray, 2001; Musacchio, 2010; Funabiki and Wynne, 2013; Hengeveld et al, 2017; Serpico and Grieco, 2020; Valles et al, 2024). The central component of the inner centromere-associated interaction network is the chromosomal passenger complex (CPC). During prometaphase and metaphase, CPC is enriched at the inner centromere and forms part of a highly regulated interaction network together with Heterochromatin Protein 1 (HP1), cohesin, and Shugoshin 1 (Sgo1) (Hindriksen et al, 2017).

[1]Institute of Cell Biology, University of Edinburgh, Edinburgh, UK. [2]Division of Experimental Pathology, Cancer Institute of the Japanese Foundation for Cancer Research, Tokyo, Japan. [3]Gene Center Munich, Ludwig-Maximilians-Universität, Munich, Germany. [4]School of Physics and Astronomy, University of Edinburgh, Peter Guthrie Tait Road, Edinburgh, UK. [5]Protein Production EPPF, University of Edinburgh, Edinburgh, UK. [6]Discovery Research Platform for Hidden Cell Biology, University of Edinburgh, Edinburgh, UK. [7]Bioanalytics, Institute of Biotechnology, Technische Universität Berlin, Berlin, Germany. [8]Institute of Quantitative Biology, Biochemistry and Biotechnology, University of Edinburgh, Edinburgh, UK. [9]These authors contributed equally: Anjitha Gireesh, Maria Alba Abad, Ryu-Suke Nozawa. ✉E-mail: jeyaprakash.arulanandam@ed.ac.uk

CPC is composed of two distinct structural and functional modules. The "localization module" includes Survivin, Borealin, and the N-terminal helix of INCENP, which form a triple helical bundle (Klein et al, 2006; Jeyaprakash et al, 2007). The "kinase module" comprises Aurora B kinase and the C-terminal region of INCENP, commonly referred to as the INCENP IN-box, which is required for full kinase activation (Bishop and Schumacher, 2002; Honda et al, 2003; Elkins et al, 2012).

CPC localization throughout mitosis is highly dynamic, which is directly linked to CPC's wide-ranging functions (Cooke et al, 1987; Parra et al, 2003). During prophase, CPC localizes along the chromosome arms facilitating chromosome condensation (Lipp et al, 2007). During prometaphase and metaphase, CPC is enriched at the inner centromere, where it is implicated in regulating sister chromatid cohesion and destabilizing faulty kinetochore-microtubule attachments (known as error-correction) (Bishop and Schumacher, 2002; Honda et al, 2003; Carmena et al, 2009; Hindriksen et al, 2017; Hengeveld et al, 2017; Haase et al, 2017). During late mitosis, the CPC translocates to the central spindle and finally accumulates at the midbody during telophase, where it controls cell abscission (Carmena et al, 2012; Hadders and Lens, 2022).

Our previous work established that the CPC harbors an intrinsic ability to bind nucleosomes through Borealin-mediated multi-partite interactions with the histone octamer and the nucleosomal DNA, which are essential for initial chromosome association of the CPC and CPC function (Abad et al, 2019). Two mitotic histone marks, phosphorylated Histone H3 Thr3 (H3T3ph) and Histone H2A Thr120 (H2AT120ph), then enrich the CPC at the centromere during prometaphase and metaphase. The H3T3ph mark created by Haspin is directly recognized by the BIR domain of Survivin (Kelly et al, 2010; Wang et al, 2010; Yamagishi et al, 2010; Jeyaprakash et al, 2011). The H2AT120ph mark created by the Bub1 kinase recruits Sgo1, which recruits the CPC mainly through its interaction with Survivin, with auxiliary contribution from Borealin and INCENP (Klebig et al, 2009; Kawashima et al, 2010; Tsukahara et al, 2010; Yamagishi et al, 2010; Trivedi and Stukenberg, 2016; Abad et al, 2022). Furthermore, a conserved positively charged "RRKKRR" motif in INCENP (amino acid residues 65-70) has also been implicated in centromere enrichment of the CPC (Serena et al, 2020).

Although the enrichment of the CPC at centromeres during mitosis and its requirement for accurate chromosome segregation are well established, we still do not understand how various nucleosome-binding elements of the CPC subunits cooperatively allow chromatin binding and what the potential role of CPC is in preserving chromatin structure and integrity. Due to the highly repetitive nature of centromeric DNA, centromeres are known to be inherently fragile (Nassar et al, 2023). Since protection of centromeric identity is crucial for ensuring accurate chromosome segregation and preserving genome identity, multiple mechanisms are proposed to be at play to ensure the protection and stability of centromeric DNA, including DNA methylation, and chromatin-binding proteins (Bakhoum and Cantley, 2018; Black and Giunta, 2018; Mellone and Fachinetti, 2021; Nassar et al, 2023). Furthermore, considering the microtubule-mediated pulling forces exerted on the centromeric chromatin upon chromosome bior-ientation, it is likely that inner centromere-enriched mitotic regulators such as the CPC, Sgo1, and HP1 contribute to chromatin stability along with cohesin-mediated centromere cohesion (Lera et al, 2019; Addis Jones et al, 2019). Using structural biology, protein biochemistry and cell biology, here we provide critical insights into the molecular and structural basis for CPC-nucleosome binding and show that centromeric association of the CPC is required to stabilize centromeric chromatin, which is crucial for error-free chromosome segregation during cell division.

# Results

## CPC binds nucleosomes through its engagement with the nucleosome acidic patch and the DNA entry-exit site

Several studies, including our own previous work, have established the requirement of the N-terminal tail (amino acid residues 1–10; Fig. EV1A) and the loop region of Borealin (amino acid residues 110–206; Fig. EV1A), the INCENP "RRKKRR" motif (amino acid residues 65–70; Fig. EV1B), and the Survivin BIR domain for efficient CPC centromere association (Jeyaprakash et al, 2007; Abad et al, 2019; Serena et al, 2020). To address how different nucleosome binding elements of CPC collectively contribute towards CPC-nucleosome binding and chromosome association, we purified recombinant $CPC_{I1-190SB}$ (containing INCENP 1–190, full-length Survivin and full-length Borealin), which includes the INCENP "RRKKRR" motif and two downstream additional stretches of highly conserved positively charged residues (which we termed basic intrinsically disordered region (basic IDR); Figs. 1A and EV1B), reconstituted a complex with homogeneously modified H3T3 phosphorylated nucleosome core particles (H3T3ph NCPs; using native chemical ligation as described in Abad et al, 2019, Fig. EV1C) and characterized its structure using cryo-electron microscopy (cryo-EM) (Figs. 1B,C, EV1D–F, EV2, and EV4A,B and Table 1).

Our 2.8 Å cryo-EM structure revealed that the highly basic Borealin N-terminal tail, with a pI of 12.01, anchors CPC to the nucleosome through its interaction with the acidic patch (Figs. 1B,C and EV4A-C). This mode of anchoring facilitates the engagement of the downstream triple helical bundle formed by Borealin, Survivin, and INCENP with the nucleosome DNA entry-exit site (Fig. 1C). 3D class analysis of a population of particles with well-defined CPC densities revealed three discrete conformational states of the CPC triple helical bundle (Figs. 2A and EV4D). This, along with 3D variability analysis, shows that CPC, with the Borealin N-terminal region tethered at the nucleosome acidic patch, swings both vertically and horizontally (Figs. 2A and EV4D and Movie EV1). Notably, throughout the data processing stages, we consistently observed a clearly defined map for the bound Borealin N-terminal tail on both acidic patches of the NCP (on both NCP faces) (Fig. EV4E). 3D Refine with Blush Regularization (performed in RELION 5.0, Kimanius et al, 2024), for the particles belonging to Class 2 yielded a 3.8 Å map with clearly defined densities for both CPCs (Fig. 2B). Alternative processing workflow, as outlined in Fig. EV3, employing a combination of particle subtraction and focused classification also yielded a 3 Å map with defined density for two CPCs (Fig. EV4F). Consistent with this, mass photometry analysis of the CPC-NCP complex revealed the presence of a species with molecular weight corresponding to two copies of CPC bound to one NCP (312 ± 34 kDa, 27%; theoretical

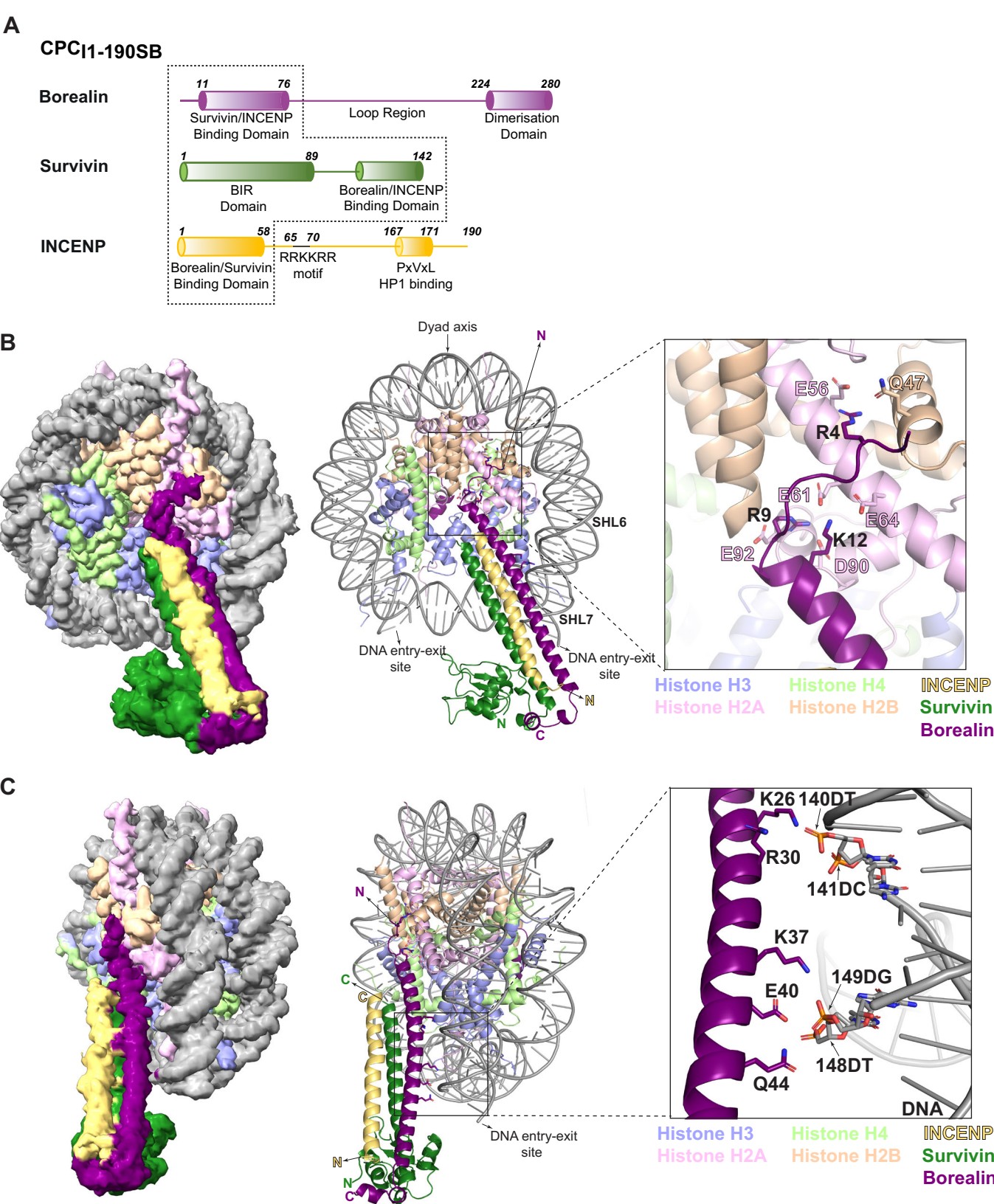

◄  **Figure 1. CPC binds to nucleosomes by interacting with both the nucleosome acidic patch and the nucleosome DNA entry-exit site.**

(A) Schematic diagram depicting the domain architecture of the recombinant CPC used in our cryo-EM studies, CPC$_{I1-190OSB}$, with Borealin full length in purple, Survivin full length in green, and INCENP 1–190 in yellow. Dashed line highlights CPC regions resolved in the cryo-EM structure. (B) Top view of the cryo-EM density of CPC$_{I1-190OSB}$ bound to H3T3ph nucleosome, with the model highlighting the Borealin N-terminal tail and acidic patch interaction. Histones are depicted in violet (Histone H3), light green (Histone H4), pink (Histone H2A), and orange (Histone H2B), and the DNA is shown in gray. CPC subunits are colored in purple (Borealin), green (Survivin), and yellow (INCENP). The zoomed-in image highlights the major residues involved in the salt-bridge interaction, Arg 4, Arg 9, and Lys 12 of the Borealin N-terminal tail (black numbers) making contacts with H2A (pink) and H2B (orange). (C) Cryo-EM density (rotated −60 degrees around y-axis with respect to (B)) with the model highlighting the residues on the Borealin helix engaging with DNA between SHL6 and SHL7. The zoomed-in image depicts the side chains of residues, Lys 26, Arg 30, Lys 37, Glu 40, and Gln 44 in Borealin facing the DNA entry-exit site of the nucleosome. All residues depicted have buried surface area indicative of interaction with DNA. Gln 44 forms H-bonding interaction with the phosphate backbone of Deoxy Guanine at position 149. Source data are available online for this figure.

308.3 kDa), although relatively a smaller population 1:1 CPC:NCP complex was also present (251 ± 18.6 kDa, 8%; theoretical 253.8 kDa) (Fig. 2C). In contrast, the CPC-NCP complex containing a Borealin mutant where Ser 266 located within the dimerization domain was mutated to Asp (S266D) (Fig. 2C) revealed predominantly a 1:1 CPC:NCP complex (256 ± 27 kDa, 17%; theoretical 253.8 kDa) along with a much smaller fraction of a 2:1 complex (309 ± 16.8 kDa, 4%; theoretical 308.3 kDa) (Fig. 2C). Based on this observation along with the cryo-EM structure, we reason that Borealin mediated dimerization of CPC facilitates the binding of CPC on both faces of the NCP.

In this mode of binding, CPC engages with the DNA entry-exit sites on both sides of the NCP, stabilizing the DNA wrapping (Figs. 2B and EV4F). Interestingly, regions in Borealin and INCENP that were previously identified as essential for CPC binding to nucleosomes (Borealin loop region and INCENP "RRKKRR" motif) were not stabilized in this structure, indicating that they likely form conformationally heterogeneous contacts.

Analysis of the intermolecular interactions at the NCP acidic patch showed that Borealin N-terminal tail amino acid residues Arg 4 and Arg 9 form the arginine anchor that interacts with the H2A/H2B Asp/Glu residues of the nucleosome acidic patch, through salt bridge interactions (Figs. 1B and EV5A). Borealin Arg 4 interacts with Gln 47 of H2B, and Glu 56 of H2A, while Borealin Arg 9 contacts Glu 61, Asp 90, and Glu 92 of H2A (Figs. 1B and EV5A). In addition to the arginine anchor, K12 of Borealin interacts with Glu 61 and Glu 64 of H2A (Figs. 1B and EV5A). The CPC$_{I1-190SB}$-H3T3ph NCP structure also revealed that the Borealin helix facing the DNA entry-exit site is highly basic with residues Lys 26, Arg 30, and Lys 37, among which Arg 30 makes Van der Waals contacts with the phosphate backbone of the DNA between super-helical location (SHL) 6 and 7 (Figs. 1C and EV5A). Additionally, Glu 40 has 60% buried surface area indicating an interaction with DNA, and Gln 44 makes H-bonding interactions with the backbone phosphates (OP1 of 149 DG) of nucleotide at the DNA entry-exit site (Figs. 1C and EV5A).

## Conformationally heterogeneous interactions involving Borealin and INCENP are required for efficient CPC-nucleosome binding

Considering that the anchoring of CPC at the NCP acidic patch is mediated via a well-defined network of electrostatic interactions involving Borealin N-terminal tail residues, we wondered if the Borealin-NCP acidic patch interactions are sufficient to achieve CPC-NCP binding. To test this, we reconstituted nucleosomes harboring acidic patch mutations (Glu 56, Glu 61, Asp 90 and Glu 91 on H2A, and

Glu 105 and Glu 113 on H2B, all mutated to Ala) and studied the effect on CPC binding to nucleosomes using electrophoretic mobility shift assays (EMSA) (Fig. 3A). Interestingly, the acidic patch mutations did not abolish CPC binding to nucleosomes, suggesting that other CPC regions also contribute to NCP binding. Furthermore, we purified a CPC complex lacking the N-terminal 10 residues of Borealin (CPC$_{I1-190SBΔN}$) and tested its ability to interact with H3T3ph NCPs using size exclusion chromatography (SEC) (Fig. 3B). SEC analysis shows that CPC$_{I1-190SBΔN}$ can form a complex with H3T3ph NCPs (Fig. 3B). We then quantitatively evaluated the contribution of Borealin N-terminal tail-NCP acidic patch (CPC$_{I1-190SBΔN}$, CPC$_{I1-190SBΔN K12A}$) and the Borealin helix-DNA entry-exit site (Borealin K26/R30/K37A) interactions, separately and in combination (CPC$_{I1-190SB3A}$ and CPC$_{I1-190SBΔN 3A}$), for CPC-NCP binding by performing EMSAs and surface plasmon resonance (SPR) (Fig. EV5B,C). Our data consistently show that the CPCs containing the Borealin N-terminal tail mutants (CPC$_{I1-190SBΔN}$ and CPC$_{I1-190SBΔN K12A}$) bind NCP relatively more weakly compared to CPC with Borealin helix mutant (CPC$_{I1-190SB3A}$) (Figs. 3C–E and EV6A,B). However, none of these Borealin mutants, either in isolation or in combination, abolished CPC-NCP binding.

Altogether, these observations suggested that in addition to the CPC-NCP interactions resolved in the cryo-EM structure, previously well characterized phosphorylated Histone H3 tail interaction with Survivin (Abad et al, 2019), and conformationally heterogeneous interactions involving different regions of Borealin and INCENP may also contribute to efficient CPC-NCP binding.

To identify the conformationally heterogeneous interactions, we performed EDC crosslinking/MS experiments on the CPC$_{I1-190SB}$-H3T3ph NCP complex (Figs. 4A and EV6C). The EDC crosslinking/MS data demonstrated that the additional CPC contacts, involving the Borealin loop region and the INCENP basic IDR, include nucleosome regions previously identified as hotspots for NCP-binding proteins, the H2B C-terminal helix, the H2B α1L1 elbow and the H3 α1L1 elbow (McGinty and Tan, 2021) (Fig. 4A). These nucleosome hotspots mediate intermolecular interactions crucial for various cellular processes, including gene regulation, chromatin remodeling and DNA repair (Lobbia et al, 2021).

We have previously shown that Borealin can directly bind to DNA (Abad et al, 2019). In addition, the INCENP "RRKKRR" motif has also been suggested to mediate DNA binding (Serena et al, 2020). To understand how CPC interacts with nucleosomal DNA, we stabilized protein-DNA interactions within the CPC-NCP complex through UV crosslinking and performed MS analysis (Stützer et al, 2020). MS analysis identified DNA crosslinked peptides for both Borealin and INCENP (Figs. 4B and EV6D). All the DNA-crosslinked peptides of Borealin map to a region spanning the end of the Borealin N-terminal α-helix until the

**Table 1. Details of cryo-EM density maps and atomic models.**

| | CPC-NCP Class 0 | CPC-NCP Class 1 | CPC-NCP Class 2 | CPC-NCP Double occupancy (RELION) | CPC-NCP Double occupancy (cryoSPARC) |
|---|---|---|---|---|---|
| Number of grids used | 1 | 1 | 1 | 1 | 1 |
| Grid type | Quantifoil R2/2 300 mesh | Quantifoil R2/2 300 mesh | Quantifoil R2/2 300 mesh | Quantifoil R2/2 300 mesh | Quantifoil R2/2 300 mesh |
| Microscope/detector | Titan Krios/Falcon4i | Titan Krios/Falcon4i | Titan Krios/Falcon4i | Titan Krios/Falcon4i | Titan Krios/Falcon4i |
| Voltage | 300 kV | 300 kV | 300 kV | 300 kV | 300 kV |
| Magnification | 165k | 165k | 165k | 165k | 165k |
| Recording mode | Counting mode | Counting mode | Counting mode | Counting mode | Counting mode |
| Dose rate | 1 e$^-$/Å$^2$/frame | 1 e$^-$/Å$^2$/frame | 1 e$^-$/Å$^2$/frame | 1 e$^-$/Å$^2$/frame | 1 e$^-$/ Å$^2$/frame |
| Defocus | −0.5 μm to −2.6 μm (step size 0.3 μm) | −0.5 μm to −2.6 μm (step size 0.3 μm) | −0.5 μm to −2.6 μm (step size 0.3 μm) | −0.5 μm to −2.6 μm (step size 0.3 μm) | −0.5 μm to −2.6 μm (step size 0.3 μm) |
| Pixel size | 0.727 | 0.727 | 0.727 | 0.727 | 0.727 |
| Total dose | 40 e$^-$/Å$^2$ | 40 e$^-$/Å$^2$ | 40 e$^-$/Å$^2$ | 40 e$^-$/Å$^2$ | 40 e$^-$/Å$^2$ |
| Number of frames/ movie | 40 | 40 | 40 | 40 | 40 |
| Total exposure time | Adjusted to keep the total dose stable | Adjusted to keep the total dose stable | Adjusted to keep the total dose stable | Adjusted to keep the total dose stable | Adjusted to keep the total dose stable |
| Number of micrographs | 22,065 | 22,065 | 22,065 | 22,065 | 22,065 |
| Number of micrographs used | 16,971 | 16,971 | 16,971 | 16,971 | 16,971 |
| Number of particles used | 73,078 | 74,429 | 75,653 | 75,653 | 28,398 |
| PDB | 9SI9 | 9SJ5 | 9SI3 | 9SLJ | – |
| EMDB | EMD-54926 | EMD-54938 | EMD-54924 | EMD-55003 | EMD-55012 |
| Map resolution (FSC 0.143) | 2.86 | 2.85 | 2.83 | 3.8 | 3 |
| Refinement (Phenix) | | | | | |
| Model resolution (Å) (FSC 0.5) | 3 | 3 | 3 | 3.2 | – |
| Map CC | 0.88 | 0.87 | 0.87 | 0.87 | – |
| Mean B factor (Å$^2$) | 90.13 (Protein) 83.89 (Nucleotide) | 100.54 (Protein) 92.49 (Nucleotide) | 95.34 (Protein) 90.97 (Nucleotide) | 358.08 (Protein) 120.65 (Nucleotide) | |
| Validation | | | | | – |
| All atom clashscore | 8.12 | 8.35 | 5.10 | 6.49 | – |
| Rotamer outliers (%) | 1.01 | 0.9 | 0.00 | 2.99 | – |
| MolProbity score | 1.63 | 1.56 | 1.44 | 1.91 | – |
| Ramachandran plot | | | | | |
| Favored (%) | 96.88 | 97.46 | 97.06 | 96.9 | – |
| Outliers (%) | 0.00 | 0.00 | 0.00 | 0.00 | – |
| RMS deviation | | | | | |
| Bond length (Å) | 0.005 (0) | 0.004 (0) | 0.004 (0) | 0.004 (0) | – |
| Bond angle (°) | 0.621 (0) | 0.594 (1) | 0.803 (1) | 0.654 (2) | – |

N-terminal half of the Borealin loop region. This is consistent with our previous SPR data, which suggested Borealin amino acid residues 110–188 as a region capable of directly binding DNA (Abad et al, 2019). The DNA-protein contacts in INCENP spanned not only IDRs, but also the N-terminal α-helix, which is part of the triple helical bundle formed by Borealin, Survivin, and INCENP, suggesting the capability of this helical bundle in making transient inter-nucleosomal contacts via INCENP under favorable conditions (Figs. 4B and EV6D). Our analysis also identified a DNA-crosslinked peptide spanning residues 71–83, which is adjacent to the "RRKKRR" motif, suggesting the capability of this region to interact with DNA (although we could not identify any DNA-

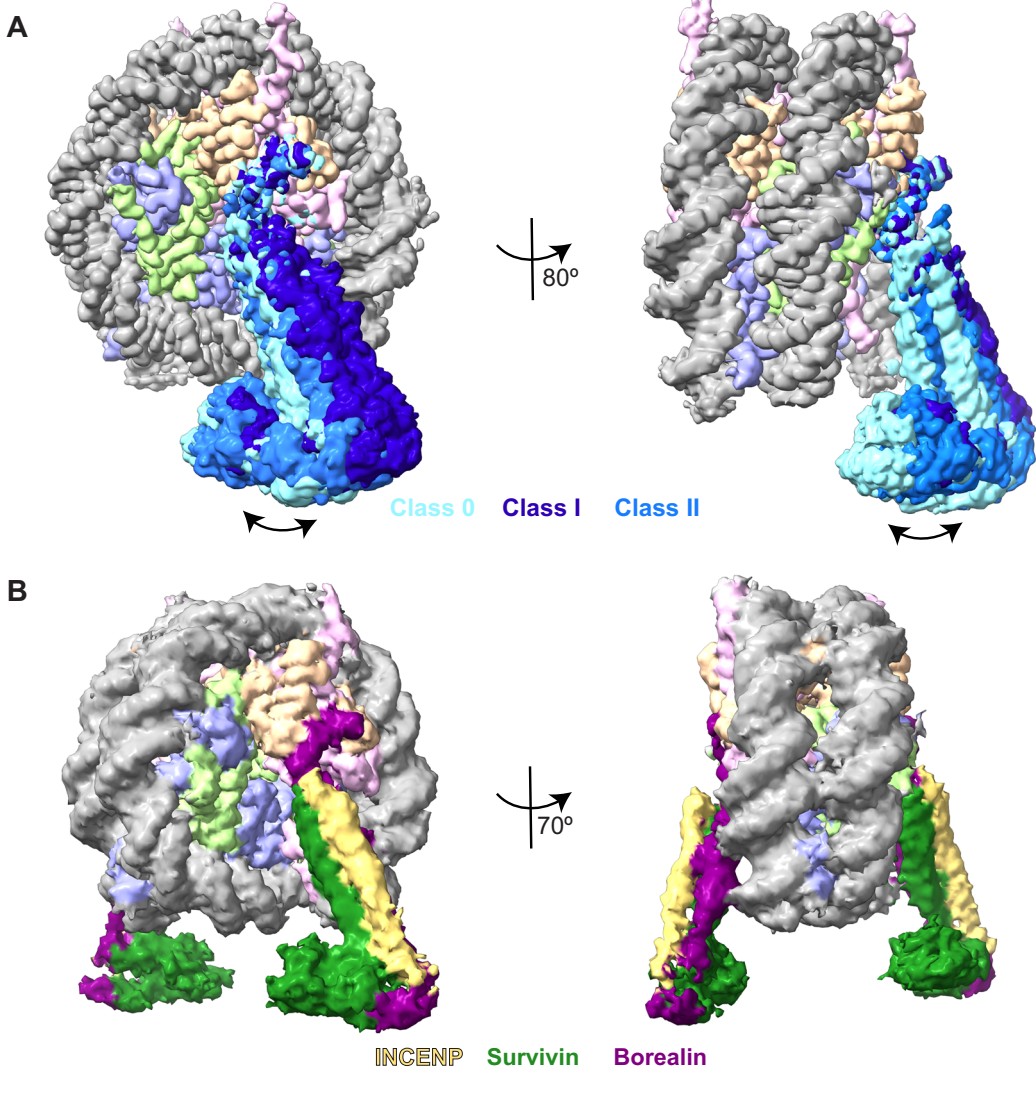

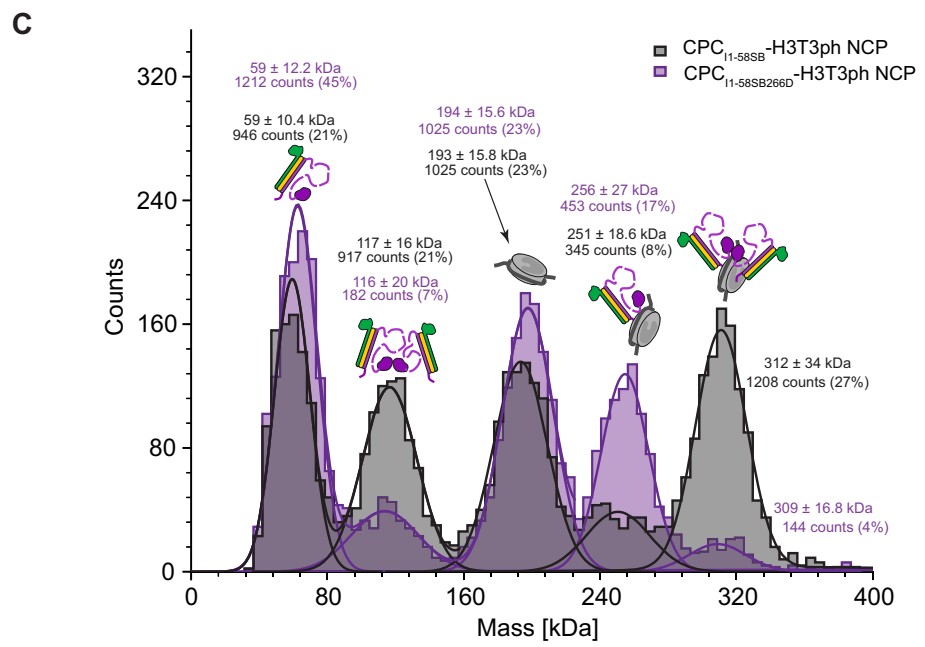

**Figure 2. Nucleosome-bound CPC displays a swinging motion, and CPC occupies both faces of the nucleosome.**

(A) Composite density map of selected classes from 3D classification (Class 0, 1, and 2) depicting the swinging motion of CPC. Histones are depicted in violet (Histone H3), light green (Histone H4), pink (Histone H2A), and orange (Histone H2B), and the DNA is shown in gray. The three classes representing the separate positions of CPC are shown in different shades of blue. CPC has flexibility in both directions, both horizontally and vertically, while being tethered to the nucleosome via the N-terminal tail of Borealin. The left image depicts the horizontal flexibility of the complex, while the right image shows the vertical motion of CPC. (B) Cryo-EM density map obtained after RELION processing (3D Refine with Blush regularization) depicting the double occupancy of CPC. CPC densities can be seen binding to both faces of the nucleosome. The nucleosome components and CPC densities are colored as in Fig. 1B. (C) Count distribution of the mass photometry analysis for $CPC_{I1-58SB}$ (containing INCENP 1–58, full-length Survivin and full-length Borealin; gray), and $CPC_{I1-58SBS266D}$ (containing INCENP 1–58, full-length Survivin and full-length Borealin dimer breaking mutant, S266D; purple) in complex with H3T3ph NCPs. Gaussian fitting identified multiple peaks corresponding to different species. The corresponding masses in kDa and event counts are indicated for each peak. Cartoons depicting the oligomerization state of the species in each peak are also indicated above the peak. CPC subunits in the cartoon are colored as in (B), and the nucleosome is shown in gray. Source data are available online for this figure.

crosslinked peptides involving the "RRKKRR" motif, possibly due to the presence of a large number of trypsin cleavage sites). Overall, this data suggests that DNA interactions involving both Borealin and INCENP also contribute to CPC-nucleosome binding.

To directly assess the contribution of INCENP "RRKKRR" and the Borealin loop region for nucleosome binding, we performed EMSA assays with different versions of CPC containing either INCENP with "RRKKRR" mutated to "AAAAAA" ($CPC_{I1-190\ 6ASB}$) or Borealin lacking the loop ($CPC_{I1-190SB\Delta loop}$) and H3T3ph nucleosomes reconstituted with either Widom 601 or centromeric α-satellite DNA (Figs. 4C–E and EV5B,C). EMSA analysis showed that while $CPC_{I1-190\ 6ASB}$ bound to Widom 601 NCPs with a similar affinity as the wild type (wt) CPC complex ($CPC_{I1-190SB}$), the $CPC_{I1-190SB\Delta loop}$ showed a 12-fold decrease in affinity compared to $CPC_{I1-190SB}$ (Fig. 4C–E). Interestingly, the INCENP "RRKKRR" mutant ($CPC_{I1-190\ 6ASB}$) bound weaker (though moderately, ~1.5 times) to α-satellite NCPs compared to $CPC_{I1-190SB}$. However, the contribution of the INCENP "RRKKRR" motif to α-satellite NCP-binding was even stronger (~13 times) when combined with the Borealin mutant lacking the loop region ($CPC_{I1-190\ 6ASB\Delta loop}$). These observations suggest that both the Borealin loop region and the INCENP "RRKKRR" motif contribute to CPC-nucleosome affinity, and, in agreement with Serena et al (2020), that the contribution of INCENP to nucleosome binding might be DNA sequence-dependent.

Altogether, our structural and biochemical analysis shows that CPC-NCP binding is mediated by multipartite interactions involving both structurally defined and intrinsically disordered regions of the CPC subunits. This, along with our observation that CPC engages with the nucleosome DNA entry-exit site on both faces of the NCP suggests that CPC likely stabilizes chromatin by protecting the wrapped state of nucleosomal DNA.

## CPC stabilizes nucleosomes through compaction

Atomic force microscopy (AFM) enables the visualization of DNA wrapping states around histone proteins within asymmetrically wrapped nucleosomes, providing insights into their conformational landscape (Lyubchenko, 2014; Konrad et al, 2021b, 2022). To assess if CPC binding stabilizes nucleosomal DNA wrapping, we performed AFM imaging experiments with H3T3ph NCPs reconstituted with a 486 bp DNA construct (comprising the Widom 601 nucleosome positioning sequence flanked by a short-106 bp and long-233 bp extra-nucleosomal DNA arm; Fig. 5A) in the absence and presence of $CPC_{I1-190SB}$. In the absence of $CPC_{I1-190SB}$, we found that 40 ± 1% (error is SD) of H3T3ph nucleosomes

occur in the fully wrapped state, and 60 ± 1% occur in a partially unwrapped state. Addition of $CPC_{I1-190SB}$ increased the fraction of fully wrapped H3T3ph nucleosomes to 52 ± 2% (Figs. 5B,C and EV7A–F). Moreover, the presence of $CPC_{I1-190SB}$ led to a highly significant change in the nucleosome wrapping landscape towards a smaller opening angle (Figs. 5C,D and EV7A–F), strengthening our hypothesis that CPC binding stabilizes the fully wrapped state of nucleosomes.

We then assessed the extent of nucleosomal DNA protection conferred by CPC binding. We incubated CPC complexes containing different Borealin and INCENP mutants ($CPC_{I1-190\ 6ASB}$, $CPC_{I1-58SB}$, $CPC_{I1-190SB\Delta loop}$, and $CPC_{1-190SB\Delta N}$; Fig. EV8A) with H3T3ph nucleosomes reconstituted with either Widom 601 or centromeric α-satellite DNA and performed Micrococcal nuclease (MNase) assays (Figs. 5E and EV8C). MNase data show that the DNA wrapped around the H3T3ph histone octamer was protected in the presence of $CPC_{I1-190SB}$, while removing the Borealin loop region ($CPC_{I1-190SB\Delta loop}$) and either mutating the INCENP "RRKKRR" motif ($CPC_{I1-190\ 6ASB}$) or deleting the entire basic IDR of INCENP ($CPC_{I1-58SB}$), all showed a comparable decrease in DNA protection. We also observed a similar reduction in DNA protection when we perturbed the Borealin interaction anchoring CPC to the nucleosome acidic patch ($CPC_{I1-190SB\Delta N}$).

To assess if CPC can exert similar protection against MNase activity in the context of chromatin, we performed the above-described MNase assay with a 12-mer H3T3ph nucleosomal array (Figs. 5F and EV8B,D) including additional structure-guided mutants designed to evaluate the individual and additive contributions of different interaction interfaces ($CPC_{I1-190SB\Delta N}$, $CPC_{I1-190SB\Delta N\ K12A}$, $CPC_{I1-190SB3A}$, $CPC_{I1-190SB\Delta N\ K12A\ 3A}$). In the absence of CPC, 2 min of MNase digestion resulted in mono- and di-nucleosomes (Figs. 5F and EV8D). Addition of $CPC_{I1-190SB}$ resulted in the protection of the high MW array species (4-mer to 12-mer; Figs. 5F and EV8B,D). Consistent with our mononucleosome MNase data, all Borealin and INCENP mutants showed reduced MNase protection activity (Figs. 5F and EV8B,D).

## CPC-nucleosome interaction contributes to centromeric chromatin protection

Based on our structural and in vitro biochemical data presented above, we hypothesized that CPC-nucleosome binding may be critical for maintaining correct centromeric chromatin structure and compaction. To test this, we generated RPE1 cell lines expressing Borealin-mNeonGreen in which endogenous INCENP could be replaced either with FLAG-INCENP wild type or FLAG-INCENP 6A mutant or RPE1 cell lines expressing Borealin-mClover in which endogenous Borealin could be replaced either

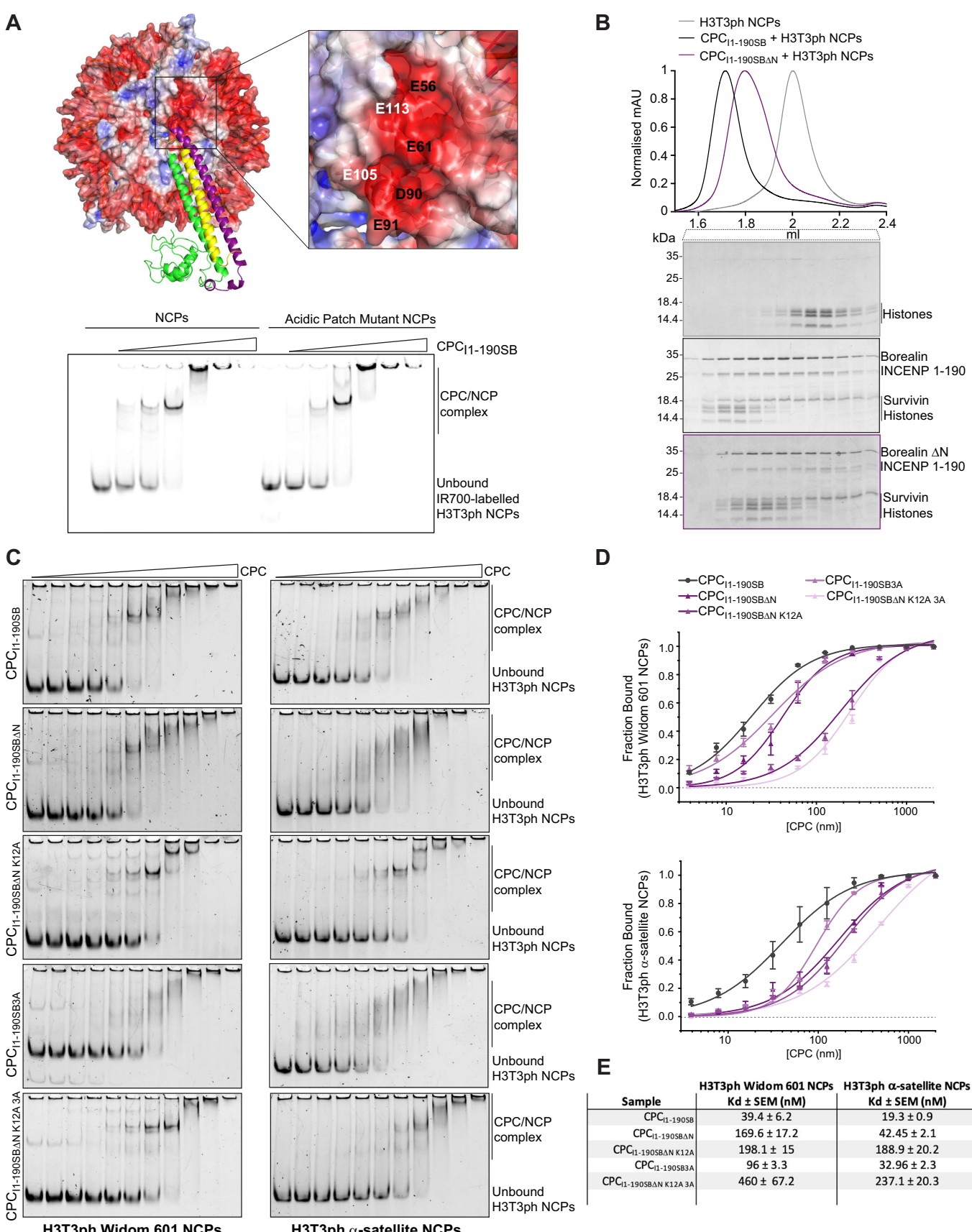

◀

**Figure 3. Structurally defined CPC-nucleosome interactions are not the sole contributors towards CPC-NCP binding.**

(A) Representative native PAGE of EMSA assay performed with 20 nM wild type NCPs and Acidic patch mutant NCPs (Glu 56, Glu 61, Asp 90 and Glu 91 on H2A, and Glu 105 and Glu 113 on H2B all mutated to Ala) with increasing amounts of recombinant CPC$_{I1-19OSB}$ (CPC containing INCENP 1–190, full-length Survivin and full-length Borealin; from 20 nM to 640 nM). Top panel shows the cryo-EM model surface colored by electrostatic potential (APBS using Pymol version 3.1) and showing the acidic patch residues mutated in the EMSA analysis (H2A in black and H2B in white). (B) Size exclusion chromatogram and corresponding SDS-PAGE analysis for H3T3ph nucleosomes (gray), CPC$_{I1-19OSB}$-H3T3ph NCP complex (black), and CPC$_{I1-19OSBΔN}$-H3T3ph NCP complex (CPC containing INCENP 1–190, full-length Survivin, and Borealin 11-end; purple). The shift on the purple profile towards the left indicates complex formation, as observed in the SDS-PAGE. (C) Representative native gels of EMSAs performed with H3T3ph NCPs wrapped with Widom 601 (left) and centromeric α-satellite (right) DNA in the presence of a concentration series ranging from 7.8 nM to 2 μM of five different CPC constructs (CPC$_{I1-19OSB}$, CPC$_{I1-19OSBΔN}$, CPC$_{I1-19OSBΔN\ K12A}$, CPC$_{I1-19OSB3A}$, and CPC$_{I1-19OSBΔN\ K12A\ 3A}$). Bands corresponding to the CPC-NCP complex and unbound H3T3ph NCPs are indicated on the right. For easy direct comparison, the representative native PAGE and the EMSA results for the control protein, CPC$_{I1-19OSB}$, are shown in two different panels ((C, D) and Fig. 4C,D). (D) Quantification of the EMSAs in (C), where the fraction of unbound NCPs is plotted against a log10 x-axis of the CPC concentration and an isotherm fitted using the "specific binding curve with Hill slope" fitting. The equation for this fitting is: $Y = Bmax * X\hat{}h/(K_d\hat{}h + X\hat{}h)$, where $Y$ is the specific binding, $X$ is the protein concentration, Bmax is the maximum fraction bound, $K_d$ is the dissociation constant, and $h$ is the Hill slope ($n \geq 3$, mean ± SEM, from at least three biological replicates). (E) Table showing $K_d$ values calculated from the EMSAs for Widom 601 NCPs and α-satellite NCPs for all the constructs tested in (D). Source data are available online for this figure.

with Borealin wt-mCherry, Borealin lacking the N-terminal tail (Borealin ΔN-mCherry) or Borealin helix mutant (Borealin K26/R30/K37/E40/Q44A-Borealin 5A-mCherry, Fig. EV9A,B) and performed MNase digestion of chromatin on prometaphase lysates. Supporting our hypotheses, cells expressing INCENP 6A, Borealin ΔN, and Borealin 5A, all showed reduced centromeric chromatin protection against MNase digestion (Figs. 6A,B and EV9C,D,G).

We then assessed the consequence of perturbing the nucleosome-stabilizing role of CPC on the centromere enrichment of CPC by performing fluorescence recovery after photobleaching (FRAP) analysis in prometaphase cells and by quantifying the bulk centromeric levels of CPC mutants (Figs. 6C and EV9A,B,G). Cells expressing either Borealin-mNeonGreen or Borealin-mCherry were photobleached at centromeres, and the signal recovery was determined from time-lapse images. FRAP analysis showed that fluorescence recovery of the CPC containing the INCENP 6A and the Borealin ΔN mutants is faster compared to the CPC containing INCENP wt or Borealin wt, indicating that perturbing the INCENP "RRKKRR" or the Borealin N-terminal tail contributions to CPC-nucleosome binding leads to less stable centromere association of CPC during prometaphase (Fig. 6C). In contrast, fluorescence recovery of Borealin 5A is comparable to that of Borealin wt. In agreement with the FRAP data, the reduction in the centromeric level of Borealin 5A is only moderate as compared to either Borealin ΔN or INCENP 6A (Fig. EV9E,F). This suggests that these mutants might influence the centromeric chromatin differently, leading to differential effects on CPC dynamics. However, live cell imaging experiments carried out with the RPE1 cell lines showed that all INCENP or Borealin mutants displayed a delay in mitotic progression from chromosome alignment to onset of anaphase (Fig. EV10A).

It has previously been shown that when centromeric chromatin integrity is perturbed, the inter-kinetochore distance can either increase or decrease in response to the tension applied on the kinetochores by microtubule-pulling forces (Gassmann et al, 2007; Ribeiro et al, 2009; Uchida et al, 2009; Maresca and Salmon, 2009; Suzuki et al, 2014; Etemad and Kops, 2016; Uchida et al, 2021; Chen et al, 2021). Thus, to further understand the role of CPC for centromeric chromatin organization, we labeled CENP-A and CENP-B and quantified the inter-kinetochore distance in

RPE1 cells expressing either the INCENP mutant or the structure-guided Borealin mutants (Fig. 6D). Our data shows that INCENP 6A and Borealin 5A mutants show altered inter-kinetochore distance (Fig. 6D). While the INCENP "RRKKRR" mutant (INCENP 6A) showed an increase in inter-kinetochore distance, Borealin 5A (perturbing the Borealin helix-DNA entry-exit site interaction) resulted in a decrease in inter-kinetochore distance (Fig. 6D). In contrast, the Borealin ΔN mutant did not show any detectable variation in the inter-kinetochore distance (Fig. 6D). It is important to note that, unlike INCENP 6A and Borealin 5A, Borealin ΔN retains likely all DNA-interacting regions, suggesting that CPC exerts its role in maintaining correct centromere features mainly via its DNA-binding properties in vivo. The observed difference in the phenotypes for INCENP 6A and Borealin 5A (increase vs decrease in inter-kinetochore distance, respectively; Fig. 6D) could be due to the differential effect of these mutants on centromeric chromatin. We speculate that INCENP 6A impacts the inner centromeric chromatin, compromising centromeric chromatin's ability to withstand spindle-associated pulling forces, while Borealin 5A impacts the kinetochore proximal centromeric chromatin, compromising the ability of the kinetochore to transfer the microtubule-associated force to centromeric chromatin.

Notably, analysis of segregation errors in RPE1s released from a monastrol-induced mitotic arrest showed that the INCENP "RRKKRR" mutant and the structure-based Borealin mutants all led to an increase in cells with chromosome segregation errors during anaphase (Fig. 6E).

Overall, this cellular data, along with the in vitro biochemical analysis, highlights the requirement of the chromatin-stabilizing role of CPC in ensuring accurate cell division.

## Discussion

Efficient enrichment of CPC at the inner centromere during early stages of mitosis is essential for ensuring faithful chromosome segregation. Over the years, numerous studies have investigated CPC localization and function (Klebig et al, 2009; Kelly et al, 2010; Wang et al, 2010; Yamagishi et al, 2010; Kawashima et al, 2010;

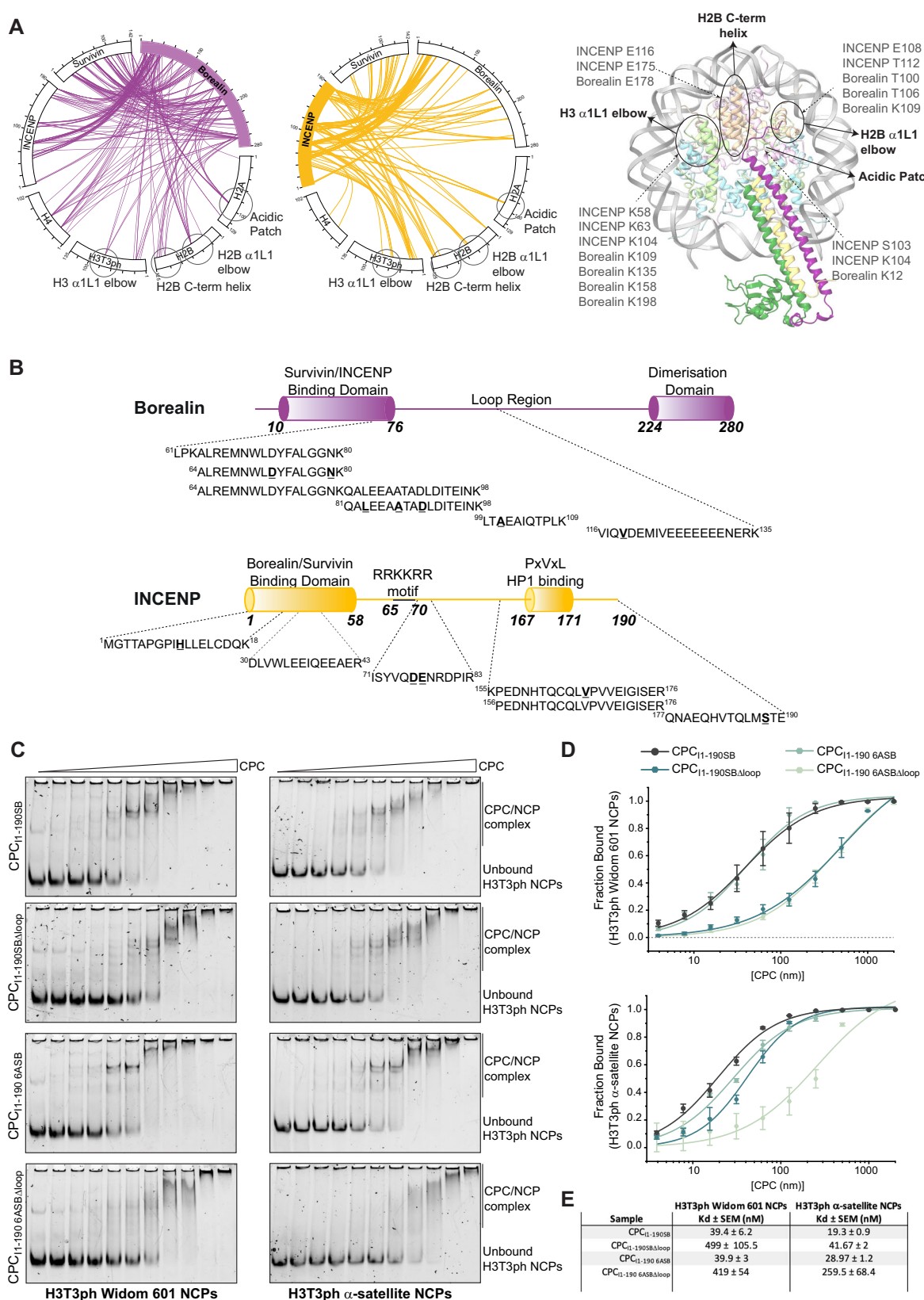

**Figure 4. Efficient CPC-nucleosome binding relies on multipartite interactions.**

(A) Circular diagram of the EDC protein crosslinks observed for the CPC$_{I1-190SB}$-H3T3ph NCP complex. Borealin and INCENP crosslink with histones are shown in purple and yellow, respectively. On the right, crosslinked regions between CPC components and the histones have been mapped on the cryo-EM model. The four main hotspots of nucleosome binding are highlighted in the crosslinking diagram and the model: the acidic patch, the Histone H3 α1L1 elbow, the Histone H2B C-term helix, and the Histone H2B α1L1 elbow. Both Borealin and INCENP make extensive contacts with these hotspots. (B) Protein-DNA crosslinks obtained through MS analysis of the UV crosslinked CPC-NCP complex mapped onto the domain architecture of Borealin and INCENP$_{1-190}$. Crosslinked residues are bold and underscored. (C) Representative native gels of EMSAs performed with H3T3ph NCPs wrapped with Widom 601 (left) and centromeric α-satellite (right) DNA in the presence of a concentration series ranging from 7.8 nM to 2 μM of three different CPC constructs (CPC$_{I1-190SBΔloop}$, CPC$_{I1-190 6ASB}$, and CPC$_{I1-190 6ASBΔloop}$). Bands corresponding to the CPC-NCP complex and unbound H3T3ph NCPs are indicated on the right. For easy direct comparison, the representative native PAGE and the EMSA results for the control protein, CPC$_{I1-190SB}$, are shown in two different panels (Fig. 3C,D and Fig. 4C,D). (D) Quantification of the EMSAs in (C), where the fraction of unbound NCPs is plotted against a log10 x-axis of the CPC concentration and an isotherm fitted using the "specific binding curve with Hill slope" fitting. The equation for this fitting is: $Y = Bmax * X^h/(K_d^h + X^h)$, where $Y$ is the specific binding, $X$ is the protein concentration, Bmax is the maximum fraction bound, $K_d$ is the dissociation constant, and $h$ is the Hill slope ($n \geq 3$, mean ± SEM, from at least three biological replicates). (E) Table of $K_d$ values calculated from the EMSAs for Widom 601 NCPs and α-satellite NCPs for all the constructs tested in (D). Source data are available online for this figure.

Tsukahara et al, 2010; Jeyaprakash et al, 2011; Trivedi and Stukenberg, 2016; Abad et al, 2019; Serena et al, 2020; Abad et al, 2022). However, how the different components of CPC collectively contribute towards chromatin interaction, and whether CPC has any Aurora B kinase-independent role in influencing chromatin organization and/or integrity, are yet to be understood.

In this study, we aimed to answer these long-standing questions by characterizing CPC-nucleosome interaction employing an integrative structure-function approach combining structural, biochemical, and cell-based approaches. The high-resolution cryo-EM structure of CPC bound to H3T3ph NCP reported here identifies interactions of CPC with nucleosome acidic patch and the nucleosomal DNA entry-exit site, mediated via a highly basic Borealin N-terminal tail and the triple helical bundle formed by Borealin, Survivin, and INCENP, respectively. We speculate that the engagement of Borealin N-terminal tail along Lys 12 with the acidic patch of NCP may facilitate the triple helical bundle interaction by restricting the mobility of the triple helical bundle. This mode of nucleosome binding by CPC broadly aligns with the recent observations reported elsewhere (Ruza et al, 2025).

Crosslinking/MS analysis and further biochemical characterization revealed conformationally heterogeneous nucleosome interactions involving the Borealin loop and INCENP basic IDR, underscoring their crucial role for efficient CPC-NCP binding. We previously showed that the Borealin dimerization domain contributes to efficient nucleosome binding, but how it does so remained unclear (Abad et al, 2019). Here, we show that a nucleosome can bind two copies of CPC in vitro and, in line with this, two copies of CPC engage with both faces of the nucleosome (related by a twofold symmetry) in our cryo-EM structure, suggesting a potential contribution of Borealin dimerization in facilitating this interaction. Supporting this notion, in the Ruza et al (2025) cryo-EM structure, just one copy of CPC was bound to the nucleosome, possibly because their CPC construct lacked more than two-thirds of the C-terminal region of Borealin (including the Borealin loop and dimerization domain). Considering the interactions observed between the CPC triple helical bundle and the DNA entry-exit site, it is likely that Borealin dimerization-mediated engagement of CPC on both faces of NCP may contribute to both stabilizing the wrapped state of DNA as well as chromatin compactness via inter-nucleosomal contacts.

Strikingly, the occupation of two CPCs on both faces of the nucleosome, as observed in our structure, while facilitating the interaction of the CPC triple helical bundle with nucleosome DNA entry-exit sites, completely occludes the H2A C-terminal tail, making it inaccessible to its binding partners. Moreover, EDC crosslinking/MS analysis showed that the H2A C-terminal tail residue T120 interacts with the CPC triple helical bundle. Phosphorylation of H2A Thr 120 by Bub1 is known to be essential for Sgo1 localization to centromeres and for Sgo1's role in protecting cohesion and preventing premature sister chromatid separation (Kitajima et al, 2005; Kawashima et al, 2010; Liu et al, 2013; Hengeveld et al, 2017; Broad et al, 2020). Remarkably, the NCP binding mode of the CPC occludes the accessibility of the histone H2A tail for Sgo1 binding (Fig. EV9H). These observations, together with our earlier study showing that the CPC binding of both Sgo1 and H3T3ph requires the same binding pocket on the Survivin BIR domain, suggest that the H3T3ph-mediated and the H2AT120ph-mediated CPC binding are mutually exclusive and likely to be restricted spatially (H3T3ph-mediated inner centromere pool vs H2AT120-mediated kinetochore proximal pool) and possibly also temporally (Liang et al, 2020; Broad et al, 2020; Hadders et al, 2020; Abad et al, 2022; Cairo et al, 2023).

Haase et al (2017) previously reported an Aurora B kinase-independent non-catalytic role of CPC in facilitating kinetochore assembly. Their work showed that centromere association of the CPC localization module was sufficient to rescue the inner kinetochore assembly defects observed upon CPC depletion (Ditchfield et al, 2003; Wang et al, 2011; Wheelock et al, 2017; Haase et al, 2017). Our data reported here show that CPC-nucleosome binding stabilizes nucleosomal array in vitro and centromeric chromatin in vivo. Interestingly, perturbing either the INCENP "RRKKRR"- nucleosome interaction or Borealin helix-nucleosome DNA entry-exit site interaction led to perturbed inter-kinetochore distance. We speculate that the CPC, via its non-catalytic chromatin binding activity, likely contributes to stabilizing and maintaining the correct local architecture of the centromeric chromatin crucial for functional kinetochore assembly and withstanding microtubule-pulling forces. It is also conceivable that CPC and the associated inner centromere interaction network (involving Sgo1, HP1, and cohesin) might also contribute to CPC-mediated tension sensing and error correction.

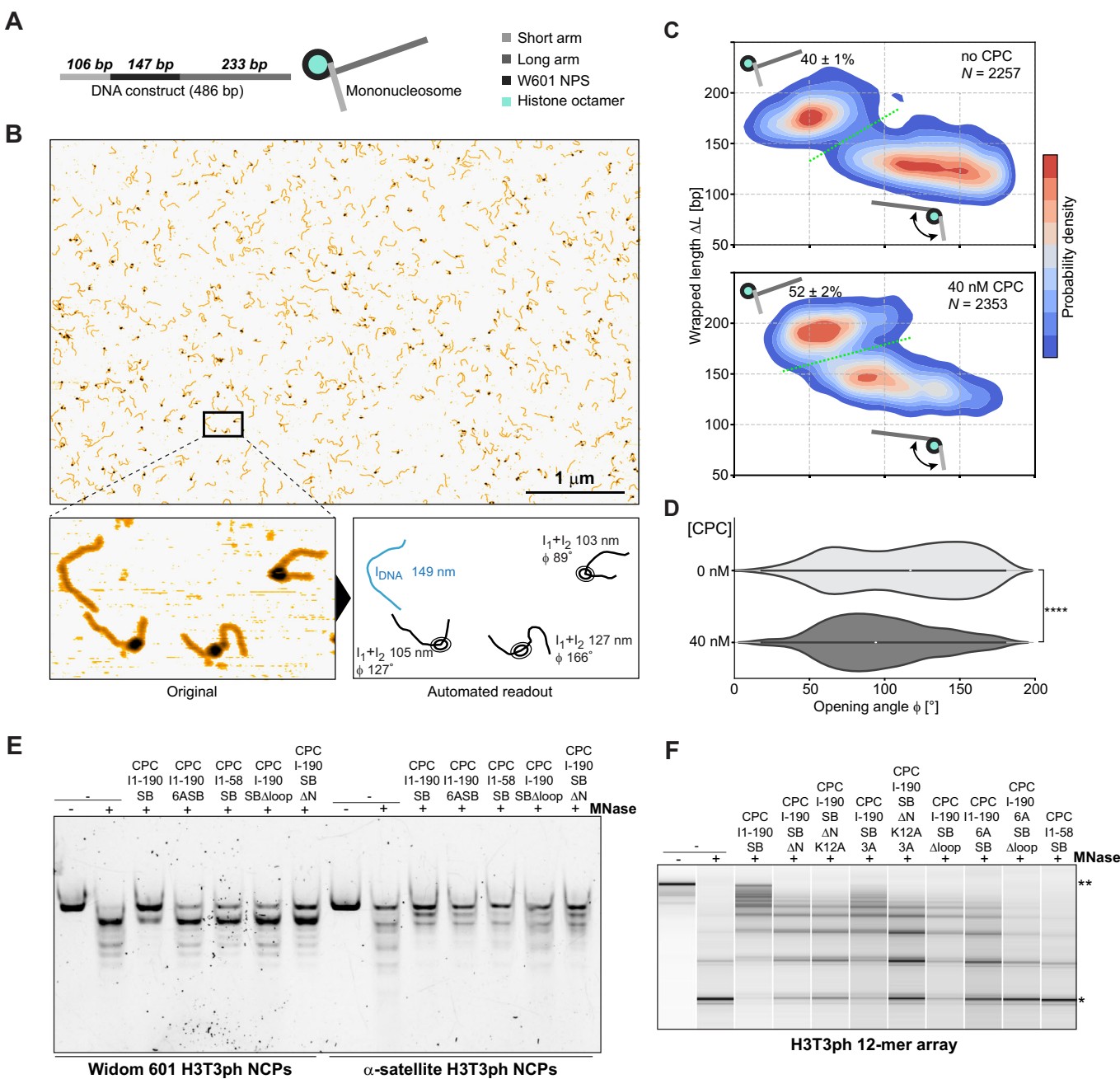

Figure 5. CPC stabilizes highly wrapped states of H3T3ph nucleosomes.

(A) Schematic depiction of the DNA construct used for nucleosome reconstitution for high-throughput AFM studies. (B) Representative overview image of H3T3ph nucleosomes incubated with 40 nM CPC$_{I1-190SB}$. Digital zoom of the boxed region is shown below, depicting three mono-nucleosomes and a bare DNA molecule, before and after automated read-out. (C) Two-dimensional kernel density estimates of wrapped length $\Delta L = < L_{DNA} > - l_1 - l_2$ versus opening angle of H3T3ph nucleosomes in the absence (top) and presence of 40 nM CPC$_{I1-190SB}$. The green dotted line indicates the cut-off separating the fully wrapped and partially unwrapped nucleosomes. (D) Violin plots of nucleosome opening angles in the absence and presence of CPC$_{I1-190SB}$. A two-sided Kolmogorov-Smirnov statistical test demonstrates a significant difference ($N_{nucleosomes} = 2257$ and $N_{nucleosomes} = 2353$ from two biological repeats in the absence and presence of CPC$_{I1-190SB}$, respectively; ****$P \leq 0.0001$; exact $P$ value $= 5 \times 10^{-49}$). (E) Native gel image of the MNase assay of H3T3ph mono-nucleosomes with Widom 601 147 bp DNA and α-satellite DNA in the presence of five different CPC constructs :CPC$_{I1-190SB}$, CPC$_{I1-190\ 6ASB}$, CPC$_{I1-58SB}$, CPC$_{1-190SBΔloop}$, and CPC$_{1-190SBΔN}$. The samples treated with MNase are indicated with "+". Corresponding quantification is shown in Fig. EV8C. (F) Representative bioanalyzer gel image of the MNase assay with H3T3ph 12-mer nucleosomal arrays in the presence of different CPC constructs: CPC$_{I1-190SB}$, CPC$_{I1-190SBΔN}$, CPC$_{I1-190SBΔN\ K12A}$, CPC$_{I1-190SB3A}$, CPC$_{I1-190SBΔN\ K12A\ 3A}$, CPC$_{1-190SBΔloop}$, CPC$_{I1-190\ 6ASB}$, CPC$_{I1-190\ 6ASBΔloop}$, and CPC$_{I1-58SB}$. The double asterisk indicates the 12-mer band, and the single asterisk indicates the monomer band. The samples treated with MNase are indicated with "+". Corresponding agarose gel and its quantification are shown in Fig. EV8D. Source data are available online for this figure.

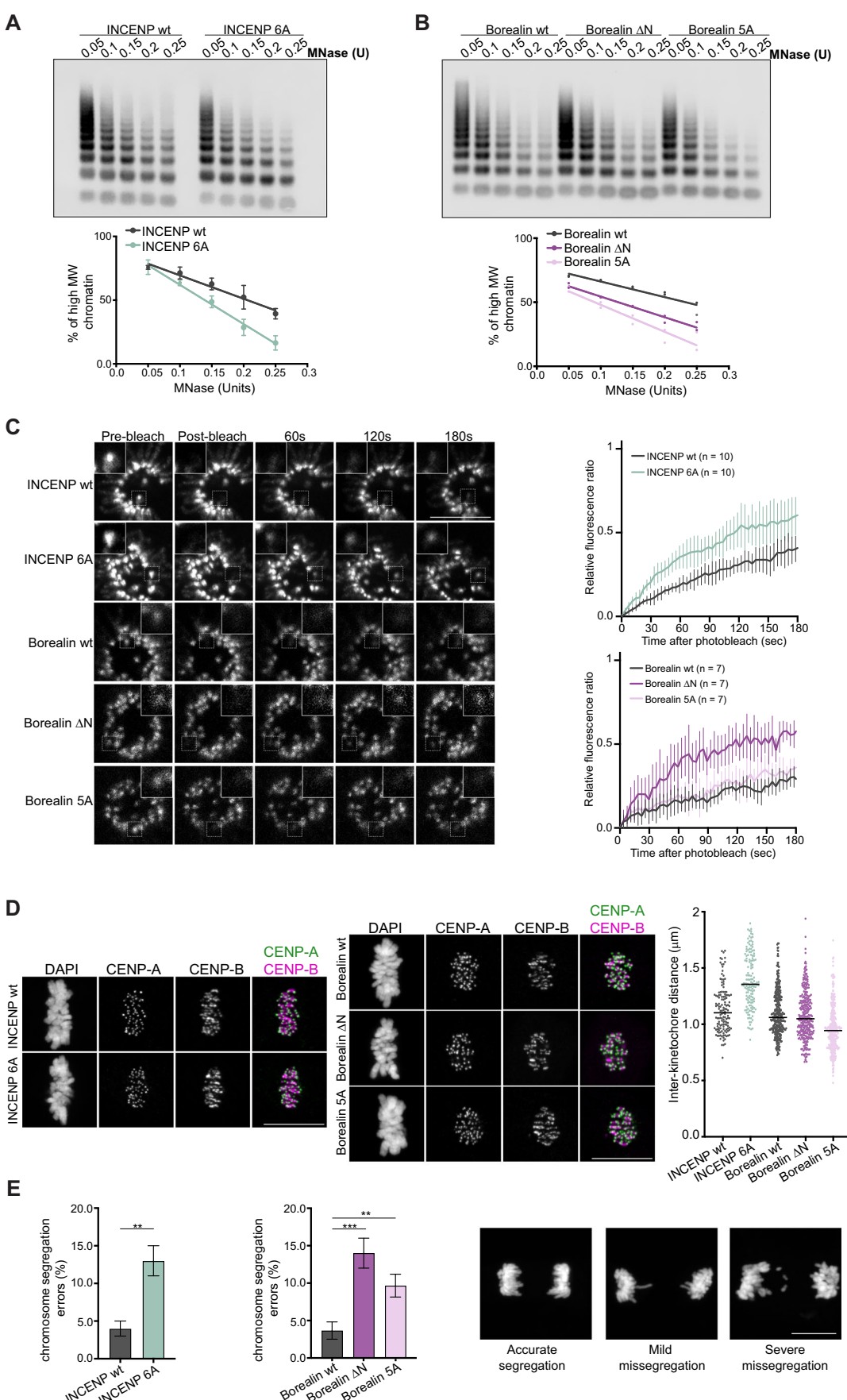

◀ **Figure 6. CPC-mediated protection of centromeric chromatin is crucial for accurate chromosome segregation.**

(A) Southern blot and corresponding quantification of MNase-digested chromatin from INCENP wt (black) and 6A (green) cell lines with an alpha-satellite probe to blot for centromeric chromatin. The MNase units (U) from left to right are 0.05, 0.1, 0.15, 0.2, and 0.25. Data are representative of three biological replicates (mean ± SD). The percentage of high molecular weight chromatin was quantified as described in Gilbert et al (2007). (B) Southern blot and corresponding quantification of MNase-digested chromatin from Borealin wt (black), ΔN (purple), and 5A (pink) cell lines using the same alpha-satellite probe to blot for centromeric chromatin. The MNase units (U) from left to right are 0.05, 0.1, 0.15, 0.2, and 0.25. Data are representative of two biological replicates. Individual data points plotted. (C) (Left panel) Representative images for the FRAP analysis of INCENP wt and 6A, and Borealin wt, ΔN, and 5A cell lines during prometaphase. The boxed regions were photobleached, and fluorescence recovery was monitored at the indicated time points. Scale bar, 10 μm. (Right panel) Quantification of FRAP experiments. Solid lines represent the mean fluorescence recovery curves obtained from individual prometaphase cells pooled from two independent biological replicates (INCENP wt—10 cells, black; INCENP 6A—10 cells, green; Borealin wt—7 cells, black; Borealin ΔN—7 cells, purple; and Borealin 5A—7 cells, pink). Vertical light lines indicate the standard deviation calculated on individual cells at each time point. (D) (Left panel) Representative images showing inter-kinetochore distance in INCENP wt and 6A cell lines. Scale bar, 10 μm. (Center panel) Representative images showing inter-kinetochore distance in Borealin wt, ΔN, and 5A cell lines. Scale bar, 10 μm. (Right panel) Quantification of the distance between CENP-A signals in at least 10 cells for INCENP wt (black) and 6A (green), and in at least 20 cells for Borealin wt (black), ΔN (purple), and 5A (pink). Data are representative of two biological replicates. Horizontal black lines indicate median values. (E) (Left panel) Quantification of chromosome segregation errors during anaphase in at least 300 cells for INCENP wt (black) and 6A (green). Data are representative of three biological replicates (mean ± SD, unpaired $t$ test with Welch's correction; $**P \leq 0.01$; $P = 0.0065$). (Center panel) Quantification of chromosome segregation errors during anaphase in at least 20 cells for Borealin wt (black), ΔN (purple), and 5A (pink). Data are representative of three biological replicates (mean ± SD, one-way ANOVA with Dunnett's multiple comparison test; $**P \leq 0.01$; $***P \leq 0.001$; exact $P$ values: $P = 0.0004$ and $P = 0.0066$, respectively). (Right panel) Representative images of cells exhibiting chromosome segregation errors, as classified in the analysis. Scale bar, 10 μm. Source data are available online for this figure.

# Methods

### Reagents and tools table

| Reagent/resource | Reference or source | Identifier or catalog number |
|---|---|---|
| **Experimental models** | | |
| INCENP-mAID (RPE1) | Hirota Lab | |
| 3xFLAG-INCENP Wt_Borealin-mNeonGreen (RPE1) | Hirota Lab | |
| 3xFLAG-INCENP 6A_Borealin-mNeonGreen (RPE1) | Hirota Lab | |
| Borealin-mAID-mClover (RPE1) | Hirota Lab | |
| Borealin-mCherry Wt (RPE1) | Hirota Lab | |
| Borealin-mCherry ΔN (RPE1) | Hirota Lab | |
| Borealin-mCherry 5A (RPE1) | Hirota Lab | |
| **Recombinant DNA** | | |
| **Antibodies** | | |
| Mouse anti-Aurora B (WB 1:500) | BD Biosciences | Cat. 611083 |
| Rabbit anti-INCENP (WB 1:5000) | Proteintech | Cat. 29419-1-AP |
| Mouse anti-Borealin (WB 1:1000) | MBL | Cat. M147-3 |
| Mouse anti-Tubulin (WB 1:5000) | Millipore | Cat. T6074 |
| Rabbit anti-mCherry (WB 1:1000) | Gene Tex | Cat. GTX128508 |
| Mouse anti-FLAG (WB 1:5000, IF:10000) | Sigma-Aldrich | Cat. F1804 |
| Mouse anti-CENP-A (IF 1:500) | MBL | Cat. 3-19 |
| Rabbit anti-CENP-B (IF1:1000) | Active motif | Cat. 61287 |
| **Oligonucleotides and other sequence-based reagents** | | |
| Primers for alpha-satellite | Chan et al, 2012 | |
| **Chemicals, enzymes, and other reagents** | | |
| Micrococcal Nuclease | Sigma-Aldrich | N3755-50UN |
| Micrococcal Nuclease | NEB | NEB M0247S |
| Proteinase K | Fujifilm | 161-28701 |
| RNase A | Invitrogen | 12091021 |

| Reagent/resource | Reference or source | Identifier or catalog number |
|---|---|---|
| Monastrol | TOCRIS | 1305 |
| Nocodazole | Selleck | S2775 |
| Palbociclib | Selleck | S1116 |
| 5-Ph-IAA | MedChemExpress | HY-141894 |
| Puromycin | ThermoFisher | A1113802 |
| G418 | ThermoFisher | 11811031 |
| Zeocin | Invitrogen | R25001 |
| SiR-DNA | Spirochrome | Cat. CY-SC007 |
| **Software** | | |
| Fiji (2.1.0/1.53c) | | |
| cryoSPARC v4.7.0 | | |
| RELION 5.0 | | |
| ChimeraX 1.9 | | |
| PyMol 3.1 | | |
| Illustrator | | |
| Prism 10 | | |
| Adobe Photoshop 2025 | | |
| **Other** | | |

## Protein expression and purification of CPC constructs

The vectors used to express and purify CPC were: pMCNcs-INCENP$_{1-58}$ for untagged INCENP$_{1-58}$, pEC-S-CDF-His INCENP for His-tagged INCENP$_{1-190}$ and His-tagged INCENP$_{1-190\ 6A}$, pRSET-GFP-Survivin for GFP-tagged full length Survivin and pETM-K-TEV-His-Borealin (pETM vector, gift from C. Romier, Institute of Genetics and Molecular and Cell Biology, Strasbourg, France) for His-tagged Borealin full length, Borealin$_{\Delta N}$, Borealin$_{\Delta N\ K12A}$, Borealin$_{3A}$, Borealin$_{\Delta N\ K12A\ 3A}$, Borealin$_{5A}$, Borealin$_{\Delta N\ K12A\ 5A}$, and Borealin$_{\Delta loop}$. The INCENP and Borealin deletions and mutations were generated using the Quikchange site-directed mutagenesis method (Stratagene).

CPC constructs were purified as described in Abad et al (2019, 2022). Briefly, the three vectors for INCENP, Survivin, and Borealin constructs were co-transformed in BL21(DE3) pLysS cells. The bacterial cultures were then grown at 37 °C until an OD of 0.6-0.8 was reached. Cultures were then induced overnight at 18 °C with 0.35 mM IPTG. The cultures were then harvested by spinning at $4000 \times g$ at 4 °C for 15 min, and the pellets were resuspended in lysis buffer containing 20 mM Tris pH 8.0, 500 mM NaCl, 35 mM imidazole, and 2 mM β-Mercaptoethanol, supplemented with complete EDTA-free cocktail tablets (Roche), 0.01 mg/ml DNase (Sigma-Aldrich), and 1 mM PMSF. The lysate was then sonicated at 40% output for 3 rounds of 5 min (pulse: 2 on and 2 off), and clarified post sonication at 22,000 rpm for 50 min. The supernatant was then loaded into a pre-equilibrated 5 ml His Trap Column (Cytiva) for affinity chromatography. The protein-bound column was washed with lysis buffer followed by chaperone buffer (20 mM Tris pH 8.0, 1000 mM NaCl, 35 mM Imidazole, 50 mM KCl, 10 mM $MgCl_2$, 2 mM ATP, and 2 mM β-Mercaptoethanol). The protein was then eluted using elution buffer (20 mM Tris pH 8.0, 500 mM NaCl, 500 mM Imidazole, 2 mM β-Mercaptoethanol) and dialyzed overnight in dialysis buffer (25 mM Hepes pH 8.0, 250 mM NaCl, 2 mM DTT) while cleaving with TEV and 3C proteases for affinity tag cleavage. The dialyzed samples were spun down, and the supernatant was loaded onto a HiTrap SP HP (Cytiva) cation exchange column to separate the $CPC_{ISB}$ stoichiometric complexes from other sub-complexes and tags. The fractions containing stoichiometric and pure $CPC_{ISB}$ complex were pooled, concentrated, and run on a Superdex 200 Increase 10/300 (Cytiva) pre-equilibrated with 25 mM Hepes pH 8.0, 250 mM NaCl, 5% Glycerol, and 1 mM DTT. Protein fractions were pooled, concentrated, and snap frozen for storage at −80 °C.

## Protein expression and purification of Histones

Histones were purified based on the protocol from Luger et al (1999). For H2A, H2B, and H3 histones, the vectors were transformed into BL21 pLysS, while H4 histone vector was transformed into BL21 Gold. The bacterial cultures were then grown at 37 °C until an $OD_{600}$ of 0.5–0.7 was reached. They were then induced with 0.2 mM IPTG for 3 h. The culture was harvested by spinning at $5000 \times g$ at RT for 10 min, and the pellet was resuspended in ice-cold wash buffer containing 50 mM Tris pH 7.5, 100 mM NaCl, 1 mM EDTA, 1 mM Benzamidine, and 1 mM β-Mercaptoethanol. The resuspended cultures were snap frozen and thawed three times to lyse the cells. The lysates were then sonicated on ice with 60% output for a total of 7.5 min. The lysate was then clarified by spinning at 22,000 rpm for 60 min at 4 °C. After centrifugation, the pellet was homogenized using a Dounce glass/glass homogenizer in wash buffer with and without 1% triton X-100. The pellet was then soaked in DMSO for 15 min and then homogenized once more in unfolding buffer containing 20 mM Tris pH 7.5, 7000 mM GuHCl, and 10 mM DTT. One more round of sonication was done for 15 s at 40% output, and the lysate was stirred at RT for an hour, and then centrifuged at $23,000 \times g$ for 20 min at 4 °C. The supernatant was then dialyzed in Urea Buffer containing 10 mM Tris pH 8, 100 mM NaCl, 7000 mM Urea, 1 mM EDTA, and 5 mM β-Mercaptoethanol overnight in three rounds. After dialysis, the samples were centrifuged at $45,000 \times g$ for 30 min at 4 °C. The samples were once again sonicated for 10 s at 40% output, and filtered through a Millipore combined glass/PVDF

filter (SLHVM25NS) to clear any remnant DNA contamination. The sample was then loaded onto a HiTrap Q column and HiTrap SP HP (Cytiva) cation exchange column sequentially to get pure histone fractions. Histones were eluted from the HiTrap SP column, and the fractions containing histones were pooled, dialyzed against water supplemented with acetic acid, and lyophilized for storage in −80 °C. The tail-less histone H3 used for native ligations (H3T32C) was purified using the same procedure with the following changes: pH 7 was used for all the buffers, and the Urea dialysis buffer contained 10 mM Tris pH 8, 50 mM NaCl, 7000 mM Urea, 1 mM EDTA, and 5 mM β-Mercaptoethanol.

## Native ligation of H3T3ph histone tail

The native ligation protocol is based on (Bartke et al, 2010; Cistrone et al, 2019) and subsequently modified based on advice from Philipp Voigt (Babraham Institute). The modified histone tail was ordered from Peptide Synthetics Ltd. Initially, the lyophilized H3 tail-less histone (H3T32C) was dissolved in methoxyamine buffer containing 6000 mM GuHCl, 400 mM methoxyamine, and 20 mM TCEP with a pH of around 3–4 to a concentration of around 32 mg/ml. The dissolved histone was then incubated at RT overnight, after which the methoxyamine was removed by dialyzing for 2 h in ligation buffer (6 M GuHCl, 200 mM $Na_2HPO_4$ pH 6.5–7, 20 mM TCEP). The modified histone tail peptide was dissolved in ligation buffer supplemented with 400 mM MPAA, and the pH adjusted to 4–5. The peptide solution was then mixed with the histone solution, and after adjusting the pH to around 7, the mixture was incubated at 23 °C at RT for 2–3 days. Following the ligation, the mixture was dialyzed against Urea buffer (10 mM Tris pH 8, 100 mM NaCl, 7000 mM Urea, 1 mM EDTA, and 1 mM DTT) overnight. The Resource S column (Cytiva) was used to separate the ligated histones from the unligated ones and the excess peptide. Fractions with ligated histones were then pooled, dialyzed overnight against water, and lyophilized for storage.

## Nucleosome preparation

Lyophilized histones were resuspended in unfolding buffer, 7000 mM Guanidine HCl, 20 mM Tris pH 7.5, 10 mM DTT, and spun down at $14,000 \times g$ for 5 min. After checking the concentration using Bradford, H3:H4:H2A:H2B were mixed in the ratio 1:1:1.1:1.1 and run on an SDS-PAGE gel to confirm the stoichiometry. The histone mixture was then dialyzed for 2 h in refolding buffer, 10 mM Tris pH 8, 2000 mM NaCl, 1 mM EDTA, 5 mM β-Mercaptoethanol, using an 8 kDa molecular weight cut-off dialysis membrane. After 2 h, the refolding buffer was replaced with fresh buffer, and the mixture was dialyzed overnight. A final dialysis was done for 2 h with fresh refolding buffer, and the octamer was then spun down, filtered, and run on a Superdex 200 Increase 10/300 (Cytiva) pre-equilibrated with refolding buffer. The stoichiometric fractions were pooled, snap frozen, and stored at −80 °C.

For Widom DNA, after PCR amplification of the Widom 147 bp sequence from a pBS-601 Widom vector, Resource Q was carried out to purify the DNA. The Widom 601 147 bp sequence used was: ACAGGATGTATATATGTGACACGTGCCTGGAGACTAGGGA GTAAT  CCCCTTGGCGGTTAAAACGCGGGGGACAGCGCGT ACGTGCGTTTAAGCGGTGCTAGAGCTGTCTACGACCAATT-GAGCGGCCTCGGCACCGGGATTCTCCAG. For the α-satellite

DNA, 10 midi preps were performed to purify the pUC57 containing 6 × 147 bp α-satellite DNA (kindly gifted by Prof. Ben Black; Falk et al, 2015), followed by an EcoRV digestion before loading the sample into a Resource Q column to obtain clean α-satellite DNA. The histone octamers were then titrated with the widom/α-satellite DNA by dialyzing the mixture in high salt refolding buffer (10 mM Tris pH 8.0, 2000 mM NaCl, 1 mM EDTA, and 5 mM β-Mercaptoethanol). The salt concentration was slowly brought down by adding low salt TE buffer (10 mM Tris pH 8.0, 50 mM NaCl, and 1 mM EDTA) to facilitate the wrapping of the widom/α-satellite DNA around the octamer as described in Luger et al (1999). The titration was initially done on a small scale to determine the optimal ratio of Octamer:DNA for nucleosome reconstitution. After determining the correct ratio, large-scale dialysis was done to obtain the nucleosomes used in our assays.

## Preparation of nucleosome arrays

The DNA template containing 12 tandem repeats of 167-bp Widom 601 sequence was kindly provided by Dr Dongyan Tan (originally from Dr. Craig Peterson). Large amounts of plasmid were prepared using a Macherey-Nagel PC 10000 kit (740593), following the manufacturer's instructions. The desired 12-mer positioning sequence was cut out by an AvaI (NEB, R0152S) and HindIII (NEB, R3104S) restriction digest. The rest of the vector was fragmented using restriction enzyme DraI (NEB, R0129L). The 12-mer nucleosomal array DNA was isolated by size-exclusion chromatography using a Sephacryl-S500 16/60 column (GE Healthcare). Reconstitution into nucleosome arrays was performed by mixing array DNA at a final concentration of 0.3 μg/μl with 3 times molar excess of H3T3ph histone octamer (in proportion to available binding sites) and 0.1 mg/ml BSA (NEB, B9200S) in the high-salt buffer HEN2000 (10 mM HEPES, pH 8; 0.1 mM EDTA; 2 M NaCl). For array assembly, a tenfold volume of buffer HE (10 mM HEPES, pH 8; 0.1 mM EDTA) was added over 24 h at 4 °C. Final assembly took place by dialyzing against fresh low-salt buffer HN50 (10 mM HEPES, pH 8; 50 mM NaCl) for at least 3 h. Arrays were then precipitated by 5 mM Mg for 15 min at RT, spun down, and the pellet was resuspended in 10 mM HEPES, pH 8; 5 mM KCl, yielding a concentrated, high-quality array.

## CPC-NCP complex formation using SEC

SEC experiments with recombinant purified proteins were performed using a MicroAKTA system (Cytiva), with Superose 6 5/150 (Cytiva) pre-equilibrated in 20 mM Hepes pH 8, 250 mM NaCl, 1 mM EDTA, and 1 mM DTT. A 5 times molar excess of CPC$_{ISB}$ was incubated with H3T3ph mono-nucleosomes and 15 mM ATP for 30 min on ice before injection. UV 280 and 260 nm wavelengths were monitored to study complex formation, and 0.05 ml fractions were collected and run on SDS-PAGE gels.

## Mass photometry

Coverslips were cleaned with isopropanol, sonicated, and air-dried prior to use. Silicone gaskets were positioned at the center of each coverslip before mounting onto a Refeyn TwoMP mass photometer. CPC-NCP complexes were diluted to 20 nM in measurement buffer (20 mM HEPES, pH 8.0; 50 mM NaCl; 1 mM EDTA; 1 mM DTT). Following buffer blanking, samples were introduced into the gasket wells, and measurements were recorded. Data acquisition was performed using a TwoMP mass photometer (Refeyn) operated with AcquireMP software (v2023 R1.1), and the analysis was carried out using DiscoverMP (Refeyn, v2023 R2).

## Cryo-EM sample preparation and data collection

CPC$_{I1-190SB}$-H3T3ph mono-nucleosome complex was prepared by SEC as mentioned above. Fractions containing the stoichiometric complex were pooled and diluted to a final DNA concentration of 115 ng/μl right before grid preparation. In all, 4.5 μl of sample was applied onto Quantifoil R2/2 300-mesh grids previously glow-discharged for 7 s in the presence of air using a current of 20 mA. Grids were blotted for 1.5 s at 10 °C and 95% humidity using a Leica EM GP plunger (Leica). 22,065 micrographs were collected using a Titan Krios transmission electron microscope operating at 300 keV equipped with a Falcon 4i direct electron detector and a Selectris X Energy Filter (energy slit width of 5 eV) (ThermoFisher). Smart EPU software (ThermoFisher) was used for the automated Data collection, MotionCor2 was used to perform motion correction of the movies (Zheng et al, 2017). Two datasets were collected using the grid. The parameters used for data collection are provided in Table 1.

Motion-corrected micrographs were processed using CryoS-PARC v4.7.0 (Punjani et al, 2017). AI-based picking tool Topaz (Bepler et al, 2019, 2020) was initially used for particle picking with a previously trained nucleosome model, and the particles were extracted with a box size of 384 px, after which multiple rounds of 2D classification were performed to filter out junk particles and select the 2D classes presenting high resolution features. 2.3 million particles (from both datasets combined) were used to generate several ab initio models and subsequent heterogeneous and homogeneous refinements, from which maps showcasing nucleosomes with well-defined bound CPC (1.2 million particles) were selected. Further focused 3D classification (with a mask for the CPC density) followed by non-uniform refinement (Punjani et al, 2020) yielded three maps displaying three different CPC orientations at a resolution of 2.83 Å (Class 2; PDB 9SI3, EMD-54924), 2.85 Å (Class 1; PDB 9SJ5, EMD-54938), and 2.86 Å (Class 0; PDB 9SI9, EMD-54926) (Fourier shell correlation (FSC) = 0.143). 3D variability was performed by combining the particles corresponding to the three final maps to study the dynamicity of CPC binding to nucleosomes (Movie EV1). For each of the three final 3D volumes, AI-based DeepEMnhancer sharpening was performed (Sanchez-Garcia et al, 2021). Particles of Class 2 were taken to RELION 5.0 to perform 3D Refine (with blush regularization) (Kimanius et al, 2024) with and without CTF correction, which enabled resolution of CPC densities on both sides (PDB 9SLJ, EMD-55003). Parallel processing aimed at resolving CPC densities on both sides of the nucleosome was performed using cryoSPARC by removing the nucleosome density using particle subtraction and subsequent 3D classifications. A 3 Å map with clear CPC densities on both faces was obtained (EMD 55012). A crystal structure of the CPC triple helical bundle (PDB: 2QFA) and of human NCP (PDB: 2CV5) were initially fitted into the cryo-EM map using the rigid body fitting built-in tool in ChimeraX (version 1.9). Model building was done by performing

iterative rounds of real-space refinement in Phenix and Coot. Interactions were analyzed using PDBePISA (https://www.ebi.ac.uk/pdbe/pisa/). Schematics of the cryo-EM processing workflow are included in Figs. EV2 and EV3.

## AFM sample preparation and imaging

Mono-nucleosomes for AFM was prepared by wrapping H3T3ph histone octamers with a 486 bp DNA construct comprising the Widom 601 nucleosome positioning sequence flanked by a short (106 bp) and long (233 bp) extra-nucleosomal DNA arm: caat-caaactggctcgtcgcgaattggagctccaccgcggtggcggccgctcgatctagtactagtgg-catgtcagctgcaggaattcgagctcaacgtgcaatccctggagaatcccggtgccgaggccgc tcaattggtcgtagacagctctagcaccgcttaaacgcacgtacgcgctgtcccccgcgttt-taaccgccaaggggattactccctagtctccaggcacgtgtcagatatatacatcctgtcgacga-cacgggtgatcgactagttctagagcgatctagtatcgatcactcttttgttccctttagtgagggt-taatttcgagcttgcgatcgtagtcgatactacgcgtaatcatggtcatagctgtttcctgtgt-gaaatgatctacttgttatccgctcacaattccacacaacatacgagccggaaaagaaataaa-gatctagactactgcataaagtgtaaagcctggg (Genart strings DNA fragment, Geneart, Thermo Fisher Scientific). The final nucleosome sample contained approximately 50% free DNA, as was desirable for the experiment.

We prepared our samples by diluting nucleosomes and $CPC_{I1-190SB}$ to intermediate concentrations that are twofold concentrated with respect to the final concentration, and incubated these on ice for 10 min. Then, equal volumes of the nucleosomes and CPC were combined and mixed by gentle stirring. This sample was left on ice for 10 min prior to sample deposition on poly-L-lysine-coated mica.

Poly-L-lysine-coated mica was prepared by drop-casting 20 µl poly-L-lysine (Merck; 0.01% w/v in autoclaved milliQ water) on freshly cleaved muscovite mica (SPI Supplies) for 30 s and subsequently rinsing the surface with 20 ml of milliQ water before drying with a gentle stream of filtered $N_2$ gas (Vanderlinden et al, 2014). Mononucleosome samples (at 0.75 ng/µl of DNA) were incubated in aqueous buffer comprising 200 mM NaCl, 20 mM HEPES-KOH (pH 7.4), and 1 mM DTT for 10 min on ice. The sample was then deposited on the poly-L-lysine-coated muscovite mica for 15 s and rinsed with 20 ml milliQ water before drying with a gentle stream of filtered $N_2$ gas.

A Nanowizard 4 XP AFM (JPK, Berlin, Germany) was used in tapping mode with silicon tips (FASTSCAN-A; drive frequency, 1400 kHz; Bruker) over fields of view of 6 × 6 µm at 4096 × 4096 pixels and captured at line rates of 3 Hz. For each condition, two independent data sets were recorded. The nucleosome samples for each data set were prepared in independent nucleosome reconstitutions.

## AFM image analysis

To process the raw topographic data, we used SPIP software for plane correction that included plane-fitting with a third-degree polynomial, and line-by-line correction with a fourth-degree polynomial. The processed AFM topographs were further analyzed through a previously published, open-source automated image analysis pipeline (Konrad et al, 2021a); available at GitHub: https://github.com/SKonrad-Science/AFM_nucleoprotein_readout). In brief, image analysis starts with molecule detection and classification. To this end, a Gaussian filter and a background subtrac-tion are applied followed by skeletonization whereby molecules are trimmed to a one-pixel-wide backbone. This skeleton is used for classification: bare DNA has exactly two endpoints in its skeleton and no branchpoints, while mono-nucleosomes have exactly two endpoints and two branchpoints. In a second analysis step, the tip shape and size is estimated from the skeleton of a subset of bare DNA molecules. Using the estimated tip shape, image erosion is applied to minimize tip dilation effects. Last, the classified molecules are analyzed with respect to the structure parameters arm lengths, volume, and opening angle for nucleosomes and contour length for bare DNA. The structure parameters are exported as text files and used for plotting and statistical analysis in Python. To minimize the effect of DNA fragments on calculation of wrapped lengths in nucleosomes (see Fig. EV7), we employ the mode instead of the mean of the bare DNA contour length distribution. To quantify the fraction of fully wrapped nucleosomes in an ensemble, we use the 2D heatmap of wrapped length versus opening angles (Figs. 5C and EV7). In this plot, we identify the two dominant local maxima (corresponding to the most probable fully wrapped respectively partially unwrapped conformations). Along the line connecting these maxima the valley point was found. Then, the minimum was detected inside the rectangle defined by the local maxima as opposing vertices. This minimum and valley point define the separation line (green dotted line in Fig. 5C): nucleo-somes are said to be fully wrapped if they fall above it. In our samples, a fraction of our nucleosomes was end-bound, without linker DNA. This phenomenon is well-understood for systems with long extra-nucleosomal DNA. Due to the intrinsic bending stiffness of DNA, long extra-nucleosomal DNA must cross or come in close proximity, which presents an energy penalty, due to electrostatic repulsion. When bound near the end of the DNA construct (and away from the positioning sequence), this penalty is removed as DNA does not need to cross. Our software automatically discards these end-bound nucleosomes and exclusively focuses on those nucleosomes where extra-nucleosomal DNA is automatically detected at each end of the nucleosome.

## Micrococcal nuclease assay

For mono-nucleosomes with Widom 601 and α-satellite DNA, 1 µg of nucleosome with or without $CPC_{ISB}$ were incubated on ice for 1 h. In all, 0.25 µl of MNase (NEB M0247S) was then added to the mix and made up to 50 µl with the reaction buffer (50 mM Tris-HCl pH 8, 100 mM NaCl, 3 mM $CaCl_2$, 1 mM DTT), and incubated at RT for 10 min. The reaction was then quenched using 10 µl of 500 mM EDTA and 20 µg of proteinase K was added per reaction and incubated at 50 °C for 30 min to digest the proteins. The DNA was then purified using Minelute Qiagen kit, and run on a 5.2% Tris-Glycine Native acrylamide gel at 100 V for 1.5 h. The gel was stained with SybrSafe before imaging.

For nucleosomal arrays, 500 ng of array with or without CPC was digested with 1 µl of 1/8th dilution of MNase (NEB M0247S) in the reaction buffer. The reaction was then quenched using 10 µl of 500 mM EDTA and 20 µg of proteinase K was added per reaction and incubated at 50 °C for 30 min to digest the proteins. The DNA was then purified using Minelute Qiagen kit, and run on an Agilent 2100 Bioanalyzer using a High Sensitivity DNA assay kit. The isolated DNA fragments were also separated using 2% agarose gel.

## EMSA

In all, 20 nM mono-nucleosomes were incubated for an hour with increasing amounts of CPC constructs (using a two-fold serial dilution, ranging from 7.8 nM to 2 μM) on ice in EMSA reaction buffer containing 10 mM Tris pH 7.5, 250 mM NaCl, 1 mM MgCl2, 1 mM DTT and 1% Glycerol supplemented with 0.1 mg/ml BSA. The complex was then run on a 5.2% polyacrylamide Tris-Glycine gel in 1× Tris-Glycine buffer at 100 V for 1 h 30 min at 4 °C. Unbound nucleosome bands were analyzed using ImageJ. Data was plotted in GraphPad Prism 10.0 with a log10 x-axis and an isotherm fitted using the "specific binding curve with Hill slope" fitting. The equation for this fitting is: $Y = \text{Bmax} * X^h / (K_d^h + X^h)$, where $Y$ is the specific binding, $X$ is the protein concentration, Bmax is the maximum fraction bound, $K_d$ is the dissociation constant, and $h$ is the Hill slope.

## SPR

SPR measurements were conducted using a BIAcore T200 instrument (GE Healthcare). Streptavidin sensor chips (also from GE Healthcare) were used for ligand immobilization. Prior to immobilization, the sensor surfaces were primed with three consecutive 30 s injections of 1 M NaCl and 50 mM NaOH at a flow rate of 30 μl/min, followed by thorough washing with running buffer (25 mM HEPES, 250 mM NaCl, 1 mM DTT, and 0.05% Tween-20, pH 8.0). Biotinylated nucleosome core particles (NCPs) were immobilized on designated flow cells by injecting a 20 nM solution in running buffer at 5 μl/min, adjusting contact time to achieve ~500 RU of immobilization.

Prior to each SPR experiment, CPC complexes were dialyzed against running buffer for 1 h at 4 °C. Binding titrations were performed at 8 °C using a twofold dilution series of the analytes (ranging from 7.8 to 1000 nM), injected over the sensor surface at 100 μl/min with an association time of 90 s and a dissociation time of 150 s. To regenerate the sensor surface between analyte injections, running buffer was flowed for 300 s at 100 μl/min to remove any residual complexes.

A reference flow cell containing a streptavidin surface without immobilized ligand was used for background subtraction. Due to the complex and multiphasic binding behavior observed between CPC constructs and nucleosomes in some cases, kinetic models were not applied during data fitting. Instead, the majority of interactions were well described by a simple steady-state binding model. Equilibrium dissociation constants ($K_d$) were calculated by globally fitting the sensorgrams to a steady-state 1:1 binding model with mass transport limitations, using the Biacore T200 Evaluation Software (v2.02). For visualization, data were replotted using GraphPad Prism 10.0.

## Crosslinking/MS

Crosslinking experiments for the CPC$_{I1\text{-}190SB}$-H3T3ph NCP complex were performed using the 1-ethyl-3-(3-dimethylaminopropyl) carbodiimide (EDC) chemical crosslinker in the presence of N-hydroxysulfosuccinimide (Thermo Fisher Scientific). EDC is a zero-length chemical crosslinker that covalently links primary amines of lysine and the protein N-terminus, and to a lesser extent hydroxyl groups of serine, threonine and tyrosine, with carboxyl groups of aspartate or glutamate. 3 μg of NCP was crosslinked with 2.5 molar excess of

CPC$_{I1\text{-}190SB}$ using either 30 or 60 μg of EDC and either 66 or 132 μg of N-hydroxysulfosuccinimide in 20 mM Hepes, pH 8, 200 mM NaCl, 1 mM EDTA, and 2 mM DTT for 2 h at room temperature (RT). The cross-linking reaction was quenched by the addition of 100 mM Tris-HCl, and crosslinking products were briefly resolved using 4–12% Bis-Tris NuPAGE (Thermo Fisher Scientific). Bands were visualized by short Instant Blue staining (Thermo Fisher), excised, reduced with 10 mM DTT for 30 min at RT, alkylated with 5 mM iodoacetamide for 20 min at RT, and digested overnight at 37 °C using 13 ng/μl trypsin (Promega). Digested peptides were loaded onto C18-Stage-tips (Rappsilber et al, 2007).

Liquid chromatography with tandem mass spectrometry (LC-MS/MS) analysis was performed as previously described (Abad et al, 2022) using an Orbitrap Fusion Lumos Tribrid Mass Spectrometer (Thermo Fisher Scientific) applying a "high-high" acquisition strategy. Peptide mixtures were injected for each mass spectrometric acquisition and separated on a 75 μm × 50 cm PepMap EASY-Spray column (Thermo Fisher Scientific) fitted into an EASY-Spray source (Thermo Fisher Scientific) and run at a column temperature of 50 °C. Mobile phase A consisted of water and 0.1% vol/vol formic acid, while mobile phase B consisted of 80% vol/vol acetonitrile and 0.1% vol/vol formic acid. Peptides were loaded at a flow-rate of 0.3 μl/min and eluted at 0.2 μl/min using a linear gradient going from 2% mobile phase B to 40% mobile phase B over 159 (or 129) min, followed by a linear increase from 40 to 95% mobile phase B in 11 min. The eluted peptides were then loaded into the mass spectrometer, and MS data was acquired in the data-dependent mode with the top-speed option. For each 3 s acquisition cycle, the mass spectrum was recorded in the Orbitrap with a resolution of 120,000. The ions with a precursor charge state between 3+ and 8+ were isolated and fragmented using higher-energy collisional dissociation (HCD) or electron-transfer/HCD (EThcD). The fragmentation spectra were recorded in the Orbitrap. Dynamic exclusion was enabled with single repeat count and 60 s exclusion duration.

The mass spectrometric raw files were processed into peak lists using ProteoWizard (v3.0.20338; Kessner et al, 2008), and cross-linked peptides were matched to spectra using Xi software (v1.7.6.3; Mendes et al, 2019); https://github.com/Rappsilber-Laboratory/XiSearch) with in-search assignment of monoisotopic peaks (Lenz et al, 2018). The search parameters used were MS accuracy, 3 ppm; MS/MS accuracy, 10 ppm; enzyme, trypsin; cross-linker, EDC; max missed cleavages, 4; missing mono-isotopic peaks, 2; fixed modification, carbamidomethylation on cysteine; variable modifications, oxidation on methionine; and fragments b- and y-type ions (HCD) or b-, c-, y-, and z-type ions (EThcD) with loss of $H_2O$, $NH_3$, and $CH_3SOH$. 1% on link level false discovery rate (FDR) was estimated based on the number of decoy identification using XiFDR (Fischer and Rappsilber, 2017).

## UV crosslinking/MS

UV crosslinking uses ultraviolet light (UV 254 nm) to form zero-length covalent links between nucleic acids and protein. Thirty μg of H3T3ph NCPs reconstituted with Widom 601 147 bp DNA were incubated with 8 times molar excess of CPC$_{I1\text{-}190SB}$ for 1 h at 4 °C and UV crosslinked, without the use of any chemical crosslinker, using a Stratalinker UV 1800 crosslinker (Stratagene) with a 254 nM UV light bulb and a dose of 1 J/cm². Samples were then loaded onto a 4–12% Bis-Tris NuPAGE gel and subjected to MS

analysis. In-gel digestion was performed as described before (Abad et al, 2019). Following digestion, 10% of each sample was spun onto the stage tips as described above, and the rest 90% of each sample was used to enrich for phospho-groups using Zr-IMAC beads (Resyn Biosciences).

LC-MS/MS analyses were performed on Orbitrap Fusion™ Lumos™ Tribrid™ Mass Spectrometer (Thermo Fisher Scientific, UK) on a Data Independent Acquisition (DIA) mode, coupled on-line, to an Ultimate 3000 HPLC (Dionex, Thermo Fisher Scientific, UK). Peptides were separated on a 50 cm (2 µm particle size) EASY-Spray column (Thermo Scientific, UK), which was assembled on an EASY-Spray source (Thermo Scientific, UK) and operated constantly at 55 °C. Mobile phase A consisted of 0.1% formic acid in LC-MS grade water and mobile phase B consisted of 80% acetonitrile and 0.1% formic acid. Peptides were loaded onto the column at a flow rate of 0.3 µl/min and eluted at a flow rate of 0.25 µl/min according to the following gradient: 2–40% mobile phase B in 150 min and then to 95% in 11 min. Mobile phase B was retained at 95% for 5 min and returned back to 2% a minute after until the end of the run (160 min).

We performed MS analysis in a Data Dependent Acquisition mode. MS1 scans were recorded at 120,000 resolution (scan range 350–1600), with an ion target set to "Standard" and maximum injection time of 50 ms. MS2 was performed in the orbitrap at 30,000 resolution and isolation window ($m/z$) of 1.6. We used HCD fragmentation with stepped collision energy of 24, 26, 28. The normalized AGC target was set to 5.0 e4 and the cycle time to 3 s. The only ions with charges 2–6 were subjected to fragmentation and the dynamic exclusion was set to 20 s.

The OpenMS NuXL App software platform (Welp et al, 2025) was used to map protein-DNA interactions against our in-house CPC protein complex. The "DNA-UV Extended" was set as a suitable preset. Carbamidomethylation of cysteine was set as fixed modification and oxidation of methionine as a variable modification. The minimum peptide length was set at 6 AA and we allowed 2 missed cleavages. For the precursor and fragment mass tolerance, we used the default values (6 and 20 ppm respectively). The peptide FDR was set to 1% and we only used crosslinked peptides with XL FDR of 0.01.

## Cell culture and cell lines

hTRET-RPE1 cells were cultured in DMEM supplemented with 10% FBS and 0.2 mM L-glutamine at 37 °C in a 5% $CO_2$ environment. To generate an RPE1 cell line stably expressing OsTIR1(F74G), RPE1 cells were first subjected to CRISPR-Cas9-mediated knockout of the originally integrated puromycin resistance gene and then transfected with pMK444, which harbors the OsTIR1(F74G) gene (Addgene #121192), and pCMV-hyBase (PiggyBac transposase) using the Neon Transfection system (Thermo Fisher Scientific) as described (Hatoyama et al, 2024). Transfected cells were selected with 1 µg/ml puromycin, and isolated as single clones. The INCENP-mAID RPE1 cell line was generated from OsTIR1 (F74G)-expressing RPE1 cells using CRISPR-Cas9-mediated homologous recombination. Briefly, approximately 500 bp homology arms corresponding to the INCENP gene were cloned into the donor vector pMK391 (Addgene #121192). A short guide RNA (sgRNA) targeting the

C-terminus of INCENP was designed using the web-based prediction tool CHOPCHOP (https://chopchop.cbu.uib.no/) and replaced the originally inserted sgRNA in the sgRNA vector (Addgene #72833). These plasmids were transfected into OsTIR1(F74G)-expressing RPE1 cells using the Neon system. Transfected cells were selected with 50 µg/mL blasticidin, isolated as single clones, and validated as biallelic knock-ins using PCR and Western blotting. For mNeonGreen-tagging or mAID-mClover-tagging of Borealin, cells were transfected with a modified pMK286 donor plasmid (in which the mAID tag was replaced with mNeonGreen) or with pMK289, together with an sgRNA plasmid targeting the C-terminus of Borealin, using the same procedure as described above.

To generate doxycycline-inducible RPE1 cells expressing wild-type or 6A mutant INCENP, 3×Flag-tagged INCENP genes were inserted into the hROSA26 targeting vector harboring the TetOn 3 G system (Addgene #114699) and transfected into the INCENP-mAID RPE1 cell line, along with the sgRNA plasmid targeting the hROSA26 locus (Addgene #105927). Similarly, doxycycline-inducible RPE1 cells expressing wild-type, ΔN, or 5A mutant Borealin-mCherry were established as described above.

For the replacement condition, cells were treated with 1 µg/mL doxycycline or 1 µM 5Ph-IAA for the indicated time, thereby inducing gene expression from the hROSA26 locus and promoting degradation of the endogenous mAID-fused protein.

## Immunofluorescence microscopy

Cells grown on coverslips were fixed with 2% paraformaldehyde in PBS for 10 min and permeabilized with 0.2% Triton X-100 in PBS for 10 min. After blocking with 3% BSA in PBS for 10 min, cells were incubated with primary antibodies diluted in 3% BSA in PBS. The following primary antibodies were used: anti-INCENP (1:3000; 29419-1-AP, Proteintech), anti-FLAG (1:10,000; F1804, Sigma-Aldrich), anti-CENP-A (1:500; 3-19, MBL), and anti-CENP-B (1:1000, 61287, Active Motif). After incubation with secondary antibodies and counterstaining of DNA with 0.1 µg/ml DAPI, cells were mounted using ProLong Gold Antifade Mountant (Invitrogen). Images were acquired using a Plan-Apochromat ×100/1.46 NA oil immersion objective lens (Zeiss) on an LSM880 confocal laser scanning microscope (Zeiss).

## Analysis of inter-kinetochore distance and mitotic defects in anaphase

To obtain metaphase cells, cells were arrested with 10 µM MG132 for 1 h and stained for CENP-A and CENP-B as described above. CENP-A pairs were identified based on CENP-B signals, and the inter-kinetochore distance was measured using ImageJ. To analyze mitotic defects in anaphase, cells were arrested with 100 µM Monastrol for 3 h and then released. After 40 min, cells were fixed and stained with DAPI.

## Antibodies used for western blotting

The following primary antibodies were used for western blotting:
Anti-FLAG (1:5000; M2, Sigma);
Anti-mCherry (1:1000; GTX128508, Gene Tex);

Anti-INCENP (1:5000; 29419-1-AP, Proteintech);
Anti-Aurora B (1:500; 611083, BD Biosciences);
Anti-Borealin (1:1000; M147-3, MBL);
Anti-α-Tubulin (1:5000; clone B-5-1-2, #T6074, Millipore).

## FRAP

RPE1 cell lines were seeded onto chambered cover glasses (Thermo) the day before the experiment and, prior to imaging, were cultured at 37 °C for at least 1 h in Leibovitz's L-15 medium (Gibco) without phenol red, supplemented with 20% fetal bovine serum (FBS; Sigma) and 20 mM HEPES (pH 7.45) (Hatoyama et al, 2024). For FRAP analysis, cells were cultured at 27 °C to minimize chromosome movement. The mobility of Borealin-mNeonGreen and Borealin-mCherry in each cell line was analyzed using a confocal microscope (LSM880, Carl Zeiss) equipped with a ×63 objective lens (1.4 NA Plan Apochromat Oil DIC M27) for region-of-interest bleaching. Two images were acquired prior to bleaching (1% transmission of a 488-nm laser, 204.8 ms/frame with a 3 s interval, 256 × 256 pixels, 2 Airy unit pinhole, 10× zoom, 4-μm z-stack with 1-μm intervals). Borealin-mNeonGreen or Borealin-mCherry signals were bleached using 10% and 100% transmission of 405- and 488-nm or 568-nm lasers, respectively, followed by the acquisition of an additional 60 images using the same settings. The fluorescence intensity in the bleached region was quantified using Fiji software after z-stack projection to correct for centromere movement along the z-axis and background subtraction. Photobleaching during imaging was monitored and corrected before plotting the recovery curve. The recovery curve was plotted as relative values, setting the fluorescence intensities before and after bleaching as 1 and 0, respectively.

## Live-cell imaging

To analyze mitotic progression, live-cell imaging was performed using a confocal quantitative image cytometer CQ1 (Yokogawa). Cells were cultured overnight in a 24-well glass-bottom plate (Iwaki). DNA was stained with 100 nM SiR-DNA (Cytoskeleton, Inc.) for 1 h before image acquisition. Images were acquired every 1 min using a ×60 objective lens. A series of projected images from five Z-sections at 1.5 μm intervals were analyzed. For data analysis, images were processed using Fiji software (http://fiji.sc).

## RPE1 MNase and Southern blot

To obtain mitotic cells, cells were cultured with 200 nM Palbociclib for 24 h, released for 12 h, and then treated with 0.1 μg/mL nocodazole for 4 h before being collected by shake-off. After washing with PBS, $5 \times 10^5$ cells were suspended in a modified CSK buffer (0.1% Triton X-100, 100 mM NaCl, 10 mM PIPES [pH 7.0], 300 mM sucrose, 1 mM $MgCl_2$, 5 mM $CaCl_2$, 1 mM EGTA, and a protease inhibitor cocktail [Roche]) containing the indicated concentration of micrococcal nuclease (SIGMA) and incubated for 5 min at 37 °C. The reaction was stopped with 50 mM EDTA, followed by treatment with RNase A and a protease inhibitor. The digested DNA was then purified by phenol/chloroform extraction and ethanol precipitation. The DNA was electrophoresed on a 1.2% agarose gel in TAE buffer. After electrophoresis, the gel was denatured in 0.5 M NaOH with 1.5 M NaCl, neutralized in 0.5 M Tris (pH 7.5) with 1.5 M NaCl, and equilibrated in 20× SSC. The DNA was then transferred onto a Hybond-N+ membrane (Amersham) by capillary blotting overnight. Hybridization and probe preparation were performed using the ECL Direct Nucleic Acid Labeling and Detection System (Cytiva). Alpha-satellite DNA fragments were amplified using the following primers (Chan et al, 2012) and monomeric unit was purified. The percentage of high molecular weight chromatin in the MNase southern blots was quantified as described in (Gilbert et al, 2007).

Forward: 5′-CTCACAGAGTTGAACCTTCC-3′
Reverse: 5′-GAAGTTTCTGAGAATGCTTCTG-3

## Data availability

Cryo-EM structure and model are deposited in Protein Data Bank (PDB: https://www.rcsb.org/) under the following accession codes: Class 0—PDB ID 9SI9 (https://doi.org/10.2210/pdb9si9/pdb) and EMD-54926; Class 1—PDB ID 9SJ5 (https://doi.org/10.2210/pdb9sj5/pdb) and EMD-54938; Class 2—PDB 9SI3 (https://doi.org/10.2210/pdb9si3/pdb) and EMD-54924; Double occupancy RELION map—PDB 9SLJ (https://doi.org/10.2210/pdb9slj/pdb) and EMD-55003; Double occupancy cryoSPARC map—EMD-55012. The MS proteomics data (Crosslinking/MS and UV crosslinking/MS datasets) have been deposited to the ProteomeXchange Consortium via the PRIDE (Perez-Riverol et al, 2019) partner repository (project accession: PXD068775). Microscopy images are available in the BioStudies SUBMISSION TOOL, with accession code S-BSST2202.

The source data of this paper are collected in the following database record: biostudies:S-SCDT-10_1038-S44318-025-00594-y.

## Peer review information

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

## Acknowledgements

AAJ and his team are supported by the Wellcome Trust (senior research fellowship 202811; Discovery Award 309153/Z/24/Z), the European Union (ERC advanced grant CHROMSEG 101054950), and the Medical Research Council (MRC grant MR/X001245/1). The Wellcome Centre for Cell Biology is supported by core funding from the Wellcome Trust (grant 203149). AG is supported by the Darwin Trust of Edinburgh. TH lab is supported by the Japan Society for the Promotion of Science (JSPS) Grant-in-Aid for Scientific Research (24H01381, 24H02286 [to R-SN]; 24H02283, 22H04996 [to TH]), and by JST, CREST Grant Number JPMJCR21E6, Japan [to TH]. This work was supported by funding for the Wellcome Discovery Research Platform for Hidden Cell Biology [226791], and we gratefully acknowledge support from the Proteomics core, Protein Production (EPPF), and the Atomic Force Microscopy facility. MDW's work is supported by the Wellcome Trust (210493), Medical Research Council (T029471/1). KPH and his team are funded by Deutsche Forschungsgemeinschaft TTR237 and SFB1361. Views and opinions expressed are, however, those of the authors only and do not necessarily reflect those of the European Union or the European Research Council. Neither the European Union nor the granting authority can be held responsible for them.

## Author contributions

**Anjitha Gireesh**: Resources; Data curation; Formal analysis; Investigation; Methodology; Writing—original draft; Writing—review and editing. **Maria Alba Abad**: Resources; Data curation; Formal analysis; Validation; Investigation; Visualization; Methodology; Writing—original draft; Writing—review and editing. **Ryu-Suke Nozawa**: Resources; Data curation; Formal analysis; Validation; Investigation; Methodology; Writing—original draft; Writing—review and editing. **Paula Sotelo-Parrilla**: Investigation; Writing—review and editing. **Lea C Dury**: Investigation; Writing—review and editing. **Mariia Likhodeeva**: Investigation; Writing—review and editing. **Martin Wear**: Investigation; Writing—review and editing. **Christos Spanos**: Investigation; Writing—review and editing. **Cristina Cardenal Peralta**: Investigation. **Juri Rappsilber**: Software. **Karl-Peter Hopfner**: Funding acquisition. **Marcus D Wilson**: Funding acquisition; Investigation. **Willem Vanderlinden**: Funding acquisition; Investigation; Writing—original draft; Writing—review and editing. **Toru Hirota**: Funding acquisition. **A Arockia Jeyaprakash**: Conceptualization; Supervision; Funding acquisition; Methodology; Writing—original draft; Writing—review and editing.

Source data underlying figure panels in this paper may have individual authorship assigned. Where available, figure panel/source data authorship is listed in the following database record: biostudies:S-SCDT-10_1038-S44318-025-00594-y.

## Disclosure and competing interests statement

The authors declare no competing interests.

# Expanded View Figures

**Figure EV1. Reconstitution and characterization of CPC$_{I1-190SB}$-NCP complex.**

(A) Sequence alignment of Borealin orthologs from *Homo sapiens* (hs), *Bos taurus* (bt), *Mus musculus* (mm), *Gallus gallus* (gg), *Danio rerio* (dr), and *Xenopus laevis* (xl). The sequences are colored based on conservation with red being highly conserved and yellow being poorly conserved. The predicted secondary structure elements are shown below the sequence alignment. Multiple sequence alignment was performed with Clustal Omega (EMBL-EBI) and edited with Jalview 2.11.0 (Waterhouse et al, 2009). The secondary structure prediction was done using the web services in Jalview, using Jpred. The N-terminal tail, loop region and dimerization domain of Borealin are highlighted with boxes. (B) Sequence alignment of the first 190 amino acids of INCENP orthologs from *Homo sapiens* (hs), *Bos taurus* (bt), *Mus musculus* (mm), *Gallus gallus* (gg), *Danio rerio* (dr), and *Xenopus laevis* (xl). The sequences are colored based on conservation with red being highly conserved and yellow being poorly conserved. The predicted secondary structure elements are shown below the sequence alignment. Multiple sequence alignment was performed with Clustal Omega (EMBL-EBI) and edited with Jalview 2.11.0 (Waterhouse et al, 2009). The secondary structure prediction was done using the web services in Jalview, using Jpred. The basic IDR region of INCENP (including the "RRKKRR" motif and two additional positive stretches) and the PxVxI motif are highlighted with boxes in the alignment. (C) SEC chromatogram for CPC$_{I1-190SB}$-H3T3ph NCP complex formation and the corresponding SDS-PAGE gel for the run. The peak fractions were pooled and used to solve the cryo-EM structure of the complex. (D) Mass photometry analysis of the CPC-NCP complex. Population distribution of the CPC-NCP complex sample at 50 nM. Gaussian fitting identified three populations; 71 ± 13 kDa corresponding to CPC monomer (theoretically calculated 69.3 kDa), 122 ± 16 kDa corresponding to CPC dimer (theoretically calculated 138.6 kDa), and 340 ± 16.9 kDa which corresponds to the CPC-NCP complex with a stoichiometry of 2:1 (CPC:NCP; theoretically calculated 337.95 kDa). The event counts for each peak are also indicated below the mass estimation. (E) Representative cryo-EM micrograph with the scale bar corresponding to 120 nm. (F) Representative 2D-classes from the processing pipeline of the cryo-EM structure from cryoSPARC. Different orientations of NCP can be clearly identified from the classes. Scale bar equals 120 Å. Source data are available online for this figure.

▶

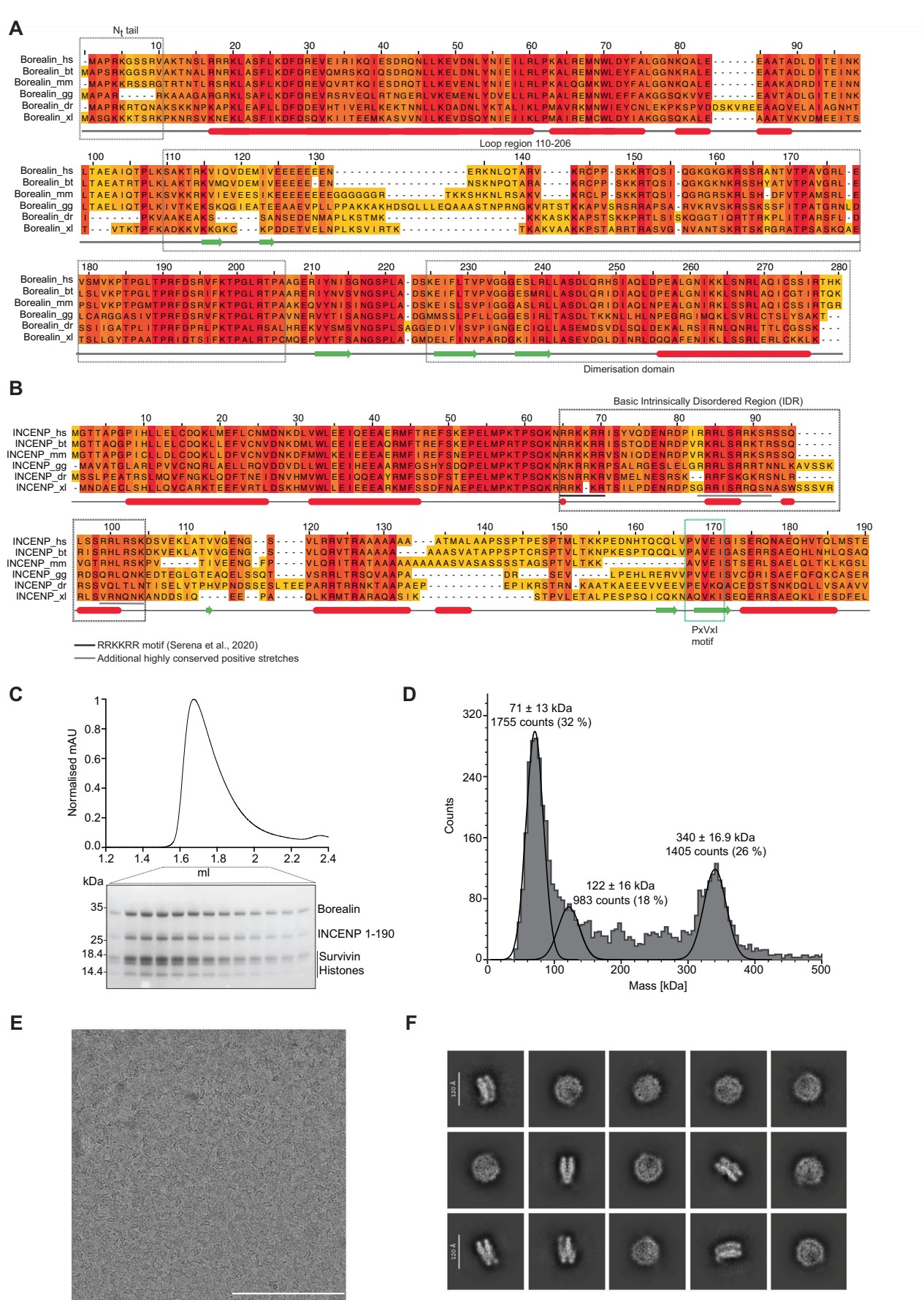

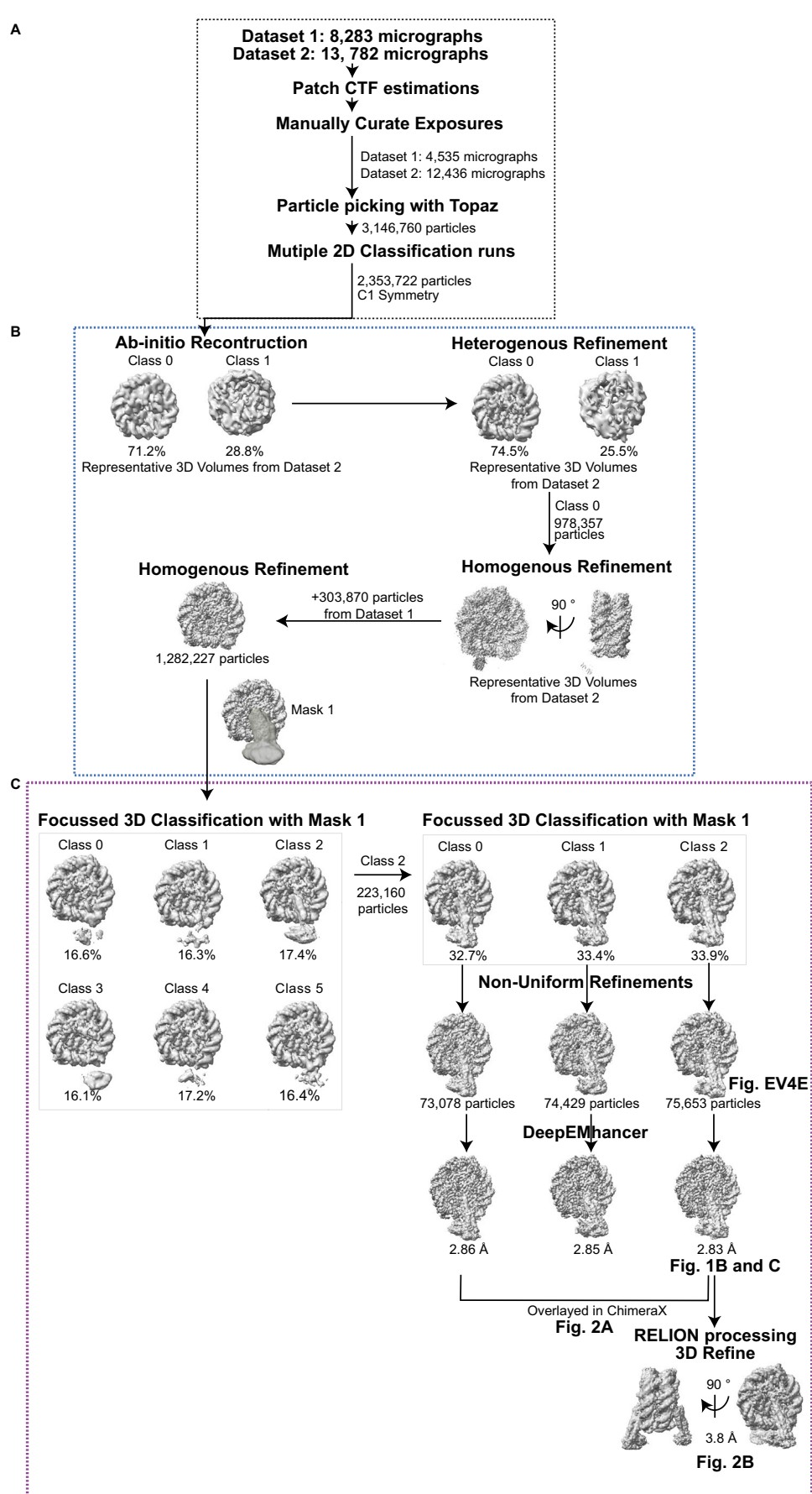

**A**

**Dataset 1: 8,283 micrographs**
**Dataset 2: 13, 782 micrographs**

**Patch CTF estimations**

**Manually Curate Exposures**

Dataset 1: 4,535 micrographs
Dataset 2: 12,436 micrographs

**Particle picking with Topaz**
3,146,760 particles

**Mutiple 2D Classification runs**

2,353,722 particles
C1 Symmetry

**B**

**Ab-initio Recontruction**
Class 0    Class 1

71.2%      28.8%
Representative 3D Volumes from Dataset 2

**Heterogenous Refinement**
Class 0    Class 1

74.5%      25.5%
Representative 3D Volumes
from Dataset 2

Class 0
978,357
particles

**Homogenous Refinement**

90°

Representative 3D Volumes
from Dataset 2

**Homogenous Refinement**

+303,870 particles
from Dataset 1

1,282,227 particles

Mask 1

**C**

**Focussed 3D Classification with Mask 1**
Class 0    Class 1    Class 2

16.6%      16.3%      17.4%

Class 3    Class 4    Class 5

16.1%      17.2%      16.4%

Class 2
223,160
particles

**Focussed 3D Classification with Mask 1**
Class 0    Class 1    Class 2

32.7%      33.4%      33.9%

**Non-Uniform Refinements**

73,078 particles    74,429 particles    75,653 particles

**Fig. EV4E**

**DeepEMhancer**

2.86 Å      2.85 Å      2.83 Å

**Fig. 1B and C**

Overlayed in ChimeraX
**Fig. 2A**

**RELION processing**
**3D Refine**

90°

3.8 Å

**Fig. 2B**

◀ **Figure EV2. The processing workflow of the CPC$_{I1-190SB}$-NCP cryo-EM dataset.**

(A) Initial stages of processing involving micrograph import, CTF estimation, particle picking and 2D classifications. (B) Generation of 3D volumes using ab initio reconstruction, heterogeneous and homogeneous refinements. Particles with good nucleosome density were selected for downstream processing. (C) Focused 3D classification runs with a mask for CPC density were performed to select particles with well-defined CPC-NCP density. Further non-uniform refinement and DeepEMhancer runs improved the quality of the EM map (Sanchez-Garcia et al, 2021). Particles from Class 2 of the final focused 3D classification were exported to RELION 4.0, where 3D Refine with blush regularization was performed to obtain a map with double occupancy (Kimanius et al, 2024). The maps shown in the different main and expanded view (EV) figures are indicated clearly with corresponding figure numbers.

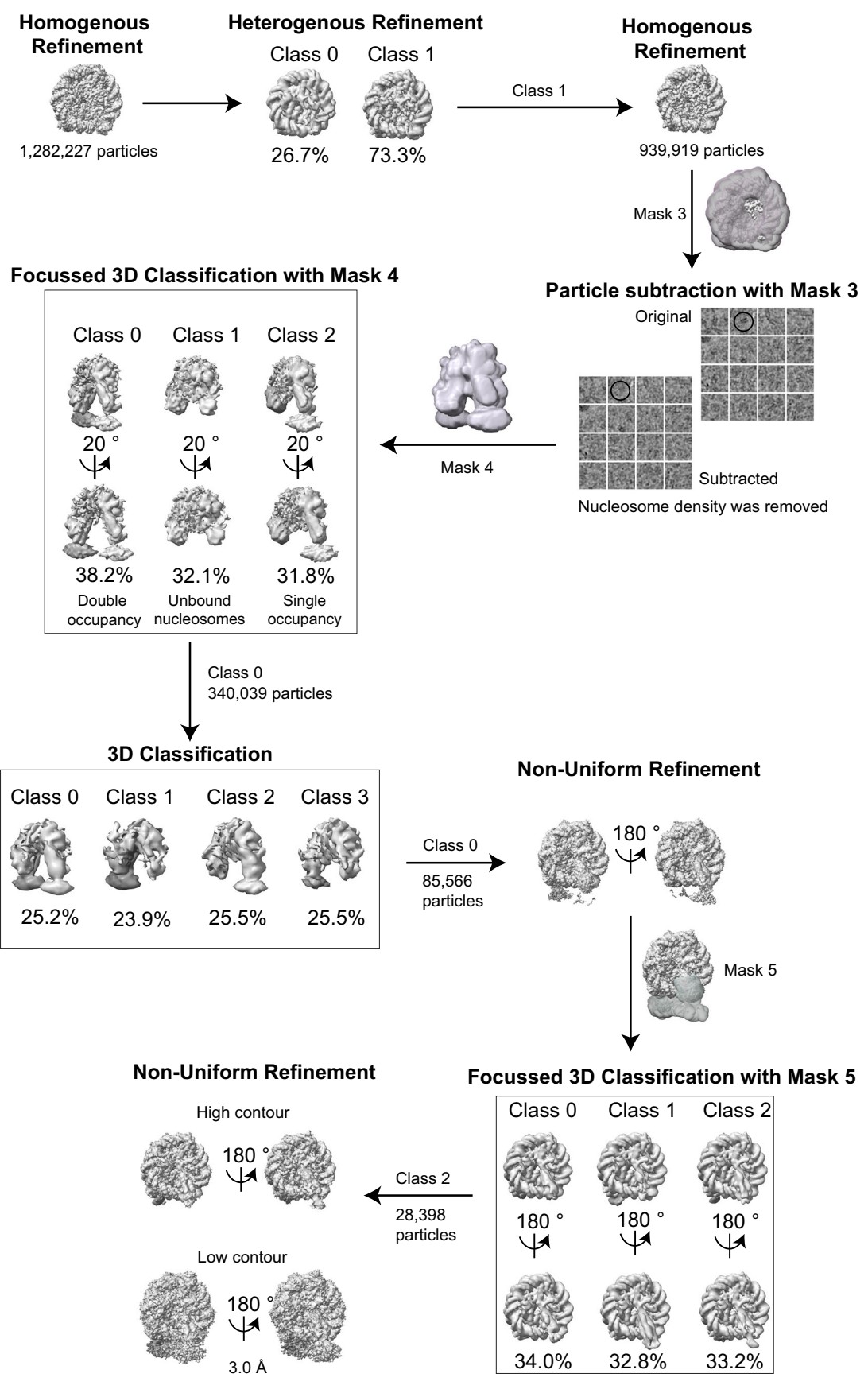

**Fig. EV4F**

◀ **Figure EV3. The alternate processing workflow of the CPC$_{I1-190SB}$-NCP cryo-EM dataset to obtain a double occupancy map.**

Particles from the homogeneous refinement job shown in Fig. EV2B were subjected to heterogeneous refinement and subsequently particle subtraction to remove the density for NCP. Focused 3D classification was performed to separate the doubly occupied NCPs from singly occupied and unbound NCPs. The selected class containing 85,566 particles (which makes up 6.67% of particles from the parent homogeneous refinement job) was further subjected to 3D classification runs and refinements to obtain the map depicted in Fig. EV4F.

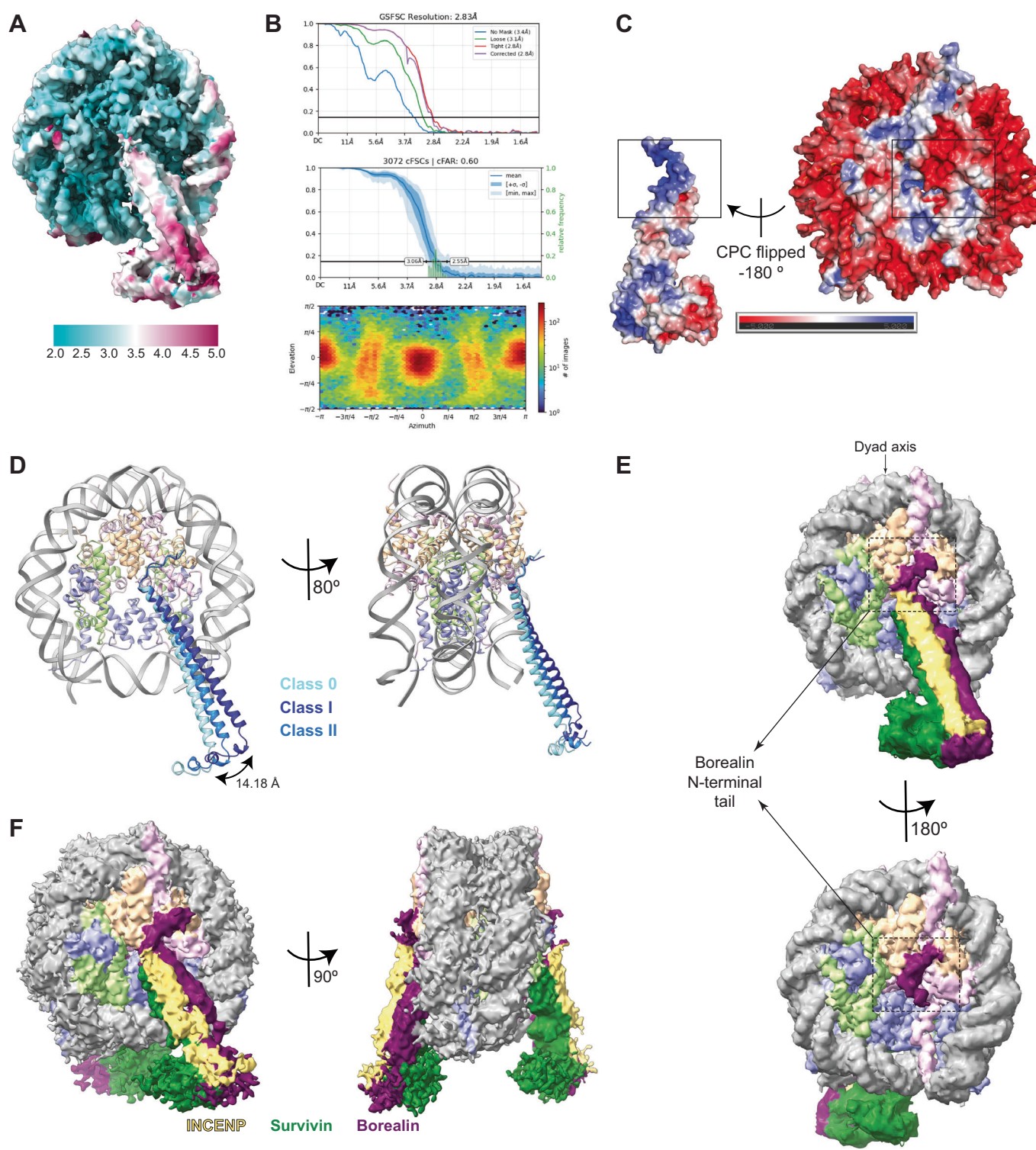

**Figure EV4.  Cryo-EM map and structure analysis of the CPC$_{I1-190SB}$-NCP complex.**

(A) Local resolution map of the cryo-EM density (from Fig. 1B,C) is depicted in a cyan to pink color gradient. The scale bar represents the corresponding resolution in Å. (B) Corresponding GSFSC plot, cFSCs curve, and azimuth plot from cryoSPARC for the cryo-EM density. (C) Cryo-EM model surface colored by electrostatic potential (APBS using Pymol version 3.1), with CPC rotated −180 degrees around the *y*-axis to highlight the highly basic Borealin N-terminal tail (box, left), which binds to the acidic patch of the nucleosome (box, right). (D) Corresponding atomic model for the composite density map shown in Fig. 2A (only the Borealin from each class is shown for clarity). Histones are depicted in violet (Histone H3), light green (Histone H4), pink (Histone H2A), and orange (Histone H2B), and the DNA is shown in gray. Borealin from the three classes representing the separate positions of CPC are shown in different shades of blue. CPC has flexibility in both directions, horizontally and vertically, while being tethered to the nucleosome via the N-terminal tail of Borealin. The left image depicts the horizontal flexibility of the complex, while the right image shows the vertical motion of CPC. The displacement is 14.18 Å (measured: Cα to Cα distance of Borealin between Class 0 and Class 1) (E) Non-Uniform refinement map for Class 2 (parent to the DeepEMhancer map depicted in Fig. 1B,C) (top), and rotated 180 degrees across *y*-axis (bottom) to visualize the Borealin N-terminal tail occupancy on the opposite face of the nucleosome. (F) Cryo-EM density map obtained through the alternate processing pipeline detailed in Fig. EV3. CPC densities can be seen on both faces of the nucleosome.

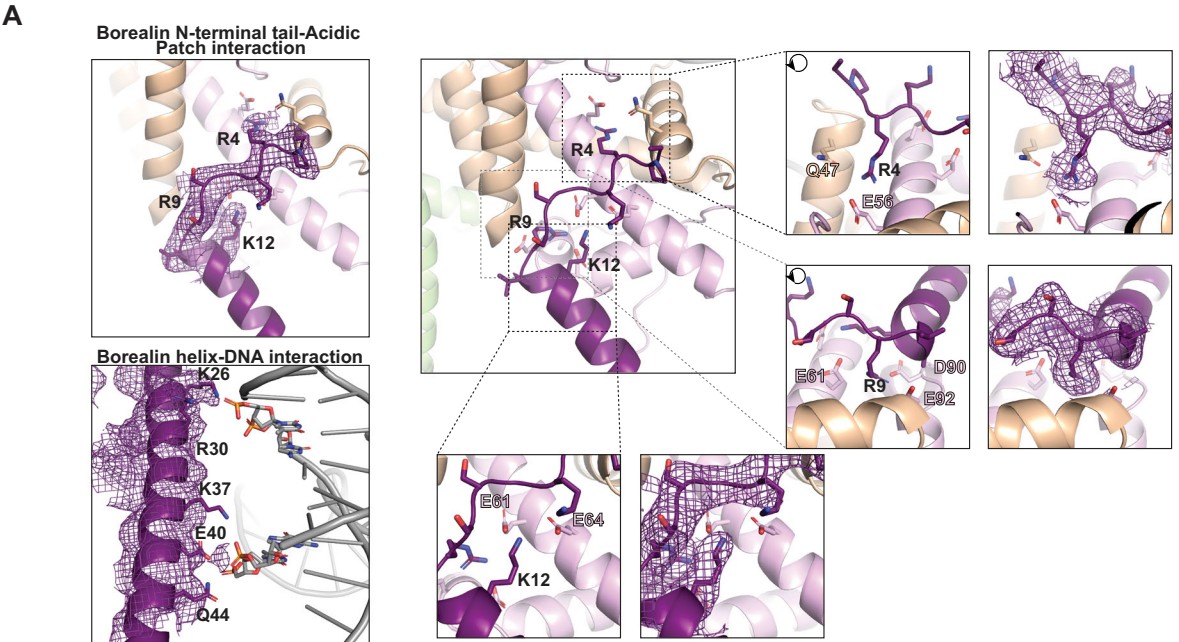

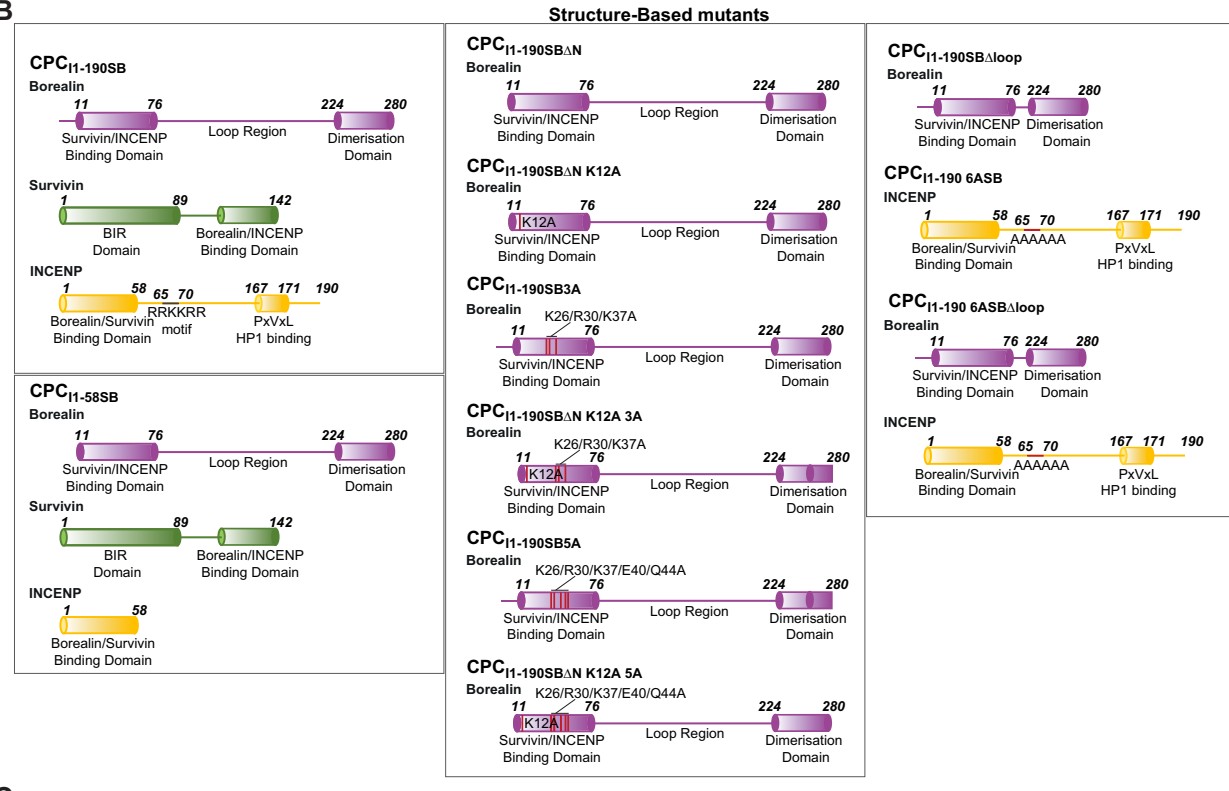

**Figure EV5.   Quality of the Cryo-EM map at key interaction interfaces and cartoons of CPC$_{I1-190SB}$ mutants highlighting the location of the mutations.**

(A) The corresponding zoomed-in image of the N-terminal tail-acidic patch interaction (left, top), and the Borealin helix-DNA interaction (left, bottom) of Fig. 1B,C, respectively, with the EM density depicted as a mesh. The zoomed-in image of the N-terminal tail-acidic patch interaction involving the Borealin residues Arg 4, Arg 9, and Lys 12 is depicted in the center. The three interactions: Borealin Arg 4 interaction with Gln 47 of H2B and Glu 56 of H2A (right, top), Borealin Arg 9 interaction with Glu 61, Asp 90, and Glu 92 of H2A (right, bottom), and Borealin Lys 12 interaction with Glu 61 and Glu 64 of H2A (center, bottom). The panels adjacent to each depict the EM density as a mesh. (B) Schematic diagram depicting the domain architectures of CPC$_{I1-190SB}$ and CPC$_{I1-58SB}$. The domain architecture for the mutated proteins is depicted for the structure-based mutants (CPC$_{I1-190SBΔN}$, CPC$_{I1-190SBΔN\ K12A}$, CPC$_{I1-190SB3A}$, CPC$_{I1-190SBΔN\ K12A\ 3A}$, CPC$_{I1-190SB5A}$, and CPC$_{I1-190SBΔN\ K12A\ 5A}$) and displayed in the center panel. Additionally, the domain architecture for CPC$_{I1-190SBΔloop}$, CPC$_{I1-190\ 6ASB}$, and CPC$_{I1-190\ 6ASBΔloop}$ constructs is depicted on the right. Point mutations are depicted with red bars. (C) Representative SDS-PAGE gel with the CPC$_{I1-190SB}$, CPC$_{I1-190SBΔN}$, CPC$_{I1-190SBΔN\ K12A}$, CPC$_{I1-190SB3A}$, CPC$_{I1-190SBΔN\ K12A\ 3A}$, CPC$_{I1-190SB5A}$, and CPC$_{I1-190SBΔN\ K12A\ 5A}$, CPC$_{I1-190SBΔloop}$, CPC$_{I1-190\ 6ASB}$, and CPC$_{I1-190\ 6ASBΔloop}$ proteins used for SPR and EMSA experiments shown in Figs. 3, 4, and EV6. Source data are available online for this figure.

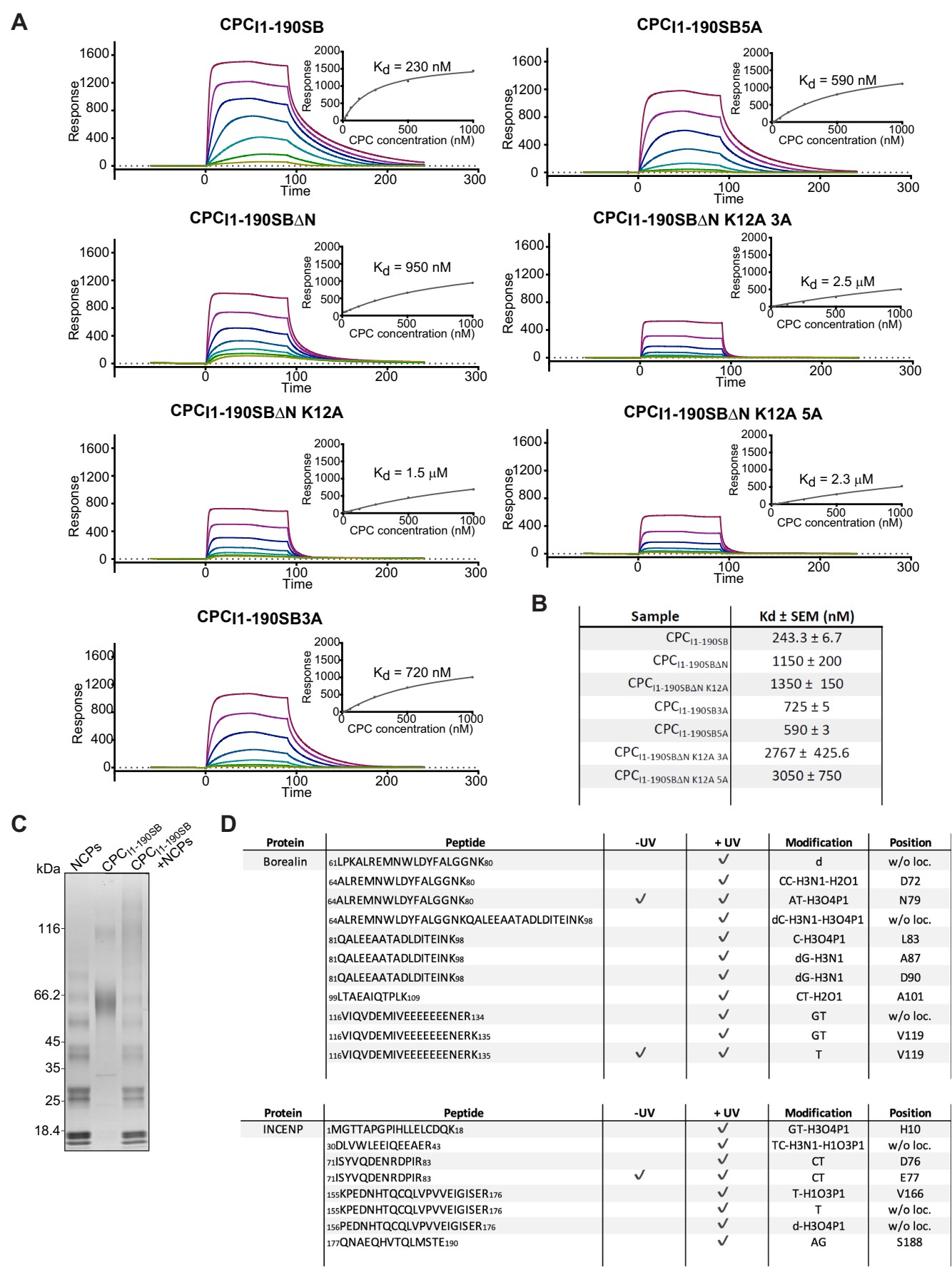

Figure EV6.   **SPR and UV crosslinking/MS analysis of the CPC$_{I1-190SB}$-NCP complex.**

(A) Representative SPR sensorgrams of the interaction between different CPC complexes (CPC$_{I1-190SB}$, CPC$_{I1-190SB\Delta N}$, CPC$_{I1-190SB\Delta N\ K12A}$, CPC$_{I1-190SB3A}$, CPC$_{I1-190SB\Delta N\ K12A\ 3A}$, CPC$_{I1-190SB5A}$, and CPC$_{I1-190SB\Delta N\ K12A\ 5A}$) and H3T3ph NPCs immobilized on the surface of a neutravidin sensor chip. Minimum of two biological replicates. (B) Mean values determined for the equilibrium $K_d$ are shown in the table. (C) Representative SDS-PAGE of the EDC-crosslinked sample from Fig. 4A. (D) Sequences of the DNA-modified Borealin and INCENP peptides identified in the UV crosslinking experiments. Presence or absence in the non-UV crosslinked (control) and the UV crosslinked conditions are shown, together with the modification assigned and the position. Source data are available online for this figure.

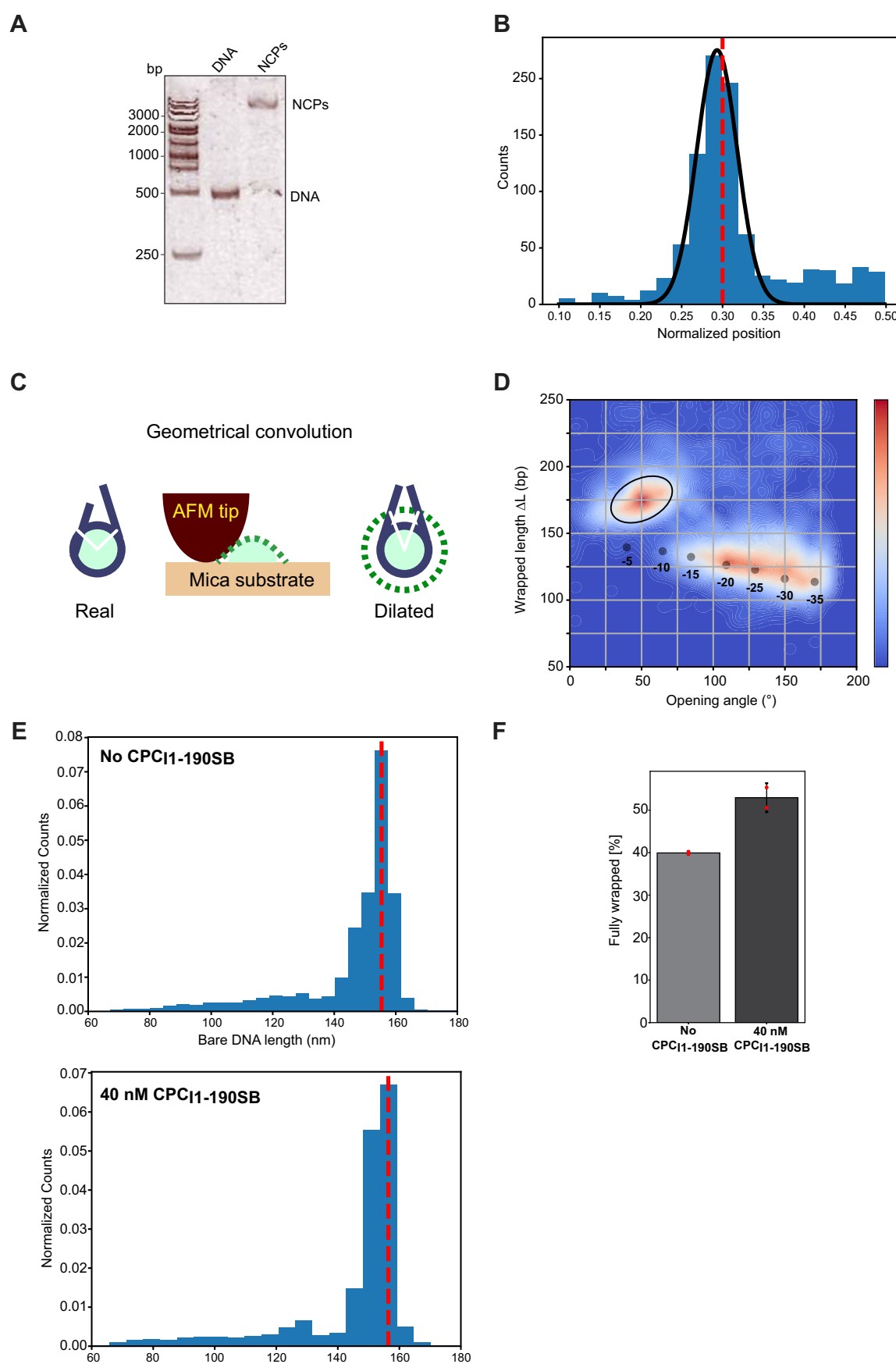

**Figure EV7.  Characterization of CPC<sub>I1-190SB</sub>-NCP complex by AFM.**

(A) Native gel with DNA construct used for AFM, before and after reconstitution with H3T3ph octamers. (B) Distribution of normalized nucleosome positions, quantified as the ratio of the short arm length $l_1$ over the sum of the arm lengths $l_1 + l_2$. Only fully wrapped nucleosomes are taken into account because asymmetric nucleosome unwrapping shifts the distribution. The red dashed line indicates the nucleosome position as expected from the DNA construct design. The full black line is a Gaussian fit to the data with a mean value of 0.29. (C) Schematic depiction indicating that geometrical convolution by the AFM tip distorts the real structure and leads to an underestimation of the nucleosome opening angle. (D) Two-dimensional kernel density estimate with indicated positions of different unwrapping states (numbers of bp unwrapped with respect to the fully wrapped state), as deduced from AFM image simulations (Konrad et al, 2022). (E) AFM characterization of bare DNA constructs in H3T3ph nucleosomes nucleosome samples in the absence (top) and presence (bottom) of CPC and reconstituted DNA contour length distribution as measured via automated readout. The mode of the distribution (red dashed line) is at contour length 155.4 nm in the absence and 156.4 nm in the presence of CPC, corresponding to 0.32 nm/bp, in good agreement with previous AFM measurements of DNA length (Rivetti et al, 1996; Konrad et al, 2021b). The experimental rise per base pair is used to convert the units of wrapped length $\Delta L$ from nm to bp. (F) The fraction of fully wrapped nucleosomes is significantly different in the absence and presence of 40 nM CPC<sub>I1-190SB</sub>. Red datapoints represents calculated fractions for each biological repeat, indicating good agreement and reproducibility. Source data are available online for this figure.

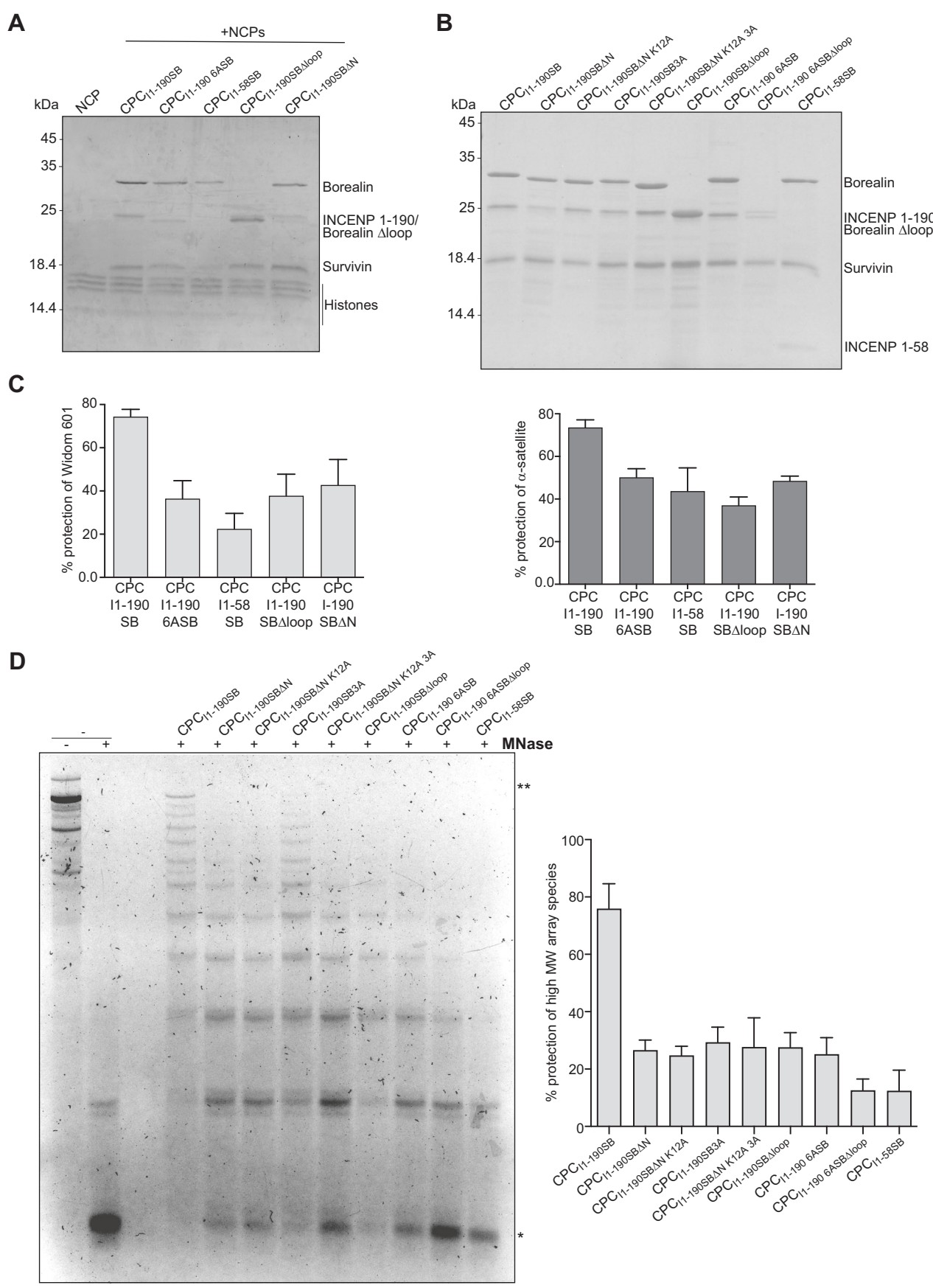

**H3T3ph 12-mer array**

**Figure EV8.   MNase protection assays with CPC$_{I1\text{-}190SB}$ mutants.**

(A) SDS-PAGE gel with the inputs for the mono-nucleosome MNase assay shown in Fig. 5E. (B) SDS-PAGE gel with the CPC inputs of the MNase assay with 12-mer nucleosomal arrays shown in Fig. 5F. (C) Quantification of the mono-nucleosome MNase assay for Widom 601 NCP (left), and α-satellite NCP (right) from Fig. 5E ($n = 3$, mean ± SEM; data representative of three biological replicates). (D) Agarose gel for the MNase assay depicted in Fig. 5F. 12-mer array DNA is highlighted with double asterisks, and monomer DNA is highlighted with a single asterisk. Percentage of protection of higher MW array (4-mer to 12-mer) against MNase degradation quantified in the bar graph on the right ($n = 3$; mean ± SEM; data representative of three biological replicates). Source data are available online for this figure.

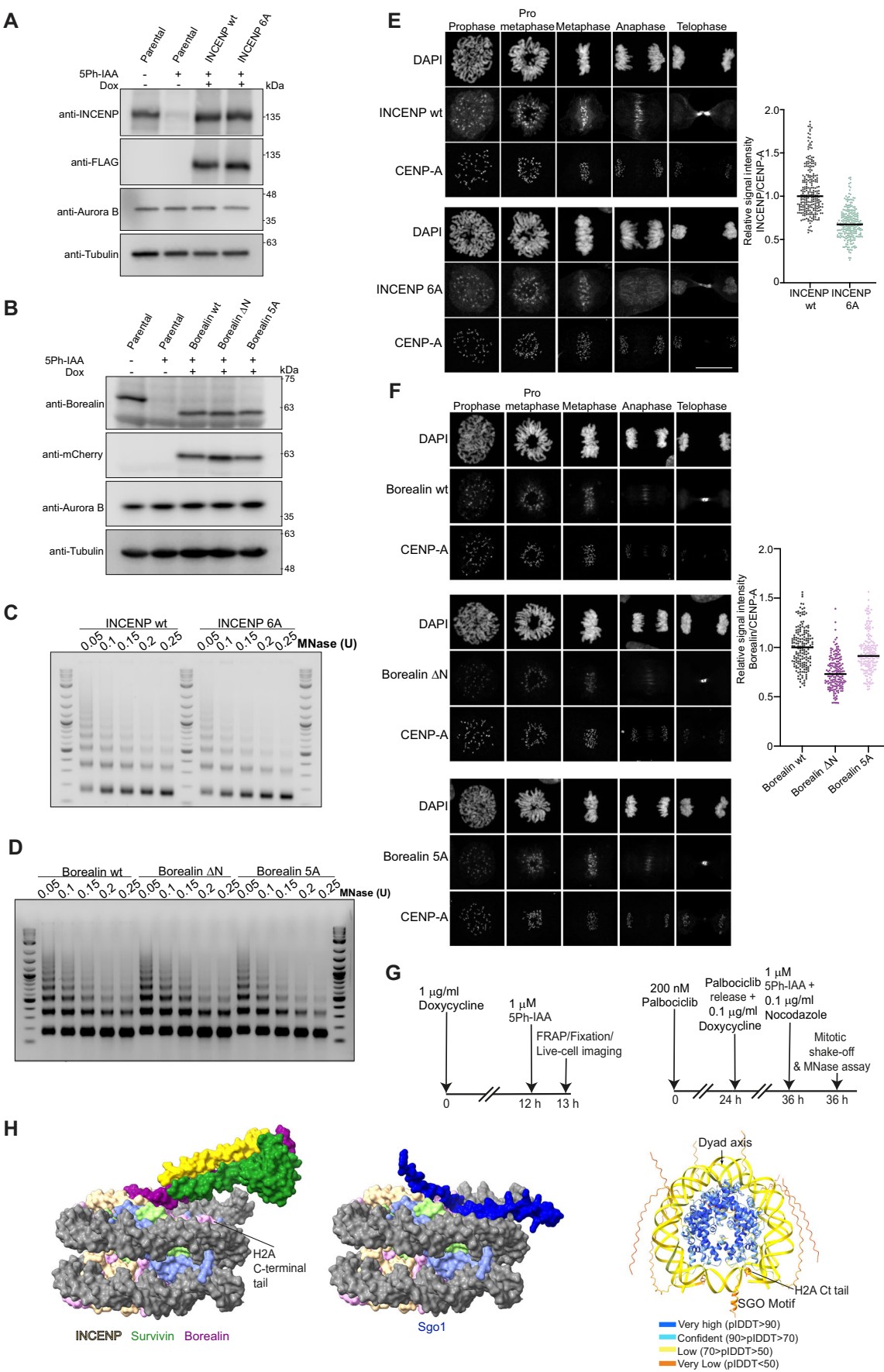

◄ **Figure EV9. Implications of CPC-NCP interaction on the centromeric levels of CPC and Sgo1 binding.**

(A) Representative immunoblot for RPE1 cell lines expressing either INCENP wt or INCENP 6A mutant showing the expression levels of the different INCENP constructs used in Fig. 6. (B) Representative immunoblot for RPE1 cell lines expressing either Borealin wt, Borealin ΔN, and Borealin 5A showing the expression levels of the different INCENP constructs used in Fig. 6. (C) Ethidium bromide-stained agarose gel image corresponding to the MNase assay presented in Fig. 6A. (D) Ethidium bromide-stained agarose gel image corresponding to the MNase assay presented in Fig. 6B. (E) (Left panel) Representative fluorescence images for INCENP wt and INCENP 6A centromere localization at different stages of mitosis. DAPI was used for DNA staining and CENP-A for centromere staining. Scale bar, 10 μm. (Right panel) Scatter plot for the quantification of INCENP wt and INCENP 6A centromere localization during prometaphase. Relative fluorescence intensities of INCENP wt (black) and INCENP 6A (green), normalized to CENP-A, were measured in at least ten cells from two biological replicates. Individual data points are plotted. Horizontal black lines indicate median values. (F) (Left panel) Representative fluorescence images for Borealin wt, Borealin ΔN, and Borealin 5A centromere localization at different stages of mitosis. DAPI was used for DNA staining and CENP-A for centromere staining. Scale bar, 10 μm. (Right panel) Scatter plot for the quantification of Borealin wt, Borealin ΔN, and Borealin 5A centromere localization during prometaphase. Relative fluorescence intensities of Borealin wt (black), Borealin ΔN (purple), and Borealin 5A (pink), normalized to CENP-A, were measured in at least ten cells from two biological replicates. Individual data points are plotted. Horizontal black lines indicate median values. (G) (left panel) Diagram with the experimental timeline followed for the RPE1 experiments shown in Fig. 6C–E. (Right panel) Diagram with the experimental timeline followed for the RPE1 experiment shown in Fig. 6A,B. (H) Atomic model for the CPC-NCP complex (Class 0) with the H2A C-terminal tail indicated (left). AlphaFold 3 prediction of SGO motif (amino acids 466–527) of Sgo1 binding to H2AT120ph NCP (Abramson et al, 2024). The prediction colored by plDDT values is depicted on the right. The SGO motif-H2A tail interaction is predicted with low confidence (70 > plDDT > 50). The predicted model agrees with previously available biochemistry on the interfaces involved in SGO motif-NCP binding (Kawashima et al, 2010; Liu et al, 2013, 2015). CPC binding depicted on the left, occludes H2A C-terminal tail, likely making it inaccessible for Sgo1 binding. Source data are available online for this figure.

# A

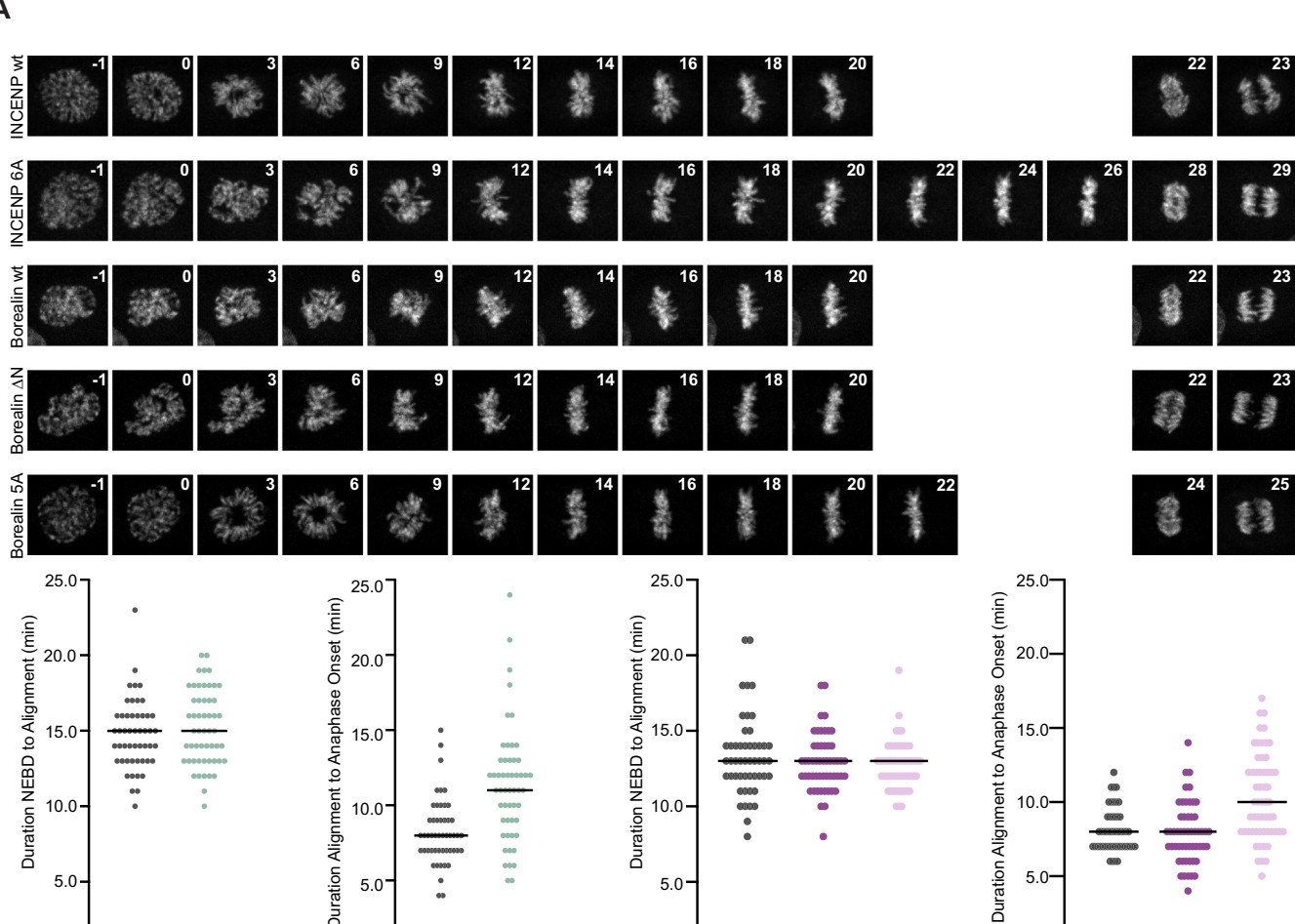

**Figure EV10. Impact of perturbing CPC-NCP interaction on the timing of anaphase onset.**

(**A**) Representative images from live-cell imaging experiments assessing mitotic timing in at least 50 cells from two biological replicates for INCENP wt (black), INCENP 6A (green), Borealin wt (black), Borealin ΔN (purple), and Borealin 5A mutant (pink). Scatter plots show the elapsed time (in minutes) from nuclear envelope breakdown (NEBD) to metaphase alignment and from metaphase alignment to anaphase onset for individual cells. Individual data points are plotted. Horizontal black lines indicate median values. Source data are available online for this figure.

