## [Peer Review File · The EMBO Journal]

Nucleosome Interaction of the CPC Secures Centromeric Chromatin Integrity and Chromosome Segregation Fidelity

Anjitha Gireesh, Maria Alba Abad, Ryu-Suke Nozawa, Paula Sotelo-Parrilla, Lea Dury, Mariia Likhodeeva, Martin A. Wear, Christos Spanos, Cristina Peralta, Juri Rappsilber, Karl-Peter Hopfner, Marcus Wilson, Willem Vanderlinden, Toru Hirota, and A. Arockia Jeyaprakash

Corresponding author(s): A. Arockia Jeyaprakash (jeyaprakash.arulanandam@ed.ac.uk)

Review Timeline:

Submission Date:	15th May 25
Editorial Decision:	13th Jun 25
Revision Received:	6th Sep 25
Editorial Decision:	21st Sep 25
Revision Received:	26th Sep 25
Accepted:	30th Sep 25

Editor: Hartmut Vodermaier

Transaction Report:

Prof. A. Arockia Jeyaprakash
University of Edinburgh
Wellcome Trust Centre for Cell Biology, Institute of Cell Biology
Kings Buildings
Max Born Crescent
Edinburgh EH9 3BF
United Kingdom

13th Jun 2025

Re: EMBOJ-2025-121378
Chromatin Protection by the Chromosomal Passenger Complex

Dear JP,

Thank you for submitting your manuscript on chromatin protection by CPC for our consideration. Three expert referees have now evaluated it, and provided the comments copied below. As you will see, they all appreciate the importance of the subject, as well as the overall technical quality of the work. However, they at the same time also raise a number of substantive concerns regarding presentation, interpretation, and decisiveness of the evidence in support of some of the conclusions.

Should you be able to satisfactorily address these issues, we would be interested in pursuing a revised version further for publication. Since it is our policy to allow only a single round of major revision, I would very much encourage you to contact me with a revision plan and preliminary point-by-point response already during the early stages of your revision work, so that we could discuss if and how the main points could be resolved; or whether a less completely revised manuscript might alternatively become suitable for publication in one of our sister journals like EMBO Reports or Life Science Alliance. We would also be open to extension of the default three-months revision period if needed; our 'scooping protection' (meaning that competing work appearing elsewhere in the meantime will not affect our considerations of your study) would of course remain valid throughout the whole period.

Detailed information on preparing, formatting and uploading a revised manuscript can be found below and in our Guide to Authors. Thank you again for the opportunity to consider this work for The EMBO Journal, and I look forward to hearing from you in due time.

With kind regards,

Hartmut

3) Revised manuscript text (including main tables, and figure legends for main and EV figures) has to be submitted as editable

text file (e.g., .docx format). We encourage highlighting of changes (e.g., via text color) for the referees' reference.

4) Each main and each Expanded View (EV) figure should be uploaded as individual production-quality files (preferably in .eps, .tif, .jpg formats). For suggestions on figure preparation/layout, please refer to our Figure Preparation Guidelines:

8) Please note that supplementary information at EMBO Press has been superseded by the 'Expanded View' for inclusion of additional figures, tables, movies or datasets; with up to five EV Figures being typeset and directly accessible in the HTML version of the article. For details and guidance, please refer to:

embopress.org/page/journal/14602075/authorguide#expandedview

9) To facilitate reproducibility and cross-laboratory adoption of methodologies, please structure the Materials & Methods section as outlined in our guide to authors, including a completed Reagents and Tools Table that can be downloaded from our author guidelines as well (<https://www.embopress.org/page/journal/14602075/authorguide#structuredmethods>).

10) Digital image enhancement is acceptable practice, as long as it accurately represents the original data and conforms to community standards. If a figure has been subjected to significant electronic manipulation, this must be clearly noted in the figure legend and/or the 'Materials and Methods' section. The editors reserve the right to request original versions of figures and the original images that were used to assemble the figure. Finally, we generally encourage uploading of numerical as well as gel/blot image source data; for details see: embopress.org/page/journal/14602075/authorguide#sourcedata

Further information is available in our Guide For Authors:

In the interest of ensuring the conceptual advance provided by the work, we recommend submitting a revision within 3 months (11th Sep 2025). Please discuss the revision progress ahead of this time with the editor if you require more time to complete the revisions. Use the link below to submit your revision:

Link Not Available

Referee #1:

The study focuses on the 2.86 Å resolution cryoEM structure of the chromosome passenger complex (CPC) in complex with the nucleosome core particle (NCP). The structure shows the N-terminal part of the borealin interacting with the acidic patch of the nucleosomes. This is consistent with the study published by the authors in 2019 (Abad et al, JCB). However, when the authors either mutate the residues of the acidic patch in the NCP or remove the N-terminal part of the borealin, they see that CPC and NCP still form the complex (readout: native gel and co-elution in SEC). From this, they conclude that other, more "dynamic" interactions (not visible in their structure) are important for the interaction between the two entities. To prove this, they performed two types of cross-linking and concluded that it must be the positively charged 'RRKKRR' motif in INCENP that interacts with histones, and the borealin loop that interacts with DNA. They also perform MNase digestion on mononucleosomes and 12-mer nucleosome arrays in the presence and absence of CPC and conclude that CPC protects NCP from MNase digestion. They use AFM to determine that CPC reduces the opening angle of DNA on NCP. Finally, they perform FRAP and MNase digestion of cell chromatin and conclude that the 'RRKKRR' motif in INCENP confers more stability to centromeric chromatin in cells.

The study addresses a highly relevant topic and provides novel structural insights into the interaction between the CPC and the nucleosome. However, there is a lack of satisfactory correlation between the proposed structural model and the functional experiments intended to validate it. For instance, the Cryo-EM data indicate that the most stable and significant interactions

occur between the nucleosome's acidic patch and the N-terminal region of Borealin (specifically, the first 10 amino acids). Although the authors generated a mutant lacking these first 10 amino acids, this mutation does not appear to disrupt nucleosome binding. This finding stands in stark contrast to previous results from the same laboratory (Abad et al., 2019), where the identical mutation impaired nucleosome binding.

To strengthen the study, it is crucial to quantitatively assess the effects of mutations on the interacting surfaces. More rigorous binding experiments would enhance the quality of the work and provide a clearer understanding of the individual contributions of various interactions to CPC-nucleosome stability.

More detail comments to help the authors elevate the quality of the manuscript are below:

- The introduction is unnecessarily long. It should focus on the mayor question that the experiments are addressing.
- Pg 5. "This mode of anchoring facilitates the engagement of the downstream triple helical bundle formed by Borealin, Survivin and INCENP with the nucleosome DNA entry/exit site (Fig 1C)."
 - o From the Figure 1, it is not clear how is nucleosome positioned. Authors should label the dyad and the entry/exit site and include the zoom in on the entry/exit site showing interactions with CPC.
- Pg 5. "3D class analysis of a population of particles with well-defined CPC densities revealed three discrete conformational states of the CPC triple helical bundle"
 - o There is no detailed particle selection protocol in the supplementary figures, explaining particle picking and refinement protocol including number of particles in each class.
- Pg5. "This, along with 3D variability analysis, shows that CPC, with the Borealin N-terminal region tethered at the nucleosome acidic patch, swings both vertically and horizontally (Fig. 2A, Movie1)"
 - o Movie1 is just showing different contours of the map in different orientations. It does not demonstrate swinging of the CPC.
- Pg 5. "Removing the signal of bound CPC using particle subtraction and subsequent refinement revealed a 2nd copy of the CPC occupying the symmetrically equivalent second face of the nucleosome in a similar conformation (Fig. 2B)"
 - o What kind of "subsequent refinement" was used here to make other CPC "appear"? In the composite map the orientation and the intensity of the "other" CPC is almost the same as of the original one. If this is the case, why was it the other molecule not visible before the substruction? What is the percentage of particles that have two CPCs bound?
- Pg 5. "A detailed analysis of the intermolecular interactions at the NCP acidic patch showed that Borealin N-terminal tail amino acid residues Arg 4 and Arg 9 form the arginine anchor that interacts with the H2A/H2B Asp/Glu residues of the nucleosome acidic patch, through salt bridge interactions (Fig. 1B)."
 - o In my opinion, this is the most important part of the results from the structure that the authors obtained and the figure is not convincingly showing that they have trustable density to conclude about those interactions. They should show a clear density around N-terminal tail and then zoom on the two Arginines in Borealin showing their connection with the side chains of the nucleosome.
- Pg 5. "The CPC11-190SB-H3T3ph NCP structure also revealed that the Borealin helix facing the DNA entry-exit site is highly basic with residues Lys 26, Arg 30, Lys 37 and Lys 48, among which Arg 30 makes Van der Waals contacts with the phosphate backbone of the DNA between super-helical location (SHL) 6 and 7 (Fig. 1C)."
 - o This is also not visible from the figure. Each statement like this has been backed with a clear figure showing both (density; could be in a supplement) and side chains of the the aminoacids making interactions.
- Pg 5. "Additionally, Gln 44 of Borealin makes H-bonding interactions with the backbone phosphates of nucleotide at the DNA entry-exit site (Fig. 1C)."
 - o Again, this is not clearly visible in the figure. Additionally, in the provided pdb model Glu 40 is modelled close to the DNA while Gln 44 is not within the salt bonding distance.
- Pg. 5. "Dynamic interactions are required for efficient CPC-nucleosome binding".
 - o The authors repeatedly use the term dynamic interaction. All non-covalent interactions - whether protein-protein or protein-ligand - are inherently dynamic, with varying degrees of strength and specificity. Authors should not use term dynamic interactions to refer specifically to interactions with low affinity, low specificity, or otherwise weak interactions.
- Fig. 4. Band C. And text on pg.7 "These observations suggest that while Borealin loop region contributes to CPC-nucleosome affinity, the dynamic contacts involving INCENP 'RRKKRR' might contribute to the dynamics of CPC-NCP binding."
 - o How does INCENP RRKKRR contribute to the dynamics of the CPC-NCP binding? It seems difficult to provide an explanation if mutations of the positive INCENP patch have no measurable effect on binding.
- Pg 6. "Furthermore, we purified a CPC complex lacking the N-terminal 10 residues of Borealin (CPC I1-190SB10-end) and tested its ability to interact with H3T3ph NCPs using size exclusion chromatography (SEC) (Fig. 3B). SEC analysis shows that CPC I1-190SB10-end can form a complex with H3T3ph NCPs (Fig. 3B)."
 - o Interestingly, in their previous work (Abad et al, JCB, 2019), the authors used CPC with borealin (10-109) and found a complete absence of binding in both EMSA and SPR, which contrasts with CPC with borealin (1-109). From this they conclude that the first 10 amino acids of borealin are essential for binding. Why do the authors not comment on the contradictory results they have now obtained? Again, a more quantitative measurement of the interaction would be required. Otherwise, the reported structure is almost useless because it leaves the reader with the impression that mutation of residues essential for binding (either on NCP or CPC) does not affect binding.
- Pg 6. "To assess the contribution of dynamic interactions crucial for CPC-nucleosome binding, we performed crosslinking/MS experiments with the CPC11-190SB -H3T3ph NCP complex (Fig. 3C and Fig. S3A)."
 - o How far are residues that this crosslinker connects and what is the chemistry behind? This is not explained neither in the text nor in the M&M.

o What is the conclusion here? Can these interactions be modeled?

- Pg.6. "UV-crosslinking of CPC-NCP complex, followed by SDS-PAGE analysis of the nucleosomal DNA revealed the presence of DNA crosslinked with Borealin migrating at the expected molecular weight (Fig. 4A and S3B)."

o Again what is chemical nature of this cross-linking? What is the experiment aiming to resolve?

- Figure 5F. What are numbers 1, 2 in the figure?

- Pg. 8. "Consistent with our mononucleosome MNase data, all Borealin and INCENP mutants showed reduced MNase protection activity (Fig 5F)."

o Is the conclusion that all different unstructured parts are interacting with DNA and are thus providing stability against MNase?

- Pg. 10. "We previously showed that Borealin dimerization domain contributes to efficient nucleosome binding, but how it does so remained unclear (Abad et al, 2019). Our observation that two copies of CPC engage with both faces of the nucleosome (related by a two-fold symmetry) suggests the potential contribution of Borealin dimerization in facilitating this interaction."

o Apart from not really having any solid evidence that nucleosomes are binding 2 CPC molecules at the same time ("composite map" is at best "sketchy"), the nucleosome, due to its symmetry has two identical faces that can theoretically accommodate the same type of ligand. So even if nucleosome would have two copies of the CPC, one on each face, that would not imply that these two are connected by the dimerization domain. They could be binding each in their monomeric form, thus such speculation does not have any grounds.

- Pg.11. "Note: A recent study on biorxiv, Ruza et al, 2025, reports a cryo-EM structure of CPC bound H3T3ph nucleosome.

While the mode of nucleosome binding by CPC broadly agrees with observations reported here, their structure shows just one copy of CPC bound to nucleosome, possibly because their CPC construct lacks more than two-thirds of the C-terminal region of Borealin (including the Borealin loop and dimerization domain)."

o If authors want to comment on the complementary study on bioRxiv (and I think they should) then they should do that in the systematic way and throughout the manuscript.

Referee #2:

This interesting manuscript by Gireesh and colleagues reports a study of the interaction of the localization module of the chromosome passenger complex (CPC) with an H3- nucleosome. The manuscript includes a cryo-EM characterization of the complex of the so-called localization module of the CPC with an H3-modified nucleosome (containing histone H3 pre-phosphorylated on threonine 3 and obtained with chemical ligation), as well as further work of biochemical and biological characterization of the interaction. The best-defined feature of the structure demonstrated binding of the N-terminal tail of Borealin to the acidic patch of the nucleosome, whereas the interaction of the H3 tail with Survivin was invisible. The Borealin N-terminal tail, however, was not necessary for the interaction, and the authors focused on other "dynamic" interaction patches, also invisible in the structure but important for overall stability of centromere localization of the CPC. Special emphasis here was given to a 'RRKKRR' sequence on INCENP previously shown by the Barr and Grunerberg laboratories to be important for CPC localization. The authors combined DNA crosslinking experiments and chromatin imaging and protection assays to conclude that the CPC has a non-catalytic role in the stabilization of centromeric chromatin. They also provide evidence that the RRKKRR' motif stabilizes the interaction of the CPC with the centromere, with relatively minor consequences for mitotic progression and a more significant effect on central spindle localization after anaphase. With nuances, this manuscript reaches conclusions that are related to, and consistent with, those reported in a manuscript by Ruza and co-workers (Barr and Gruneberg laboratories) currently in the bioRxiv.

In general, I feel that this manuscript addresses an important topic from multiple angles, providing a very good basis for further work in this complex area of research. The work is technically very well done. The presentation requires more clarity on some key points, as detailed below. I am therefore supportive towards publication, and would kindly ask the authors to consider the following points:

Specific points

-The first part of the Abstract is well written, but the second could be improved. Three claims may require the authors' attention. First, the sentence "CPC employs multipartite interactions involving both static and dynamic interaction" sounds awkward because of the repetition. It is also partly unsupported and somewhat arbitrary. What exactly do the author mean when classifying interactions as static or dynamic? I guess that a quantitative perspective on this would require measuring dissociation rates for each individual interaction in the multipartite interface, and this would not even be informative of how the same interface would behave as part of a greater interface. Second, the sentence "Perturbing the CPC-nucleosome interaction compromises...the dynamic centromere association of CPC..." falls short of representing the authors' observations, as the perturbation they refer to appears to make the CPC even more dynamic. Finally, the final sentence on the non-catalytic role is somewhat cryptic, it has not been clarified anywhere else in the abstract, however briefly, what is meant by this (I assume they refer to the stabilization of centromeric chromatin). It is also at odds with the previous remark that "whether CPC has any non-catalytic role at centromere" is an open question, as this last sentence seems instead to imply it as an established fact this role exists.

-Introduction, first and third paragraphs: The CPC occupies at least two separate locations within the centromere, including one

that is kinetochore proximal and one that is more "central", and that reflect interactions with BUB1 and Haspin substrates, respectively. There is almost no evidence that the SAC is controlled from the centromere, and there is very minor evidence that biorientation is controlled from the centromere, and therefore neither contribution merits the adjective "essential". Furthermore, depletion of the CPC has no effect on sister chromatid cohesion either, and only Sgo1 depletion or mutation, but not its mislocalisation, affect cohesin. This is not to say that the problem studied here is not important (I believe it is), but I am not convinced that the authors' arguments to justify its importance are entirely based on established facts.

-Introduction, fourth paragraph: "...Borealin-mediated multivalent interactions..." It is unclear if the authors are referring to the multipartite interaction mode described in the Abstract, or to the purported dimerization of Borealin. A note on this below.

-Figure 1, panels B and C: labels for the N- and C-termini and approximate positions in the overall view, i.e. not only in the inset, would facilitate the interpretation of these panels.

-Results: "This observation suggests that multivalent and dynamic interactions involving different regions of Borealin and INCENP, not involving the nucleosome acidic patch, are essential for CPC-NCP binding (Abad et al, 2019; Serena et al, 2020). These dynamic interactions, likely involving protein-protein and protein-DNA contacts, may facilitate high-affinity binding of CPC to nucleosomes." This sentence lacks clarity. First, I would recommend defining multivalent (throughout the manuscript). It is not synonymous to multipartite, the term used in the abstract. If used as synonym, please clarify it in the text, but I would limit 'multivalent' to conditions created by oligomerization through identical or strictly similar modules. I feel that 'multipartite' is perfect for the case described here. Second, it may not be obvious to many readers that by 'dynamic' they imply that it is not observed in the density maps. If that is what justified the use of 'dynamic', it should be stated clearly. Third, the authors have not shown that the dynamic interactions they refer to are essential, but rather that they are sufficient in the absence of interactions from the Borealin N-terminal tail. Fourth, the PIs should clarify why they don't think that the interaction with the phosphorylated N-terminal tail of H3 is sufficient for the residual binding they observe when the interaction at the acidic patch is eliminated.

-Figure 3B: Shouldn't the profile with the longer Borealin construct be also shown? Is any loss of affinity evident?

-Figure 4A: Apologies but I seem to have missed the point: are these Coomassie-stained gels? What is the cause of the high background in the uncrosslinked samples? The inputs seem very clean, is it background crosslinking in the absence of UV activation? And on what bands are the ratios shown in the histograms calculated?

-Results: How do the authors arrive to the conclusion that the Borealin Δ loop construct has a 3-fold decrease in affinity? Is this based on the gels in Figures

-Results: "Altogether, our structural and biochemical analysis shows that CPC-NCP binding is mediated by multi-partite interactions involving both stable and dynamic interactions. This, along with our observation that CPC engages with the nucleosome DNA entry-exit site on both faces of the NCP suggests that CPC likely stabilizes chromatin by protecting the wrapped state of nucleosomal DNA." This refers again to the somewhat arbitrary and rather unconvincing definition of stable and dynamic. If they want to use this distinction, could they at least indicate which interactions are dynamic and which static, and what is the evidence support this distinction?

-Figure 6B: very hard to take these data to support the authors' conclusion of protection in vivo.

Minor points

-For future reference, adding page and line numbers to the text and figure numbers to the figures facilitates the reviewers' job!

-Introduction: "...what CPC's potential..." please check the grammar of this sentence.

-In the Bishop & Schuniacher 2002 reference the correct author's name is Schumacher.

-Results, first two lines: As the authors indicate that the loop region encompasses residues 110-206, they could do so for the N-terminal tail too. Else, they could simply refer to S1 for both.

-Results: there is a call to Figure 1C but no panel 1C in figure 1, as far as I can tell.

-Results, lines 5-6: "However, how the different...remains an open question". This has already been said in the Introduction, so I would simply connect to the next sentence: "To address how different nucleosome binding elements of CPC collectively contribute towards CPC-nucleosome binding and chromosome association, we purified recombinant..."

-"To contact" is transitive, i.e. you don't contact with residue X, you contact residue X

-"We also observed a comparable decrease in DNA protection when we perturb..." 'perturbed' would be more appropriate.

Referee #3:

Gireesh et al. present insightful analyses of the chromosomal passenger complex (CPC), proposing that CPC-nucleosome interactions contribute to centromeric chromatin protection independently of the kinase activity of Aurora B, the catalytic subunit of the CPC. Combining high-resolution cryo-EM, crosslinking mass spectrometry, biochemical reconstitution, AFM, and cell-based chromatin assays, the authors provide evidence that the CPC engages nucleosomes through both stable (H2A/H2B acidic patch binding) and dynamic (DNA/histone tail interactions), and that these interactions all contribute to "protection" of chromatin - measured by openness of the nucleosomes. Mutations that compromise the CPC-DNA interaction leads to mitotic delays, supporting the importance of the non-catalytic role of the CPC on the centromere integrity, though it does not rule out the possibility that the defect is caused by compromised centromeric enrichment of Aurora B.

The manuscript is well executed, and the major conclusions are mostly supported by the presented data, except for the difficulty in demonstrating the physiological importance of non-catalytic functionality of the CPC in centromeric chromatin protection. The only mutant that the authors tested its functional importance in cells was INCENP-6A mutant, where INCENP basic residues that are supposed to interact with DNA was mutated. Unfortunately, these INCENP basic residues were not visualized in the presented cryo-E structure. The structure revealed a novel interaction between Borealin N-terminus and the acidic patch of the nucleus, but physiological importance of this interaction was not tested. Additionally, the structural analysis workflow is not explicitly organized or sufficiently detailed, particularly in terms of data processing rationale, which limits the interpretability of the cryo-EM findings. Overall, the manuscript presents compelling evidence that the CPC can limit chromatin accessibility in a manner independent of the catalytic activity of Aurora B *in vitro*. This is an important advancement to understand the mechanism by which the CPC engages and potentially modulates the centromeric chromatin architecture, while demonstrating its functional importance *in vivo* is challenging.

Major points:

1. Table 1. Please include the resolution of the refined atomic model based on the FSC criterion, in addition to the existing map resolution metrics. This information is essential for evaluating the quality and interpretability of the structural model.
2. Fig. S2. The data processing workflow should be expanded to include greater detail. Specifically, please indicate the symmetry applied during 3D reconstruction and show a representative cryo-EM micrograph. Notably, the final map was generated using 73,078 particles from 12,436 micrographs, suggesting that the majority of particles were discarded. The workflow should address the rationale for this, such as potential structural flexibility of the CPC-nucleosome complex, and detail how particles were classified and refined. This is important to get a sense of whether the configuration where two CPC molecules interact with a nucleosome represents a minor fraction of the reconstituted molecules. The stoichiometry of the CPC and the nucleosome may be addressed by the SEC (Fig. 3B).
3. Fig. S2c. Please show the structure in a different viewing angle. It is not clear if another CPC molecule associates with the nucleosome on the other side.
4. Fig. 1B. In this presentation, it is impossible to evaluate the amino residue assignments and the side-chain interactions. The figure can be split at least in two, one presenting an atomic model to illustrate the side-chain interactions and the other presenting the map-overlaid model to justify the side chain assignments. Such an example can be seen in Fig. S7 of PMID 39088653.
5. Figure 5E. While the authors report that deleting the Borealin N-terminus that is important for acid patch interaction reduced nucleosome binding, this is only seen for the 601 nucleosome but not for the alpha-satellite nucleosome. Please discuss the implications of this result given that the acidic patch mutant does not impact the NCP binding of the CPC in Fig 3A.
6. Many of the bar graphs do not define the sample numbers and the error bars (Fig. 4, Fig. 5, Fig. 6). It would be recommended to plot each data point and show data distribution.
7. Figure 4A, left panel. I cannot see the band that representing the interaction between CPC I1-190SB and CPC I1-190-6ASB. It seems odd that the clear band is much clearer seen with Δ loop, which is supposed to decrease the CPC-nucleosome interaction, according to Fig. 4B
8. Figure 4B and C. Please indicate the concentration of the CPC used. It is not clear which lane represents the 160 nM condition analyzed in the right panels. Please describe how the quantitation was done, and what is the basis for "a 3-fold decrease in affinity" (page 7).
9. Figure 5C. It would be informative if the distribution is reproducible between the two independent experiments. In methods, please describe how the CPC was added to the nucleosome. Close examinations of the AFM images indicate that several nucleosomes do not have clear linker DNAs, making it difficult to analyze their linker DNA angles. How did you analyze them? I can also see many fragmented DNAs, indicating that the estimated wrapped lengths could be affected by DNA fragmentation. It would be informative to measure lengths of naked DNA in each condition. If the CPC treatment artificially induces DNA

fragmentation, you may see more fragmented DNA. Conversely, if the CPC indeed protects the nucleosome, you may see less fragmented DNA. In any case, such analysis will tell you if the apparent differences between the experimental conditions are not due to technical issues related to the exposure of the nucleosome to the mica.

10. Figure 6. Although the title of the figure legends state, "CPC-mediated protection of centromeric chromatin is crucial for accurate chromosome segregation", chromosome segregation accuracy was not reported in this figure. From the represented images shown in D, it seems that chromosome segregation was normal in 6A mutant. I assume that the authors must have data to quantitatively analyze chromosome segregation errors. It would be also informative to report the duration between NEBD to metaphase alignment. In any case, from this experiment, it would be difficult to attribute the phenotype to the defect in CPC-mediated protection of centromeric chromatin, since the amount of INCENP on the centromere was reduced.

11. Since condensin depletion is known to mitotic delay through compromising the mechanical tension applied to the centromeric chromatin (PMID 19188492), it would be informative to measure the inter-kinetochore distance in 6A cells to see if 6A compromises chromatin integrity at the centromere.

12. Figure 6B. This is the only data to show that INCENP 6A mutant increases centromeric chromatin accessibility. However, the effect is subtle. It would be important to make quantitative analysis with multiple experiments to support the non-catalytic role of the CPC on chromatin protection. Ideally, an additional experiment with a different methodology is advised, to support the claim that the CPC protects centromeric chromatin. Without solidifying this part, the conclusion, described as a headline in page 8, "CPC-nucleosome interaction is essential for centromeric chromatin protection", must be softened.

13. Discussion can be expanded, for example, by including the potential implications of the acidic patch's role in the CPC binding, and the mechanism by which 6A mutant causes the metaphase delay. The possibility that the effect of 6A mutant is indirect must be discussed.

Minor points:

1. Please include the page numbers, and figure numbers in the figure display. Unless it is instructed by the journal, I prefer to have figure legends associated with the figures. Since figure legends are placed after Discussion and Methods, it was cumbersome to relate the main text, figure legends, and the figure in the current arrangement.

2. Please clearly explain the acronyms for the constructs, such as CPC I1-190_6ASB and INCENP 6A.

3. Fig 1A. By this presentation, the readers may think that the structure of the entire colored segments are solved and shown in B, although only the portions of each subunit were resolved in B. It will be helpful to modify the diagram to clearly indicate the segments that were solved by the structure so that readers can readily understand which segments support the stable and dynamic interactions with the nucleosome.

4. Fig S5D. Please specify the AlphaFold prediction version and plot the confidence of prediction as well.

5. Fig. 3C, right panel. The labeling text overlaps with the DNA backbone, making the annotation difficult to read and visually cluttered. To improve clarity, please adjust the placement, size, or color contrast of the text accordingly.

6. Fig.S5D. Please label the position of H2A T120 accordingly to show how it interacts with the CPC tri-helical bundle.

8. Typos.

Page 1. Phosphorylations -> Phosphorylation

Page 2. Spindle Assembly Checkpoint -> the spindle assembly checkpoint

Numerous places. CPC -> the CPC

Referee #1:

The study focuses on the 2.86 Å resolution cryoEM structure of the chromosome passenger complex (CPC) in complex with the nucleosome core particle (NCP). The structure shows the N-terminal part of the borealin interacting with the acidic patch of the nucleosomes. This is consistent with the study published by the authors in 2019 (Abad *et al.*, JCB). However, when the authors either mutate the residues of the acidic patch in the NCP or remove the N-terminal part of the borealin, they see that CPC and NCP still form the complex (readout: native gel and co-elution in SEC). From this, they conclude that other, more "dynamic" interactions (not visible in their structure) are important for the interaction between the two entities. To prove this, they performed two types of cross-linking and concluded that it must be the positively charged 'RRKKRR' motif in INCENP that interacts with histones, and the borealin loop that interacts with DNA. They also perform MNase digestion on mononucleosomes and 12-mer nucleosome arrays in the presence and absence of CPC and conclude that CPC protects NCP from MNase digestion. They use AFM to determine that CPC reduces the opening angle of DNA on NCP. Finally, they perform FRAP and MNase digestion of cell chromatin and conclude that the 'RRKKRR' motif in INCENP confers more stability to centromeric chromatin in cells.

The study addresses a highly relevant topic and provides novel structural insights into the interaction between the CPC and the nucleosome. However, there is a lack of satisfactory correlation between the proposed structural model and the functional experiments intended to validate it. For instance, the Cryo-EM data indicate that the most stable and significant interactions occur between the nucleosome's acidic patch and the N-terminal region of Borealin (specifically, the first 10 amino acids). Although the authors generated a mutant lacking these first 10 amino acids, this mutation does not appear to disrupt nucleosome binding. This finding stands in stark contrast to previous results from the same laboratory (Abad *et al.*, 2019), where the identical mutation impaired nucleosome binding.

To strengthen the study, it is crucial to quantitatively assess the effects of mutations on the interacting surfaces. More rigorous binding experiments would enhance the quality of the work and provide a clearer understanding of the individual contributions of various interactions to CPC-nucleosome stability.

We thank the reviewer for their constructive suggestions. As outlined below, we have now addressed the concerns by performing additional experiments to quantitatively assess the contribution of different interaction interfaces and revising original figures describing the structure and cryoEM workflow to improve overall quality, along with providing additional new figures. We would like to stress that the data presented here on the contribution of the N-terminal tail of Borealin for NCP binding is not in contradiction with our previous report (Abad *et al.*, 2019), as outlined in our response to this reviewer's concern no. 11.

More detail comments to help the authors elevate the quality of the manuscript are below:

(1)- The introduction is unnecessarily long. It should focus on the mayor question that the experiments are addressing.

We respectfully disagree with the reviewer. We do not believe the introduction is overly long; however, we have attempted to shorten it without compromising the necessary background information that is essential for fully appreciating our findings. We have removed the following sentence in the first paragraph of the introduction: 'In humans and other primates, centromeres are also characterized by the presence of highly repetitive DNA sequences known as α -satellite (Manuelidis, 1978; Fukagawa & Earnshaw, 2014)'.

(2)- Pg 5. "This mode of anchoring facilitates the engagement of the downstream triple helical bundle formed by Borealin, Survivin and INCENP with the nucleosome DNA entry/exit site (Fig 1C)."

o From the Figure 1, it is not clear how is nucleosome positioned. Authors should label the dyad and the entry/exit site and include the zoom in on the entry/exit site showing interactions with CPC.

We have now amended the suggested changes to Figure 1 by labelling the dyad axis and the DNA entry-exit site. We have also included the zoom in image of the DNA entry-exit site clearly showing the interactions with the Borealin helix. Please find the edited Figure 1 below.

the supplementary file, Fig. EV5A, we have now included two panels depicting the Borealin density contacting the nucleosome acidic patch and the DNA entry-exit site, as shown below.

A

(3)- Pg 5. "3D class analysis of a population of particles with well-defined CPC densities revealed three discrete conformational states of the CPC triple helical bundle"

o There is no detailed particle selection protocol in the supplementary figures, explaining particle picking and refinement protocol including number of particles in each class.

We thank the reviewer for this suggestion. We have now prepared Fig. EV2, and Fig. EV3 (also in response to #5) with a detailed workflow for the cryoEM densities (please find the workflow below).

(4)- Pg5. "This, along with 3D variability analysis, shows that CPC, with the Borealin N-terminal region tethered at the nucleosome acidic patch, swings both vertically and horizontally (Fig. 2A, Movie1)"

o Movie1 is just showing different contours of the map in different orientations. It does not demonstrate swinging of the CPC.

We would like to reiterate that movie 1 was included to highlight the swinging motion of the CPC. The movie was created using the volume series output of a 3D variability analysis carried out in cryoSPARC. We acknowledge that the reviewer might not have appreciated the swinging motion of

the CPC from the movie we included, likely due to the orientation we chose to represent was not ideal. We have now updated this movie with an orientation that clearly shows the swinging motion.

(5)- Pg 5. "Removing the signal of bound CPC using particle subtraction and subsequent refinement revealed a 2nd copy of the CPC occupying the symmetrically equivalent second face of the nucleosome in a similar conformation (Fig. 2B)"

o What kind of "subsequent refinement" was used here to make other CPC "appear"? In the composite map the orientation and the intensity of the "other" CPC is almost the same as of the original one. If this is the case, why was it the other molecule not visible before the subtraction? What is the percentage of particles that have two CPCs bound?

We thank the reviewer for their critical assessment of the map and the concerns raised. Throughout the data processing stages, we consistently observed a clearly defined map for the bound Borealin N-terminal tail on both acidic patches of the NCP (on both NCP faces). However, the triple helical bundle of CPC is better resolved on only one of the two faces, as shown below in the 3D reconstruction used for structural analysis in Fig. EV4E.

We reasoned that due to the flexible and possibly anti-correlated swinging motion of the CPC, we could not obtain a volume with equally well-defined densities for the CPC triple helical bundles on both NCP faces in a single cryoEM reconstruction. To visualise the 'poorly resolved' second CPC, we attempted to subtract the 'well-defined' CPC density and performed 3D reconstruction and refinement using the subtracted particles. This resulted in a 'near-identical' well-defined density for the second CPC on the opposite face. The 'original' map and the map generated using 'subtracted particle' were combined in ChimeraX to generate the composite map (shown in original Fig. 2B). However, following the reviewer's concern, we critically assessed each step of the workflow, which led us to identify a technical oversight during the particle subtraction step (lack of mask flipping resulting in incomplete particle subtraction) as the cause of generating a near-identical map for the symmetrically related second CPC. We sincerely apologise for this oversight.

In order to resolve the CPC density on both faces of the nucleosome, the particles from the three well-defined classes were taken to RELION, and 3D Refine with Blush Regularisation was performed. This yielded a 3.8 Å map for the particles belonging to Class 2 with clearly defined densities for both CPCs. We have now updated Figure 2B with the corresponding map.

As

B

an

alternative approach, in cryoSPARC, we performed a focused 3D classification run with a mask covering CPC density on both faces on ‘nucleosome-subtracted’ particles. Particles with apparent CPC density on both faces were analysed using additional 3D classification runs to enrich particles contributing to volume reconstruction with ‘well-defined’ density for both CPCs (~7 % of total particles). We have now updated Figure EV4F with the corresponding map. This figure, along with the detailed workflow for the reconstruction, has been included in the revised manuscript as Fig. EV3 and EV4F. We would like to highlight that this change does not change any of the conclusions of the original manuscript.

Fig. EV3

Fig. EV4F

(6)- Pg 5. "A detailed analysis of the intermolecular interactions at the NCP acidic patch showed that Borealin N-terminal tail amino acid residues Arg 4 and Arg 9 form the arginine anchor that interacts with the H2A/H2B Asp/Glu residues of the nucleosome acidic patch, through salt bridge interactions (Fig. 1B)."

o In my opinion, this is the most important part of the results from the structure that the authors obtained and the figure is not convincingly showing that they have trustable density to conclude about those interactions. They should show a clear density around N-terminal tail and then zoom on the two Arginines in Borealin showing their connection with the side chains of the nucleosome.

We have now created new figure panels to show the quality of the map density corresponding to the N-terminal tail of Borealin, along with zoomed-in views of the two Arginines of Borealin, highlighting the interactions with the nucleosome acidic patch residues. These panels have been included in the supplementary figures, Fig. EV5A.

(7)- Pg 5. "The CPC11-190SB-H3T3ph NCP structure also revealed that the Borealin helix facing the DNA entry-exit site is highly basic with residues Lys 26, Arg 30, Lys 37 and Lys 48, among which Arg 30 makes Van der Waals contacts with the phosphate backbone of the DNA between super-helical location (SHL) 6 and 7 (Fig. 1C)."

o This is also not visible from the figure. Each statement like this has been backed with a clear figure showing both (density; could be in a supplement) and side chains of the amino acids making interactions.

As mentioned in the response to point 2 of this reviewer, we have now amended Figure 1 to depict the Borealin helix-DNA interaction clearly. We have also created the corresponding images showing the Borealin density, which have been added to Fig. EV5A (also attached in response to point 2).

(8)- Pg 5. "Additionally, Gln 44 of Borealin makes H-bonding interactions with the backbone phosphates of nucleotide at the DNA entry-exit site (Fig. 1C)."

o Again, this is not clearly visible in the figure. Additionally, in the provided pdb model Glu 40 is modelled close to the DNA while Gln 44 is not within the salt bonding distance.

As mentioned in the response to point 2 of this reviewer, we have now amended Figure 1 to include these changes.

In the original version of the manuscript, we included an atomic model only for the 'Class 0' map. Since submitting the manuscript to the journal, we have now built atomic models also for Class 1 and Class 2 maps to assess contacts. In all models, Borealin residues K26/R30/K37/E40/Q44 face the DNA entry-exit site, with variable quality of side chain densities (in line with the local resolution of the CPC triple helical bundle, 3.5-4 Å for Class 2). As suggested by the reviewer, we have updated Figure 1 to clearly show the interacting residues. In the revised model, the Q44 side chain is in H-bonding distance with the phosphate backbone of the Deoxyguanine at position 149 (see the updated Figure 1 in response to point 2 of the reviewer). We have now assessed the contribution of these Borealin residues for NCP binding and MNase protection activity and included the data in the revised manuscript, Fig. 3C-E (EMSA), Fig. EV6A and B (SPR) and Fig. 5E and F, EV8C and D (MNase).

Fig 3C-E

Fig. EV6A and EV6B

Fig. 5E and 5F

Fig. EV8C and D

(9)- Pg. 5. "Dynamic interactions are required for efficient CPC-nucleosome binding".

The authors repeatedly use the term dynamic interaction. All non-covalent interactions - whether protein-protein or protein-ligand - are inherently dynamic, with varying degrees of strength and specificity. Authors should not use term dynamic interactions to refer specifically to interactions with low affinity, low specificity, or otherwise weak interactions.

We appreciate that our usage of the terms 'static' and 'dynamic' to describe the 'stable' (observed in the cryoEM structure) and 'conformationally heterogeneous' (not observed in cryoEM structure, but contributes to CPC-NCP binding based on biochemical and cellular studies) nature of interactions might lead to confusion. In the revised manuscript, we have removed/replaced the term 'dynamic' with 'conformationally heterogeneous' throughout the text.

(10)- Fig. 4. Band C. And text on pg.7 "These observations suggest that while Borealin loop region contributes to CPC-nucleosome affinity, the dynamic contacts involving INCENP 'RRKKRR' might contribute to the dynamics of CPC-NCP binding."

o How does INCENP RRKRR contribute to the dynamics of the CPC-NCP binding? It seems difficult to provide an explanation if mutations of the positive INCENP patch have no measurable effect on binding.

We have now quantitatively assessed the binding affinities of INCENP wt and INCENP RRKRR mutant using EMSAs (Fig. 4C, D and E). The data show that while the INCENP RRKRR mutant (INCENP 6A) binds to H3T3ph Widom 601 NCPs with similar binding affinity compared to the INCENP wt, in the case of H3T3ph α -satellite NCPs, INCENP 6A binds moderately (~ 1.5 times) weaker to NCPs compared to INCENP wt and the contribution of the RRKRR motif is even stronger when combined with the Borealin Δ loop mutant (Fig. 4C, D and E). This data suggests that the contribution of INCENP to nucleosomal DNA binding might be DNA sequence-dependent. We have now amended the text as follows in page 8 (2nd paragraph): “To directly assess the contribution of INCENP ‘RRKRR’ and the Borealin loop region for nucleosome binding, we performed EMSA assays with different versions of CPC containing either INCENP with ‘RRKRR’ mutated to ‘AAAAA’ (CPC_{I1-190} 6ASB) or Borealin lacking the loop (CPC_{I1-190SB} Δ loop) and H3T3ph nucleosomes reconstituted with either Widom 601 or centromeric α -satellite DNA (Fig. 4C-E and EV5B and C). EMSA analysis showed that while CPC_{I1-190} 6ASB bound to Widom 601 NCPs with a similar affinity as the wild type (wt) CPC complex (CPC_{I1-190SB}), the CPC_{I1-190SB} Δ loop showed a 12-fold decrease in affinity compared to CPC_{I1-190SB} (Fig. 4C-E). Interestingly, the INCENP ‘RRKRR’ mutant (CPC_{I1-190} 6ASB) bound weaker (though moderately, ~ 1.5 times) to α -satellite NCPs compared to CPC_{I1-190SB}. However, the contribution of the INCENP ‘RRKRR’ motif to α -satellite NCP-binding was even stronger (~ 13 times) when combined with the Borealin mutant lacking the loop region (CPC_{I1-190} 6ASB Δ loop). These observations suggest that both the Borealin loop region and the INCENP ‘RRKRR’ motif contribute to CPC-nucleosome affinity, and, in agreement with Serena et al, 2020, that the contribution of INCENP to nucleosome binding might be DNA sequence-dependent.”

Please find the Fig.4 C-E below:

(11)- Pg 6. "Furthermore, we purified a CPC complex lacking the N-terminal 10 residues of Borealin (CPC I1-190SB10-end) and tested its ability to interact with H3T3ph NCPs using size exclusion chromatography (SEC) (Fig. 3B). SEC analysis shows that CPC I1-190SB10-end can form a complex with H3T3ph NCPs (Fig. 3B)."

o Interestingly, in their previous work (Abad *et al*, JCB, 2019), the authors used CPC with borealin (10-109) and found a complete absence of binding in both EMSA and SPR, which contrasts with CPC with borealin (1-109). From this they conclude that the first 10 amino acids of borealin are essential for binding. Why do the authors not comment on the contradictory results they have now obtained? Again, a more quantitative measurement of the interaction would be required. Otherwise, the reported structure is almost useless because it leaves the reader with the impression that mutation of residues essential for binding (either on NCP or CPC) does not affect binding.

We would like to emphasise that the observations we reported previously (Abad *et al.*, JCB 2019) and the data presented here are not contradictory. In Abad *et al.*, JCB 2019, we showed that removing the N-terminal tail of Borealin, while reduced the CPC-NCP binding affinity, did not abolish it. We have now quantitatively assessed the contribution of the Borealin N-terminus in the context of CPC_{11-190SB} (CPC containing Borealin lacking N-terminal tail (Borealin ΔN), Survivin full length and INCENP 1-190 (longer INCENP that includes the RRKKRR motif)), both using EMSAs and Surface Plasmon resonance (SPR). Both EMSAs and SPR show that deletion of the N-terminus of Borealin weakens the interaction with H3T3ph Widom 601 NCPs approximately 5-fold. Please find Fig. 3C-E (EMSA) and EV6A and B (SPR) below:

Fig. 3C-E

Fig. EV6A and B

And added the following text in the revised manuscript on page 6 and 7: “We then quantitatively evaluated the contribution of Borealin N-terminal tail – NCP acidic patch (CPC_{11-190SBΔN}, CPC_{11-190SBΔN K12A}) and the Borealin helix – DNA entry-exit site (Borealin K26/R30/K37A) interactions, separately and in combination (CPC_{11-190SB3A} and CPC_{11-190SBΔN K12A 3A}), for CPC-NCP binding by performing EMSAs and Surface Plasmon Resonance (SPR) (Fig. EV5B and C). Our data consistently show that the CPCs containing the Borealin N-terminal tail mutants (CPC_{11-190SBΔN} and CPC_{11-190SBΔN K12A}) bind NCP relatively more weakly compared to CPC with Borealin helix mutant (CPC_{11-190SB3A}) (Fig. 3C-E and EV6A and B). However, none of these Borealin mutants, either in isolation or in combination, abolished CPC-NCP binding.

Altogether, these observations suggested that in addition to the CPC-NCP interactions resolved in the cryoEM structure, previously well characterized phosphorylated Histone H3 tail interaction with Survivin (Abad et al, 2019) and conformationally heterogeneous interactions involving different regions of Borealin and INCENP may also contribute to efficient CPC-NCP binding.”

(12)- Pg 6. "To assess the contribution of dynamic interactions crucial for CPC-nucleosome binding, we performed crosslinking/MS experiments with the CPC11-190SB -H3T3ph NCP complex (Fig. 3C and Fig. S3A)."

o How far are residues that this crosslinker connects and what is the chemistry behind? This is not explained neither in the text nor in the M&M.

The crosslinker used is the 1-ethyl-3-(3-dimethylaminopropyl) carbodiimide (EDC). EDC is a zero-length chemical crosslinker that covalently links primary amines of lysine and the protein N-terminus, and to a lesser extent hydroxyl groups of serine, threonine and tyrosine, with carboxyl groups of aspartate or glutamate. We apologise for the oversight. We have now included this information in the main text (page 7 line 209 and 211) and the Materials and Methods section (page 37, line 900 of the revised MS).

o What is the conclusion here? Can these interactions be modeled?

From the EDC crosslinking/MS data, we conclude that in addition to the contacts we observe in our CryoEM structure between Borealin and the nucleosome acidic patch and DNA entry-exit site, there are further interactions involving the Borealin loop and the INCENP basic IDR with the nucleosome hotspots. Our efforts to model these interactions through molecular dynamics (MD) simulations using the crosslinking data and interactions observed in the CryoEM map as constraints proved technically challenging. While we continue to pursue this, due to time constraints, including such analysis is beyond the scope of this manuscript.

(13) - Pg.6. "UV-crosslinking of CPC-NCP complex, followed by SDS-PAGE analysis of the nucleosomal DNA revealed the presence of DNA crosslinked with Borealin migrating at the expected molecular weight (Fig. 4A and S3B)."

o Again what is chemical nature of this cross-linking? What is the experiment aiming to resolve?

The UV crosslinking experiment was performed to determine which CPC subunits contribute to NCP binding through direct nucleosomal DNA interactions. We have now performed new UV crosslinking experiments that have allowed us to map the regions of Borealin and INCENP that interact with nucleosomal DNA. We have added the new data to the manuscript. Please find Fig. 4B below:

And we have also added the following paragraph to the main text (p. 7): "To understand how CPC is interacting with nucleosomal DNA, we stabilized protein-DNA interactions within the CPC-NCP complex through UV crosslinking and used MS analysis (Stützer et al, 2020) to map the protein-DNA interaction sites. MS analysis identified DNA crosslinked peptides for both Borealin and INCENP (Fig. 4B and EV6D). All the DNA-crosslinked peptides of Borealin map to a region spanning the end of the Borealin N-terminal α -helix until the N-terminal half of the Borealin loop region. This is consistent with our previous SPR data, which suggested Borealin amino acid residues 110 to 188 as a region capable of directly binding DNA (Abad et al, 2019). The DNA-protein contacts in INCENP spanned not only IDRs, but also the N-terminal α -helix, which is part of the triple helical bundle formed by Borealin, Survivin and INCENP, suggesting the capability of this helical bundle in making transient inter-nucleosomal contacts via INCENP under favourable conditions (Fig. 4B and EV6D). Our analysis also identified a DNA-crosslinked peptide spanning residues 71-83, which is adjacent to the 'RRKKRR' motif, suggesting the capability of this region to interact with DNA (although we could not identify

any DNA-crosslinked peptides involving the 'RRKKRR' motif, possibly due to the presence of a large number of trypsin cleavage sites). Overall, this data suggests that DNA interactions involving both Borealin and INCENP also contribute to CPC-nucleosome binding."

We have also included the following statement in the Materials and Methods section (page 39 the revised MS) to clarify how we performed the UV crosslinking: "UV crosslinking uses ultraviolet light (UV 254 nm) to form zero-length covalent links between nucleic acids and protein".

(14) - Figure 5F. What are numbers 1, 2 in the figure?

We apologise that this was not clear. 1 and 2 denote the time of MNase treatment in minutes. We have now changed this figure and only used one time point (2 min), please find the new Fig. 5F below:

(15)- Pg. 8. "Consistent with our mononucleosome MNase data, all Borealin and INCENP mutants showed reduced MNase protection activity (Fig 5F)."

o Is the conclusion that all different unstructured parts are interacting with DNA and are thus providing stability against MNase?

Our new UV crosslinking analysis reveal that both Borealin and INCENP make contacts with DNA predominantly via their IDRs, although, as discussed in the revised MS, page number 7, the INCENP helix, which is part of a triple helical bundle formed by Borealin, Survivin, and INCENP, also interacts with DNA. This observation suggests that while the IDRs of Borealin and INCENP likely mainly contribute to protection against MNase activity, the structure regions may also contribute to the protection via inter-nucleosomal contacts.

(16)- Pg. 10. "We previously showed that Borealin dimerization domain contributes to efficient nucleosome binding, but how it does so remained unclear (Abad *et al*, 2019). Our observation that two copies of CPC engage with both faces of the nucleosome (related by a two-fold symmetry) suggests the potential contribution of Borealin dimerization in facilitating this interaction."

o Apart from not really having any solid evidence that nucleosomes are binding 2 CPC molecules at the same time ("composite map" is at best "sketchy"), the nucleosome, due to its symmetry, has two identical faces that can theoretically accommodate the same type of ligand. So even if nucleosome would have two copies of the CPC, one on each face, that would not imply that these two are connected by the dimerization domain. They could be binding each in their monomeric form, thus such speculation does not have any grounds.

The mass photometry analysis of the CPC-NCP complex sample (containing a molar excess of CPC) subjected to cryoEM analysis showed a major species with a molecular weight corresponding to 2 CPCs bound to 1 NCP (340 ± 16.9 kDa; theoretically calculated 337.95 kDa). Please find new Figure EV1D below:

In addition, as discussed on Page 5 of the revised manuscript, we have also analysed the subunit stoichiometry of CPC:NCP complex containing a Borealin version harbouring a mutation within the dimerization domain, which resulted mainly in a 1:1 CPC:NCP complex, suggesting that Borealin dimerization stabilises the 2:1 CPC:NCP complex. The revised text is as follows (page 5): “Consistent with this, mass photometry analysis of the CPC-NCP complex revealed the presence of a species with molecular weight corresponding to two copies of CPC bound to one NCP (312 ± 34 kDa, 27%; theoretical 308.3 kDa), although relatively a smaller population 1:1 CPC:NCP complex was also present (215 ± 18.6 kDa, 8%; theoretical 253.8 kDa) (Fig. 2C). In contrast, the CPC-NCP complex containing a Borealin mutant where Ser 266 located within the dimerization domain was mutated to Asp (S266D) (Fig. 2C) revealed predominantly a 1:1 CPC:NCP complex (256 ± 27 kDa, 17%; theoretical 253.8 kDa) along with a much smaller fraction of a 2:1 complex (309 ± 16.8 kDa, 4%; theoretical 308.3 kDa) (Fig. 2C). Based on this observation along with the cryoEM structure, we reason that Borealin mediated dimerization of CPC facilitates the binding of CPC on both faces of the NCP.”

Please find the new Fig. 2C below:

(17)- Pg.11. "Note: A recent study on biorxiv, Ruza *et al*, 2025, reports a cryo-EM structure of CPC bound H3T3ph nucleosome. While the mode of nucleosome binding by CPC broadly agrees with observations reported here, their structure shows just one copy of CPC bound to nucleosome, possibly because their CPC construct lacks more than two-thirds of the C-terminal region of Borealin (including the Borealin loop and dimerization domain)."

o If authors want to comment on the complementary study on biorXiv (and I think they should) then they should do that in the systematic way and throughout the manuscript.

We have now included a discussion on the Ruza *et al.*, work in our 'Discussion' section.

Referee #2:

This interesting manuscript by Gireesh and colleagues reports a study of the interaction of the localization module of the chromosome passenger complex (CPC) with an H3- nucleosome. The manuscript includes a cryo-EM characterization of the complex of the so-called localization module of the CPC with an H3-modified nucleosome (containing histone H3 pre-phosphorylated on threonine 3 and obtained with chemical ligation), as well as further work of biochemical and biological characterization of the interaction. The best-defined feature of the structure demonstrated binding of the N-terminal tail of Borealin to the acidic patch of the nucleosome, whereas the interaction of the H3 tail with Survivin was invisible. The Borealin N-terminal tail, however, was not necessary for the interaction, and the authors focused on other "dynamic" interaction patches, also invisible in the structure but important for overall stability of centromere localization of the CPC. Special emphasis here was given to a 'RRKKRR' sequence on INCENP previously shown by the Barr and Grunerberg laboratories to be important for CPC localization. The authors combined DNA crosslinking experiments and chromatin imaging and protection assays to conclude that the CPC has a non-catalytic role in the stabilization of centromeric chromatin. They also provide evidence that the RRKKRR' motif stabilizes the interaction of the CPC with the centromere, with relatively minor consequences for mitotic progression and a more significant effect on central spindle localization after anaphase. With nuances, this manuscript reaches conclusions that are related to, and consistent with, those reported in a manuscript by Ruza and co-workers (Barr and Gruneberg laboratories) currently in the bioRxiv.

In general, I feel that this manuscript addresses an important topic from multiple angles, providing a very good basis for further work in this complex area of research. The work is technically very well done. The presentation requires more clarity on some key points, as detailed below. I am therefore supportive towards publication, and would kindly ask the authors to consider the following points:

Specific points

(1)-The first part of the Abstract is well written, but the second could be improved. Three claims may require the authors' attention. First, the sentence "CPC employs multipartite interactions involving both static and dynamic interaction" sounds awkward because of the repetition. It is also partly unsupported and somewhat arbitrary. What exactly do the author mean when classifying interactions as static or dynamic? I guess that a quantitative perspective on this would require measuring dissociation rates for each individual interaction in the multipartite interface, and this would not even be informative of how the same interface would behave as part of a greater interface. Second, the sentence "Perturbing the CPC-nucleosome interaction compromises...the dynamic centromere association of CPC..." falls short of representing the authors' observations, as the perturbation they refer to appears to make the CPC even more dynamic. Finally, the final

sentence on the non-catalytic role is somewhat cryptic, it has not been clarified anywhere else in the abstract, however briefly, what is meant by this (I assume they refer to the stabilization of centromeric chromatin). It is also at odds with the previous remark that "whether CPC has any non-catalytic role at centromere" is an open question, as this last sentence seems instead to imply it as an established fact this role exists.

We appreciate this reviewer's suggestion. We have altered the abstract where appropriate to improve the overall clarity:

"The chromosomal passenger complex (CPC; Borealin-Survivin-INCENP-Aurora B kinase) ensures accurate chromosome segregation by orchestrating sister chromatid cohesion, error-correction of kinetochore-microtubule attachments and spindle assembly checkpoint. Correct spatiotemporal regulation of CPC localization is critical for its function. Phosphorylations of Histone H3 Thr3 and Histone H2A Thr120 and modification-independent nucleosome interactions involving Survivin and Borealin contribute to CPC centromere enrichment. However, mechanistic basis for how various nucleosome binding elements collectively contribute to CPC centromere enrichment and whether CPC has any non-catalytic role at centromere remain open questions. Combining a high-resolution cryoEM structure of CPC-bound H3Thr3ph nucleosome with atomic force microscopy and biochemical and cellular assays, we demonstrate that CPC employs multipartite interactions, which facilitate its engagement at nucleosome acidic patch and DNA entry-exit site. Perturbing the CPC-nucleosome interaction compromises chromatin protection against MNase digestion in vitro, centromeric chromatin stability and error-free chromosome segregation in vivo. Our work suggests a non-catalytic chromatin stabilizing role of CPC in maintaining centromeric chromatin features critical for kinetochore function."

As already stated in our response to point no. 9 from reviewer 1, we have rephrased sentences referring to 'dynamic' interactions (not observed in the cryoEM structure but that contribute to CPC-NCP binding based on biochemical and cellular studies) as 'conformationally heterogeneous' interactions.

(2)-Introduction, first and third paragraphs: The CPC occupies at least two separate locations within the centromere, including one that is kinetochore proximal and one that is more "central", and that reflect interactions with BUB1 and Haspin substrates, respectively. There is almost no evidence that the SAC is controlled from the centromere, and there is very minor evidence that biorientation is controlled from the centromere, and therefore neither contribution merits the adjective "essential". Furthermore, depletion of the CPC has no effect on sister chromatid cohesion either, and only Sgo1 depletion or mutation, but not its mislocalisation, affect cohesin. This is not to say that the problem studied here is not important (I believe it is), but I am not convinced that the authors' arguments to justify its importance are entirely based on established facts.

We broadly agree with the reviewer. Published reports in the field on the 'essentiality' of CPC to sister chromatid cohesion and error correction are contentious. As per the reviewer's suggestion, we will rephrase these statements as follows:

Introduction paragraph 1: We have replaced the word 'essential' with 'contributing to'. *"The inner centromere, by recruiting several enzymatic activities (kinases, phosphatases and motor proteins) serves as a signaling platform contributing to the regulation of sister chromatid cohesion, kinetochore-microtubule attachments and spindle assembly checkpoint (SAC), processes that are fine-tuned by the interplay between the inner centromere associated kinases and phosphatases (Biggins & Murray, 2001; Musacchio, 2010; Funabiki & Wynne, 2013; Hengeveld et al, 2017; Grieco & Serpico, 2020; Valles et al, 2024)"*

Introduction paragraph 3: *"During prometaphase and metaphase, CPC is enriched at the inner centromere, where it is implicated in regulating sister chromatid cohesion and destabilizing faulty*

kinetochore-microtubule attachments (known as error-correction) (Bishop & Schumacher, 2002; Honda et al, 2003; Carmena et al, 2009; Hindriksen et al, 2017; Hengeveld et al, 2017; Haase et al, 2017)."

(3)-Introduction, fourth paragraph: "...Borealin-mediated multivalent interactions..." It is unclear if the authors are referring to the multipartite interaction mode described in the Abstract, or to the purported dimerization of Borealin. A note on this below.

We apologise that this was not clear. We agree with the reviewer that 'multivalent' is not the best choice of word here; we have replaced this with 'multipartite' in page 3 and page 8.

(4)-Figure 1, panels B and C: labels for the N- and C-termini and approximate positions in the overall view, i.e. not only in the inset, would facilitate the interpretation of these panels.

We have now labelled the N-termini of Borealin, INCENP and Survivin in Figure 1B. Please find Figure 1 attached in the response to point 2 by reviewer 1.

(5)-Results: "This observation suggests that multivalent and dynamic interactions involving different regions of Borealin and INCENP, not involving the nucleosome acidic patch, are essential for CPC-NCP binding (Abad et al, 2019; Serena et al, 2020). These dynamic interactions, likely involving protein-protein and protein-DNA contacts, may facilitate high-affinity binding of CPC to nucleosomes." This sentence lacks clarity. First, I would recommend defining multivalent (throughout the manuscript). It is not synonymous to multipartite, the term used in the abstract. If used as synonym, please clarify it in the text, but I would limit 'multivalent' to conditions created by oligomerization through identical or strictly similar modules. I feel that 'multipartite' is perfect for the case described here. Second, it may not be obvious to many readers that by 'dynamic' they imply that it is not observed in the density maps. If that is what justified the use of 'dynamic', it should be stated clearly.

As we already noted above, we completely agree with the reviewer that 'multipartite' is the appropriate word to describe the nature of CPC-NCP interaction. We have incorporated this change throughout the manuscript.

We also agree that what we mean by 'dynamic interaction' may not be obvious to the readers without a clear definition. As explained in our response to point no. 9 of reviewer 1 and point no. 1 of this reviewer, we have rephrased sentences referring to 'dynamic' interactions.

Third, the authors have not shown that the dynamic interactions they refer to are essential, but rather that they are sufficient in the absence of interactions from the Borealin N-terminal tail.

We agree with the reviewer. We have now assessed if the 'dynamic' interactions involving Borealin and INCENP are essential for CPC-NCP binding by testing the NCP binding ability of the CPC version lacking Borealin loop region and INCENP RRKRR motif through EMSAs. In our EMSA experiments, CPC_{I1-190SBΔloop} (containing the Borealin loop deletion) shows a 12-fold reduction in H3T3ph Widom 601 NCPs compared to wt CPC_{I1-190SB}, while mutation of the RRKRR motif to Alanines (CPC_{I1-190 6ASB}) does not seem to have an effect on NCP binding. Interestingly, when we perform the EMSAs with H3T3ph nucleosomes with α-satellite DNA, CPC_{I1-190SBΔloop} weakens the affinity for NCPs 2-fold, while adding the RRKRR motif mutation (CPC_{I1-190 6ASBΔloop}) weakens the affinity by 13-fold. This data together indicates that the nucleosomal interactions mediated by the Borealin loop region and the INCENP RRKRR motif are important for high-affinity CPC-NCP binding and that the contribution of the INCENP RRKRR motif might be DNA sequence-specific.

In the revised manuscript the corresponding text on Page 7 now reads as: "Altogether, these observations suggested that in addition to the CPC-NCP interactions resolved in the cryoEM structure, previously well characterized phosphorylated Histone H3 tail interaction with Survivin (Abad et al,

2019), and conformationally heterogeneous interactions involving different regions of Borealin and INCENP may also contribute to efficient CPC-NCP binding”

Fourth, the PIs should clarify why they don't think that the interaction with the phosphorylated N-terminal tail of H3 is sufficient for the residual binding they observe when the interaction at the acidic patch is eliminated.

We agree with the reviewer that Survivin interaction with phosphorylated N-terminal tail of H3 will likely contribute the CPC-NCP interaction. As discussed in previous response, our revised text on Page 7 accommodates this now: “Altogether, these observations suggested that in addition to the CPC-NCP interactions resolved in the cryoEM structure, previously well characterized phosphorylated Histone H3 tail interaction with Survivin (Abad et al, 2019), and conformationally heterogeneous interactions involving different regions of Borealin and INCENP may also contribute to efficient CPC-NCP binding.”

(6)-Figure 3B: Shouldn't the profile with the longer Borealin construct be also shown? Is any loss of affinity evident?

We have included the SEC profile and the corresponding SDS-PAGE of CPC_{11-190SB} containing INCENP 1-190, Survivin full length and Borealin full length in Figure 3B as suggested. Please find the new Figure 3 below:

As noted in our response to points 10 and 11 of reviewer 1 and point 5 of this reviewer, we have now quantitatively measured the NCP binding affinities of different CPC mutants. Thus, we have included the following paragraph (page 6 and 7 of the MS) describing the contribution of the N-terminal 10 residues of Borealin for CPC-NCP binding affinity: “We then quantitatively evaluated the contribution of Borealin N-terminal tail - NCP acidic patch (CPC_{11-190SBΔN}, CPC_{11-190SBΔN K12A}) and the Borealin helix-DNA entry-exit site (Borealin K26/R30/K37A) interactions, separately and in combination (CPC_{11-190SB3A} and CPC_{11-190SBΔN K12A 3A}), for CPC-NCP binding by performing EMSAs and Surface Plasmon Resonance (SPR) (Fig. EV5B and C). Our data consistently show that the CPCs containing the Borealin N-terminal tail mutants (CPC_{11-190SBΔN} and CPC_{11-190SBΔN K12A}) bind NCP relatively more weakly compared to CPC with Borealin helix mutant (CPC_{11-190SB3A}) (Fig. 3C-E and EV6A and B).

However, none of these Borealin mutants, either in isolation or in combination, abolished CPC-NCP binding.

Altogether, these observations suggested that in addition to the CPC-NCP interactions resolved in the cryoEM structure, previously well characterized phosphorylated Histone H3 tail interaction with Survivin (Abad et al, 2019), and conformationally heterogeneous interactions involving different regions of Borealin and INCENP may also contribute to efficient CPC-NCP binding."

(7)-Figure 4A: Apologies but I seem to have missed the point: are these Coomassie-stained gels? What is the cause of the high background in the uncrosslinked samples? The inputs seem very clean, is it background crosslinking in the absence of UV activation? And on what bands are the ratios shown in the histograms calculated?

We apologise that this was not clear. The representative SDS PAGEs shown in Fig. 4A of the original manuscript were imaged with a Licor Odyssey scanner with the 700 nm channel to visualise the DNA (nucleosomes were reconstituted with IR700-labelled 601 Widom 147 bp DNA). The samples visualised in the gels correspond to the samples that were submitted for the MS analysis shown in the right panel. The signal observed above the IR700-labelled Widom in the uncrosslinked conditions is background signal. In an SDS-PAGE, DNA can often appear as a smear or different bands, depending on the properties of the DNA and the amount of DNA loaded (in these experiments, in order to visualise the UV-crosslinked bands, we loaded 3ug of Nucleosomes per reaction). The histograms were calculated from the MS analysis of the UV-crosslinked samples vs the uncrosslinked samples to compare the number of peptides detected for Borealin and INCENP through MS. Our analysis showed that in the UV-crosslinked sample, the number of Borealin peptides detected by MS is lower than the number of INCENP peptides, indicating that Borealin might have more modified peptides (UV-crosslinked and nucleotide-modified peptides).

However, we have now performed new UV crosslinking experiments combined with MS analysis to map the regions of CPC interacting with nucleosomal DNA. This resulted in the identification of peptides of Borealin/INCENP crosslinked with DNA. As the new data is clearer compared to the data we showed in the first version of the MS, we are now only showing the new data (please find new Fig. 4B below) and we have added this paragraph to the results section (page 7 and 8): *"To understand how CPC is interacting with nucleosomal DNA, we stabilized protein-DNA interactions within the CPC-NCP complex through UV crosslinking and used MS analysis (Stützer et al, 2020) to map the protein-DNA interaction sites. MS analysis identified DNA crosslinked peptides for both Borealin and INCENP (Fig. 4B and EV6D). All the DNA-crosslinked peptides of Borealin map to a region spanning the end of the Borealin N-terminal α -helix until the N-terminal half of the Borealin loop region. This is consistent with our previous SPR data, which suggested Borealin amino acid residues 110 to 188 as a region capable of directly binding DNA (Abad et al, 2019). The DNA-protein contacts in INCENP spanned not only IDRs, but also the N-terminal α -helix, which is part of the triple helical bundle formed by Borealin, Survivin and INCENP, suggesting the capability of this helical bundle in making transient inter-nucleosomal contacts via INCENP under favourable conditions (Fig. 4B and EV6D). Our analysis also identified a DNA-crosslinked peptide spanning residues 71-83, which is adjacent to the 'RRKKRR' motif, suggesting the capability of this region to interact with DNA (although we could not identify any DNA-crosslinked peptides involving the 'RRKKRR' motif, possibly due to the presence of a large number of trypsin cleavage sites). Overall, this data suggests that DNA interactions involving both Borealin and INCENP also contribute to CPC-nucleosome binding."*

B

(8)-Results: How do the authors arrive to the conclusion that the Borealin Δ loop construct has a 3-fold decrease in affinity? Is this based on the gels in Figures

We apologise that the quantifications were not clear. As stated in the figure legend, the quantification CPC-NCP binding was originally carried out using the reaction performed with 160 nM CPC. We have now updated this figure with the new EMSAs, where a concentration series of the CPC proteins ranging from 7.8 nM to 2 μ M was used to calculate the affinity between different CPC constructs and NCPs. Unbound nucleosome bands were quantified using ImageJ and data plotted in GraphPad Prism 10.0 with a log₁₀ x-axis and an isotherm fitted using the 'specific binding curve with Hill slope' fitting. The equation for this fitting is : $Y = B_{max} * X^h / (Kd^h + X^h)$, where Y is the specific binding, X is the protein concentration, B_{max} is the maximum fraction bound, K_d is the dissociation constant, and h is the Hill slope. This information is now included in the material and method. Please find the new Figure 4 below:

(9)-Results: "Altogether, our structural and biochemical analysis shows that CPC-NCP binding is mediated by multipartite interactions involving both stable and dynamic interactions. This, along with our observation that CPC engages with the nucleosome DNA entry-exit site on both faces of the NCP suggests that CPC likely stabilizes chromatin by protecting the wrapped state of nucleosomal DNA." This refers again to the somewhat arbitrary and rather unconvincing definition of stable and

dynamic. If they want to use this distinction, could they at least indicate which interactions are dynamic and which static, and what is the evidence support this distinction?

As explained in our response to point 9 (reviewer 1) and point 3 and 5 (reviewer2) this has now been addressed.

(10)-Figure 6B: very hard to take these data to support the authors' conclusion of protection in vivo.

We have now performed additional replicates of the in vivo MNase assay in RPE1 cells and quantified the level of protection contributed through the CPC-NCP interaction. Please find below the new figure (Fig. 6A):

Minor points

(11)-For future reference, adding page and line numbers to the text and figure numbers to the figures facilitates the reviewers' job!

We apologise for the inconvenience caused. We have included this in the revised manuscript.

(12)-Introduction: '...what CPC's potential...' please check the grammar of this sentence.

We have rewritten the sentence as follows: '...we still do not understand how various nucleosome-binding elements of the CPC subunits cooperatively allow chromatin binding and what the potential role of CPC is in preserving chromatin structure and integrity' (page 3, line 99) and we have updated this sentence in the revised version of the manuscript.

(13)-In the Bishop & Schuniacher 2002 reference the correct author's name is Schumacher.

Thank you for the point. We have now corrected this.

(14)-Results, first two lines: As the authors indicate that the loop region encompasses residues 110-206, they could do so for the N-terminal tail too. Else, they could simply refer to S1 for both.

We have now specified in the text that the N-terminal tail of Borealin encompasses amino acid residues 1-10 (as indicated in Fig. EV1) and in the text of the revised manuscript we will refer to Fig. EV1 where appropriate.

(15)-Results: there is a call to Figure 1C but no panel 1C in figure 1, as far as I can tell.

We apologise if we are getting this point wrong. We confirm that panel 1C, showing the side view of the CPC-NCP structure and a close-up view of the CPC triple helical bundle interaction with the entry-exit site of the DNA, is present in Figure 1.

(16)-Results, lines 5-6: "However, how the different...remains an open question". This has already been said in the Introduction, so I would simply connect to the next sentence: "To address how different nucleosome binding elements of CPC collectively contribute towards CPC-nucleosome binding and chromosome association, we purified recombinant..."

Thank you for the comment. We have revised the sentence as suggested (page 4, line 124).

(17)-"To contact" is transitive, i.e. you don't contact with residue X, you contact residue X

We have corrected this throughout the text.

(18)-"We also observed a comparable decrease in DNA protection when we perturb..." 'perturbed' would be more appropriate.

We have corrected this.

Referee #3:

Gireesh et al. present insightful analyses of the chromosomal passenger complex (CPC), proposing that CPC-nucleosome interactions contribute to centromeric chromatin protection independently of the kinase activity of Aurora B, the catalytic subunit of the CPC. Combining high-resolution cryo-EM, crosslinking mass spectrometry, biochemical reconstitution, AFM, and cell-based chromatin assays, the authors provide evidence that the CPC engages nucleosomes through both stable (H2A/H2B acidic patch binding) and dynamic (DNA/histone tail interactions), and that these interactions all contribute to "protection" of chromatin - measured by openness of the nucleosomes. Mutations that compromise the CPC-DNA interaction leads to mitotic delays, supporting the importance of the non-catalytic role of the CPC on the centromere integrity, though it does not rule out the possibility that the defect is caused by compromised centromeric enrichment of Aurora B.

The manuscript is well executed, and the major conclusions are mostly supported by the presented data, except for the difficulty in demonstrating the physiological importance of non-catalytic functionality of the CPC in centromeric chromatin protection. The only mutant that the authors tested its functional importance in cells was INCENP-6A mutant, where INCENP basic residues that are supposed to interact with DNA was mutated. Unfortunately, these INCENP basic residues were not visualized in the presented cryo-E structure. The structure revealed a novel interaction between Borealin N-terminus and the acidic patch of the nucleus, but physiological importance of this interaction was not tested. Additionally, the structural analysis workflow is not explicitly organized or sufficiently detailed, particularly in terms of data processing rationale, which limits the interpretability of the cryo-EM findings. Overall, the manuscript presents compelling evidence that the CPC can limit chromatin accessibility in a manner independent of the catalytic activity of Aurora B in vitro. This is an important advancement to understand the mechanism by which the CPC engages and potentially modulates the centromeric chromatin architecture, while demonstrating its functional importance in vivo is challenging.

Major points:

1. Table 1. Please include the resolution of the refined atomic model based on the FSC criterion, in addition to the existing map resolution metrics. This information is essential for evaluating the quality and interpretability of the structural model.

	CPC-NCP Class	CPC-NCP Class	CPC-NCP Class	Double	Double
--	---------------	---------------	---------------	--------	--------

The resolution of the refined atomic model based on the FSC criterion was included in Table 1 in a row named 'Resolution' under the 'Refinement (Phenix)' section. We have now amended the table to include the FSC cut-off (0.5) used for the resolution of the atomic model and have renamed the 'Resolution' to 'Model Resolution' to make it clearer. We have deposited the EM maps and corresponding atomic models for all three classes (Classes 0, 1, and 2), as well as the map obtained through RELION processing and its atomic model. In addition, the cryoSPARC-derived map corresponding to the double occupancy has been deposited to the EMDB. The details of data collection and accession codes for all have been updated in Table 1 (see below).

Table1: Details of cryo-EM data collection and processing.

	0	1	2	occupancy 1	occupancy 2
Number of grids used	1	1	1	1	1
Grid type	Quantifoil R2/2 300 mesh	Quantifoil R2/2 300 mesh	Quantifoil R2/2 300 mesh	Quantifoil R2/2 300 mesh	Quantifoil R2/2 300 mesh
Microscope/detector	Titan Krios / Falcon4i	Titan Krios / Falcon4i	Titan Krios / Falcon4i	Titan Krios / Falcon4i	Titan Krios / Falcon4i
Voltage	300 kV	300 kV	300 kV	300 kV	300 kV
Magnification	165k	165k	165k	165k	165k
Recording mode	Counting mode	Counting mode	Counting mode	Counting mode	Counting mode
Dose rate	1 e ⁻ /Å ² /frame	1 e ⁻ /Å ² /frame	1 e ⁻ /Å ² /frame	1 e ⁻ /Å ² /frame	1 e ⁻ /Å ² /frame
Defocus	-0.5 μm to -2.6 μm (step size 0.3 μm)	-0.5 μm to -2.6 μm (step size 0.3 μm)	-0.5 μm to -2.6 μm (step size 0.3 μm)	-0.5 μm to -2.6 μm (step size 0.3 μm)	-0.5 μm to -2.6 μm (step size 0.3 μm)
Pixel size	0.727	0.727	0.727	0.727	0.727
Total dose	40 e ⁻ /Å ²	40 e ⁻ /Å ²	40 e ⁻ /Å ²	40 e ⁻ /Å ²	40 e ⁻ /Å ²
Number of frames/movie	40	40	40	40	40
Total exposure time	Adjusted to keep total dose stable	Adjusted to keep total dose stable	Adjusted to keep total dose stable	Adjusted to keep total dose stable	Adjusted to keep total dose stable
Number of micrographs	22,065	22,065	22,065	22,065	22,065
Number of micrographs used	16971	16971	16971	16971	16971
Number of particles used	73,078	74,429	75,653	75,653	28,398
PDB	9SI9	9SJ5	9SI3	9SLJ	-
EMDB	EMD-54926	EMD-54938	EMD-54924	EMD-55003	EMD-55012
Map resolution (FSC 0.143)	2.86	2.85	2.83	3.8	3
Refinement (Phenix)					
Model Resolution (Å) (FSC 0.5)	3	3	3	3.2	-
Map CC	0.88	0.87	0.87	0.87	-
Mean B factor (Å ²)	90.13 (Protein) 83.89 (Nucleotide)	100.54 (Protein) 92.49 (Nucleotide)	95.34 (Protein) 90.97 (Nucleotide)	358.08 (Protein) 120.65 (Nucleotide)	
Validation					-
All atom clashscore	8.12	8.35	5.10	6.49	-
Rotamer outliers (%)	1.01	0.9	0.00	2.99	-
MolProbity score	1.63	1.56	1.44	1.91	-
Ramachandran plot					
Favored (%)	96.88	97.46	97.06	96.9	-

Outliers (%)	0.00	0.00	0.00	0.00	-
RMS deviation					
Bond length (Å)	0.005 (0)	0.004 (0)	0.004 (0)	0.004 (0)	-
Bond angle (°)	0.621 (0)	0.594 (1)	0.803 (1)	0.654 (2)	-

2. Fig. S2. The data processing workflow should be expanded to include greater detail. Specifically, please indicate the symmetry applied during 3D reconstruction and show a representative cryo-EM micrograph. Notably, the final map was generated using 73,078 particles from 12,436 micrographs, suggesting that the majority of particles were discarded. The workflow should address the rationale for this, such as potential structural flexibility of the CPC-nucleosome complex, and detail how particles were classified and refined. This is important to get a sense of whether the configuration where two CPC molecules interact with a nucleosome represents a minor fraction of the reconstituted molecules. The stoichiometry of the CPC and the nucleosome may be addressed by the SEC (Fig. 3B).

As discussed in the response to points 3 and 5 of Reviewer 1, we have included figures detailing the data processing workflow (new Fig. EV2 and EV3). The processing workflow indicates that 100 % of the 1.2M particles show different levels of CPC density, among which 220k particles (18 %) show a well-defined CPC density on one side and 85.5k particles (6.67 %) show reasonably two well-defined CPC (one in each face of the nucleosome). Please find the Fig. EV2 and EV3 attached below:

Fig. EV2

Fig. EV4D

Fig. EV3

As discussed in the response to point 16 of reviewer 1, our Mass Photometry analysis confirms the presence of two CPCs within the CPC-NCP complex used for cryoEM structure determination. The mass photometry data is now included in the Fig. EV1D. Please find the figure attached below:

3. Fig. S2c. Please show the structure in a different viewing angle. It is not clear if another CPC molecule associates with the nucleosome on the other side.

We have now made new figures (attached in response to comment 5 from reviewer 1) depicting both faces of the nucleosome. These have been added to the supplementary file in the revised manuscript, Fig. EV4E.

4. Fig. 1B. In this presentation, it is impossible to evaluate the amino residue assignments and the side-chain interactions. The figure can be split at least in two, one presenting an atomic model to

illustrate the side-chain interactions and the other presenting the map-overlaid model to justify the side chain assignments. Such an example can be seen in Fig. S7 of PMID 39088653.

As can be seen in our response to point 6 of reviewer 1, we have now included more ‘zoomed-in’ images of the two Arginines (R4 and R9) and Lysine K12 of Borealin contacting the nucleosome acidic patch. These have been included in the supplementary file, new EV5A.

5. Figure 5E. While the authors report that deleting the Borealin N-terminus that is important for acid patch interaction reduced nucleosome binding, this is only seen for the 601 nucleosome but not for the alpha-satellite nucleosome. Please discuss the implications of this result given that the acidic patch mutant does not impact the NCP binding of the CPC in Fig 3A.

We thank the reviewer for their careful assessment. We have now performed both EMSAs and SPR experiments to assess the contribution of the N-terminal tail of Borealin for NCP binding. Our analysis shows that deleting the N-terminus of Borealin weakens the affinity for H3T3ph 601 Widom NCPs 5-fold, while in H3T3ph α -satellite NCPs it only weakens it 2-fold. We have now included the following text to the results section (page 6): “We then quantitatively evaluated the contribution of Borealin N-terminal tail - NCP acidic patch ($CPC_{I1-190SB,\Delta N}$, $CPC_{I1-190SB,\Delta N K12A}$) and the Borealin helix-DNA entry-exit site (Borealin K26/R30/K37A) interactions, separately and in combination ($CPC_{I1-190SB3A}$ and $CPC_{I1-190SB,\Delta N K12A 3A}$), for CPC-NCP binding by performing EMSAs and Surface Plasmon Resonance (SPR) (Fig. EV5B and C). Our data consistently show that the CPCs containing the Borealin N-terminal tail mutants ($CPC_{I1-190SB,\Delta N}$ and $CPC_{I1-190SB,\Delta N K12A}$) bind NCP relatively more weakly compared to CPC with Borealin helix mutant ($CPC_{I1-190SB3A}$) (Fig. 3C-E and EV6A and B). However, none of these Borealin mutants, either in isolation or in combination, abolished CPC-NCP binding.

Altogether, these observations suggested that in addition to the CPC-NCP interactions resolved in the cryoEM structure, previously well characterized phosphorylated Histone H3 tail interaction with Survivin (Abad et al, 2019), and conformationally heterogeneous interactions involving different regions of Borealin and INCENP may also contribute to efficient CPC-NCP binding.”

6. Many of the bar graphs do not define the sample numbers and the error bars (Fig. 4, Fig. 5, Fig. 6). It would be recommended to plot each data point and show data distribution.

We have now defined the sample numbers in the figure legends and, when required, plotted each data point to show data distribution.

7. Figure 4A, left panel. I cannot see the band that representing the interaction between CPC I1-190SB and CPC I1-190-6ASB. It seems odd that the clear band is much clearer seen with Δ loop, which is supposed to decrease the CPC-nucleosome interaction, according to Fig. 4B

As noted in the response to point 7 of reviewer 2, the samples visualised in the gels correspond to the samples that were submitted for the MS analysis shown in the right panel of the former Fig 4A. We agree that, based on the expected MW, we could not clearly assign a band corresponding to INCENP UV-crosslinked to DNA (it should be around the MW of the Borealin Δ loop band). The only clear band that appeared when we compared the UV-crosslinked NCP/DNA with and without CPC is the one corresponding to Borealin (mainly based on the downward shift of the band in the sample containing CPC containing Borealin Δ loop). As the reviewer pointed out, the loop of Borealin is one of the regions of Borealin we believe might be interacting with DNA. However, as observed in the cryoEM structures, Borealin residues K26, R30, K37, E40 and Q44 contact the DNA entry-exit site and might explain the DNA-crosslinking ability of Borealin Δ loop.

However, as discussed in our response to Reviewer no. 2, we have now performed new UV crosslinking experiments combined with MS analysis (following the workflow published in Stutzer *et al.*, Nat Communications 2020) to map the regions of CPC interacting with nucleosomal DNA. This resulted in the identification peptides of Borealin/INCENP crosslinked with DNA. As the new data is clearer compared to the data we showed in the first version of the MS, we are now only showing the new data (please find new Fig. 4B below):

8. Figure 4B and C. Please indicate the concentration of the CPC used. It is not clear which lane represents the 160 nM condition analyzed in the right panels. Please describe how the quantitation was done, and what is the basis for "a 3-fold decrease in affinity" (page 7).

We apologise that this was not specified in the text. In line with our response to point 8 of reviewer 2, we have now performed new EMSA analysis of the different CPC mutants. The new figure can be found in response to point 8 of reviewer 2.

9. Figure 5C. It would be informative if the distribution is reproducible between the two independent experiments. In methods, please describe how the CPC was added to the nucleosome. Close examinations of the AFM images indicate that several nucleosomes do not have clear linker DNAs, making it difficult to analyze their linker DNA angles. How did you analyze them? I can also see many fragmented DNAs, indicating that the estimated wrapped lengths could be affected by DNA fragmentation. It would be informative to measure lengths of naked DNA in each condition. If the

CPC treatment artificially induces DNA fragmentation, you may see more fragmented DNA. Conversely, if the CPC indeed protects the nucleosome, you may see less fragmented DNA. In any case, such analysis will tell you if the apparent differences between the experimental conditions are not due to technical issues related to the exposure of the nucleosome to the mica.

We thank the reviewer for this comment. We can confirm that the fractions of fully wrapped nucleosome states are reproducible between the two independent experiments. We observe that the fraction of fully wrapped nucleosomes is significantly higher in the presence of 40 nM CPC_{11-190SB}. We have now included the following panel in supplementary figure EV7F:

F

We have now explicitly explained the sample preparation in the revised version of the methods section (page 34).

We agree that a certain fraction of nucleosomes is end-bound in our samples. This phenomenon is well-understood for systems with long extranucleosomal DNA such as is the case here. Due to the intrinsic bending stiffness of DNA, long extranucleosomal DNA must cross or come in close proximity. This presents an energy penalty, due to electrostatic repulsion. When bound near the end of the DNA construct (and away from the positioning sequence), this penalty is removed as DNA does not need to cross. Our software automatically discards these end-bound nucleosomes and exclusively focuses on those nucleosomes where extranucleosomal DNA is automatically detected at each end of the nucleosome. We will explicitly mention the case of end-bound nucleosomes in the revised version of the Methods section.

We also agree that our sample includes some fragmented DNA. We have now quantified the bare DNA lengths in the absence and presence of CPC_{11-190SB} and our results show that there is no CPC-induced fragmentation (see figure below). DNA deposition on poly-lysine coated mica should not introduce any DNA breaks either; this type of surface deposition very successfully preserves the topology of supercoiled plasmids which are very sensitive to even single strand breaks (see our previous work published as Brouns *et al.*, ACS Nano 2019; Kolbeck *et al.*, Nucleic Acids Research 2024). In addition, to minimize the effect of DNA fragments on calculation of wrapped lengths in nucleosomes, we employ the mode instead of the mean of the distribution. We have added the panel below in the supplementary figures (Fig. EV7E) and we have added this information in the revised version of the Methods section.

10. Figure 6. Although the title of the figure legends state, "CPC-mediated protection of centromeric chromatin is crucial for accurate chromosome segregation", chromosome segregation accuracy was not reported in this figure. From the represented images shown in D, it seems that chromosome segregation was normal in 6A mutant. I assume that the authors must have data to quantitatively analyze chromosome segregation errors. It would be also informative to report the duration between NEBD to metaphase alignment. In any case, from this experiment, it would be difficult to attribute the phenotype to the defect in CPC-mediated protection of centromeric chromatin, since the amount of INCENP on the centromere was reduced.

We thank the reviewer for this valuable suggestion. As suggested, we have quantitatively analysed chromosome segregation errors. Please find below the new Fig. 6E:

And we have added the following text to the results section: "Notably, analysis of segregation errors in RPE1s released from a monastrol-induced mitotic arrest showed that the INCENP 'RRKRR' mutant

and the structure-based Borealin mutants all led to an increase in cells with chromosomes segregation errors during anaphase (Fig. 6E).”

We also measured the inter-kinetochore distance (new Fig. 6D) and quantified the duration between NEBD to metaphase alignment and generated the new Fig. EV10A (please find it below):

Fig. 6D

Fig. EV10A

As observed in Fig. EV10A the duration between NEBD and metaphase alignment of the INCENP or Borealin mutants is comparable to INCENP wt and Borealin wt, respectively. However, the timing between the chromosome alignment and anaphase onset is increased in the case of the INCENP 6A and Borealin 5A (mutating the Borealin helix residues contacting the DNA entry-exit site) mutants.

We agree that the loss of centromeric chromatin protection we observe for the 6A mutant could be due to the reduced centromere association of the CPC. However, Borealin 5A, although it shows only a very moderate reduction in the centromeric levels of CPC (compared to the Borealin wt, Fig. EV9F), exhibits perturbed inter-kinetochore distance (and increased chromosome segregation errors. This strengthens our notion that CPC mediated chromatin protection may likely contribute to maintaining centromeric chromatin features critical for withstanding microtubule-pulling forces. We now discuss this on pages 10 and 11 of the revised manuscript as follows (page 11):

“Thus, to further understand the role of CPC for centromeric chromatin organization, we labelled CENP-A and CENP-B and quantified the inter-kinetochore distance in RPE1 cells expressing either the INCENP mutant or the structure-guided Borealin mutants (Fig. 6D). Our data shows that INCENP 6A and Borealin 5A mutants show altered inter-kinetochore distance (Fig. 6D). While the INCENP ‘RRKKRR’ mutant (INCENP 6A) showed an increase in inter-kinetochore distance, Borealin 5A (perturbing the Borealin helix – DNA entry-exit site interaction) resulted in a decrease in inter-kinetochore distance (Fig. 6D). In contrast, the Borealin DN mutant did not show any detectable variation in the inter-kinetochore distance (Fig. 6D). It is important to note that, unlike INCENP 6A and Borealin 5A, Borealin DN retains likely all DNA-interacting regions, suggesting that CPC exerts its role in maintaining correct centromere features mainly via its DNA-binding properties in vivo. The observed difference in the phenotypes for INCENP 6A and Borealin 5A (increase vs decrease in inter-kinetochore distance, respectively; Fig. 6D) could be due to the differential effect of these mutants on centromeric chromatin. We speculate that INCENP 6A impacts the inner centromeric chromatin, compromising centromeric chromatin’s ability to withstand spindle-associated pulling forces, while Borealin 5A impacts the kinetochore proximal centromeric chromatin, compromising the ability of the kinetochore to transfer the microtubule-associated force to centromeric chromatin. Notably, analysis of segregation errors in RPE1s released from a monastrol-induced mitotic arrest showed that the INCENP ‘RRKKRR’ mutant and the structure-based Borealin mutants all led to an increase in cells with chromosomes segregation errors during anaphase (Fig. 6E).”

11. Since condensin depletion is known to mitotic delay through compromising the mechanical tension applied to the centromeric chromatin (PMID 19188492), it would be informative to measure the inter-kinetochore distance in 6A cells to see if 6A compromises chromatin integrity at the centromere.

We thank the reviewer for the suggestion. As responded to point 10 of this reviewer, we have now measured the changes in inter-kinetochore distance for INCENP 6A and Borealin 5A. As shown in the new Fig. 6D, while INCENP 6A resulted in an increased inter-kinetochore distance, the Borealin 5A resulted in decreased inter-kinetochore distance. The observed difference in phenotypes for INCENP 6A and Borealin 5A (increase vs decrease in inter-kinetochore distance, respectively) could be due to the differential effect of these mutants on centromeric chromatin. We speculate that INCENP 6A impacts the inner centromeric chromatin, compromising centromeric chromatin’s ability to withstand spindle-associated pulling forces. In contrast, Borealin 5A impacts the kinetochore proximal centromeric chromatin, compromising the ability of the kinetochore to transfer the microtubule-associated force to centromeric chromatin. As can be seen in our response to point 10, we have updated the section **‘CPC-nucleosome interaction contributes to centromeric chromatin protection’** with a revised text (on pages 10 and 11).

12. Figure 6B. This is the only data to show that INCENP 6A mutant increases centromeric chromatin accessibility. However, the effect is subtle. It would be important to make quantitative analysis with multiple experiments to support the non-catalytic role of the CPC on chromatin protection. Ideally, an additional experiment with a different methodology is advised, to support the claim that the CPC protects centromeric chromatin. Without solidifying this part, the conclusion, described as a

headline in page 8, 'CPC-nucleosome interaction is essential for centromeric chromatin protection', must be softed.

We have now quantified the level of centromeric chromatin protection conferred by INCENP wt as compared to INCENP 6A by including more replicates of MNase assay and generated Fig. 6A and B (please find them below). We have also included Borealin mutants in the same assay to strengthen our conclusion.

And we have also included the following text in the results section (p. 10): “Based on our structural and in vitro biochemical data presented above, we hypothesized that CPC-Nucleosome binding may be critical for maintaining correct centromeric chromatin structure and compaction. To test this, we generated RPE1 cell lines expressing mNeonGreen-Borealin in which endogenous INCENP could be replaced either with FLAG-INCENP wild type or FLAG-INCENP 6A mutant or RPE1 cell lines expressing Borealin-mClover in which endogenous Borealin could be replaced either with Borealin wt-mCherry, Borealin lacking the N-terminal tail (Borealin DN-mCherry) or Borealin helix mutant (Borealin K26/R30, K37, E40, Q44A-Borealin 5A-mCherry, Fig. EV 9A and B) and performed MNase digestion of chromatin on prometaphase lysates. Supporting our hypotheses, cells expressing INCENP 6A, Borealin DN and Borealin 5A, all showed reduced centromeric chromatin protection against MNase digestion (Fig. 6A, B, EV9C, D and G).”

We have also changed the title of the section as follows “**CPC-nucleosome interaction contributes to centromeric chromatin protection**”

13. Discussion can be expanded, for example, by including the potential implications of the acidic patch's role in the CPC binding, and the mechanism by which 6A mutant causes the metaphase delay. The possibility that the effect of 6A mutant is indirect must be discussed.

We thank the reviewer for this suggestion. We have now expanded the discussion as suggested on pages 12 to 13 within the ‘Discussion’ section, and as indicated in the response to point 10 of this reviewer, we have included further discussion on how chromatin protection of CPC may contribute to achieving error-free chromosome segregation on pages 10 and 11 within the section “**CPC-nucleosome interaction contributes to centromeric chromatin protection**”

Minor points:

1. Please include the page numbers, and figure numbers in the figure display. Unless it is instructed by the journal, I prefer to have figure legends associated with the figures. Since figure legends are

placed after Discussion and Methods, it was cumbersome to relate the main text, figure legends, and the figure in the current arrangement.

As suggested, we have included page numbers and figure numbers to the main text.

2. Please clearly explain the acronyms for the constructs, such as CPC I1-190_6ASB and INCENP 6A.

We have now defined the constructs as requested in the figure legends and created a panel in supplementary information (Fig. EV5B) to depict the different mutants.

3. Fig 1A. By this presentation, the readers may think that the structure of the entire colored segments are solved and shown in B, although only the portions of each subunit were resolved in B. It will be helpful to modify the diagram to clearly indicate the segments that were solved by the structure so that readers can readily understand which segments support the stable and dynamic interactions with the nucleosome.

We have now highlighted the regions resolved in the cryoEM structure using a dotted box in the domain architecture schematic in new Fig. 1A. Please find Fig. 1A below.

4. Fig S5D. Please specify the AlphaFold prediction version and plot the confidence of prediction as well.

AlphaFold 3 was used to model the SGO motif bound to NCP. The AlphaFold 3 model coloured by confidence is attached below. The SGO motif binding prediction is low confidence ($50 < \text{pLDDT} < 70$). However, as the predicted model agrees with previously available biochemistry on the interfaces involved in SGO motif – NCP binding (Kawashima *et al*, 2010; Liu *et al*, 2013, 2015), we decided to refer to this model. We have explicitly included the confidence diagram (please find the diagram below) as Figure EV9H.

5. Fig. 3C, right panel. The labeling text overlaps with the DNA backbone, making the annotation difficult to read and visually cluttered. To improve clarity, please adjust the placement, size, or color contrast of the text accordingly.

We have now modified the figure (now Fig. 4A) accordingly (please see below).

6. Fig.S5D. Please label the position of H2A T120 accordingly to show how it interacts with the CPC tri-helical bundle.

We have now labelled the H2A C-terminal tail in the new Fig. EV9H. In our EDC crosslinking/MS analysis, the H2A T120 residue was identified to be making contact with the triple helical bundle (Fig. 4A). The model presented in the original manuscript Fig. S5D, is now Fig. EV9H in the revised manuscript. The atomic model (of Class 2) shown in this figure does not show a direct contact between the H2AT120 residue and the triple helical bundle. The H2A tail, however, is inaccessible for Sgo1 binding due to CPC binding (the CPC triple helical bundle occludes the accessibility of the H2A tail), as can be seen from the model shown in Fig. EV9H.

H

8. Typos.

Page 1. Phosphorylations -> Phosphorylation

Page 2. Spindle Assembly Checkpoint -> the spindle assembly checkpoint

Numerous places. CPC -> the CPC

We apologise for these typos. These have been amended.

Prof. A. Arockia Jeyaprakash
University of Edinburgh
Wellcome Trust Centre for Cell Biology, Institute of Cell Biology
Kings Buildings
Max Born Crescent
Edinburgh EH9 3BF
United Kingdom

21st Sep 2025

Re: EMBOJ-2025-121378R
Chromatin Protection by the Chromosomal Passenger Complex

Dear JP,

Thank you for submitting your revised manuscript to The EMBO Journal. It has now been re-reviewed by one of the original referees (see comments below), and I am happy to say that they considered the study substantially improved and all original concerns satisfactorily clarified. We are therefore from the scientific side ready to quickly proceed with acceptance and production of the article. However, there are at this point still a number of important editorial points that would need to be urgently addressed:

- Please upload text and figures separately. We need the manuscript text as a text-only file, which should also include all main and EV figure legends (and no more mark-ups, except tracked changes). On the other hand, we need one individual image file for each main and each EV figure. Each of these should be of sufficient resolution/quality for production, and not contain any legend text.
- Please adjust the order of the manuscript sections, and also make sure to use the correct section headers: Title page with complete author information, Abstract, Keywords, Introduction, Results, Discussion, Methods, Data Availability, Acknowledgements, Disclosure and Competing Interests Statement, References, Main Figure Legends, Tables, Expanded Figure Legends.
- Please make sure to specify the email addresses for corresponding authors on the title page.
- On the abstract page of the manuscript, please include 4-5 general keyword terms to enhance searchability.
- Please carefully go through the reference list, which currently contains many wrongly formatted entries:
 - * many citations are incomplete, missing e.g. citation year, volume, and page/locator numbers
 - * some are wrongly listed as "preprints" and amended with DOI, even though they are regularly published. DOI information is only required for "advance online publications" that really do not have a formal citation information yet
 - * Please adjust the format for citation of real preprints: The citation in the text should be: "(preprint: NAME1 et al, YEAR)"; and in the reference list: "NAME1, NAME2, ... (YEAR) article title. BIORXIV doi: XXX"
- Please double-check to make sure to all relevant funding information in the manuscript is congruent with the info entered into our submission system. Currently missing in the submission system are:
the Japan Society for the Promotion of Science (JSPS) Grant-in-Aid for Scientific Research (24H01381, 24H02286; 24H02283, 22H04996), and by JST, CREST Grant Number PMJCR21E6, Japan; Wellcome Discovery Research Platform for Hidden Cell Biology [226791]; the Proteomics core, Protein Production (EPPF), and the Atomic Force Microscopy facility; the Wellcome Trust (210493), Medical Research Council (T029471/1); Deutsche Forschungsgemeinschaft TTR237 and SFB1361; the Wellcome Trust (grant 203149)
- Please include a Disclosure and competing interests statement (next to the Acknowledgment section) - for details, see <https://www.embopress.org/competing-interests>
- As we are switching from a free-text author contribution statement towards a more formal statement based on Contributor Role Taxonomy (CRediT) terms, please remove the present Author Contribution section and instead specify each author's contribution(s) directly in the Author Information page of our submission system during upload of the final manuscript. See <https://casrai.org/credit/> for more information.
- Please combine the Source Data for the Expanded View Figures in one single ZIP archive. Those for the main figures are correctly uploaded as individual folders for each figure.

- In the Data Availability section, please add direct URLs to the PDB and EMDB repositories in which data generated in the study have been deposited; and make sure to initiate their public release at this stage.

- During routine pre-acceptance checks, our data editors have raised the following queries regarding figures, data, and legends, which I would ask you to address (ideally using the Track Changes option):

1. Please note that the box plots need to be defined in terms of minima, maxima, centre, bounds of box and whiskers, and percentile in the legends of figures EV9 E, F; EV10 A
2. Please note that information related to n (number of replicates) is missing in the legends of figures 3D, 4D, 5D
3. Figures 6A, B, D present data from only 2 biological replicates (n=2). Since no statistical tests or error bar can be meaningfully applied in such cases, please alter the presentation by plotting individual data points instead
4. Please note that the error bars need to be defined in the legends of figures 3D, 4D, (6A-C)
5. Please note that the measure of center for the error bars, as well as the statistical test used for data analysis, needs to be defined in the legend of figure 6E
6. Please define the annotated p values ****/**/**/* as well as provide the exact p-values for the same in the legend of figure 6E as appropriate.
7. Please note that the exact p values are not provided in the legends of figures 6D, EV9 E, F; EV10 A

- Please rename the included movie file as Expanded View movie (in-text callouts "Movie EV1/2/..."). Make sure to move the movie legend(s) from the main text into individual text files, each of which should be combined with the respective movie file into a separate ZIP file and uploaded as such.

- Please provide suggestions for a short 'blurb' text prefacing and summing up the study in two sentences (max. 250 characters), followed by 3-5 one-sentence 'bullet points' with brief factual statements of key results of the paper; they will form the basis of an editor-written 'Synopsis' accompanying the online version of the article.

- Finally, I would strongly encourage the use of a more explicit, less generic title, to ensure that the study will be appealing to a broad readership and experts alike. I am happy to discuss possible alternatives with you.

I am returning the manuscript to you for a final round of minor revision - hoping that you can resubmit a final version comprehensively and satisfactorily each of these points within the coming week, in order to facilitate that the paper can swiftly go into production.

With kind regards,

Hartmut

- size of the scale bars that are mandatory for all micrograph panels
- the statistical test used to generate error bars and P-values
- the type error bars (e.g., S.E.M., S.D.)
- the number (n) and nature (biological or technical replicate) of independent experiments underlying each data point
- Figures may not include error bars for experiments with n<3; scatter plots showing individual data points should be used instead.

4) Each main and each Expanded View (EV) figure should be uploaded as individual production-quality files (preferably in .eps, .tif, .jpg formats). For suggestions on figure preparation/layout, please refer to our Figure Preparation Guidelines:

9) To facilitate reproducibility and cross-laboratory adoption of methodologies, please structure the Materials & Methods section as outlined in our guide to authors, including a completed Reagents and Tools Table that can be downloaded from our author guidelines as well (<https://www.embopress.org/page/journal/14602075/authorguide#structuredmethods>).

10) Digital image enhancement is acceptable practice, as long as it accurately represents the original data and conforms to community standards. If a figure has been subjected to significant electronic manipulation, this must be clearly noted in the figure legend and/or the 'Materials and Methods' section. The editors reserve the right to request original versions of figures and the original images that were used to assemble the figure. Finally, we generally encourage uploading of numerical as well as gel/blot image source data; for details see: embopress.org/page/journal/14602075/authorguide#sourcedata

In the interest of ensuring the conceptual advance provided by the work, we recommend submitting a revision within 3 months (20th Dec 2025). Please discuss the revision progress ahead of this time with the editor if you require more time to complete the revisions. Use the link below to submit your revision:

Link Not Available

Referee #2:

I thank the authors for submitting a greatly improved manuscript. I am delighted to support its publication.